# Workflow enabling deepscale immunopeptidome, proteome, ubiquitylome, phosphoproteome, and acetylome analyses of sample-limited tissues

Jennifer G. Abelin [1,3] ✉, Erik J. Bergstrom[1,3], Keith D. Rivera [1],
Hannah B. Taylor [1], Susan Klaeger [1], Charles Xu[1], Eva K. Verzani[1],
C. Jackson White[1], Hilina B. Woldemichael[1], Maya Virshup[1], Meagan E. Olive [1],
Myranda Maynard[1], Stephanie A. Vartany[1], Joseph D. Allen [1], Kshiti Phulphagar[1],
M. Harry Kane [1], Suzanna Rachimi[1], D. R. Mani [1], Michael A. Gillette [1,2],
Shankha Satpathy [1], Karl R. Clauser[1], Namrata D. Udeshi [1] ✉ &
Steven A. Carr [1] ✉

Serial multi-omic analysis of proteome, phosphoproteome, and acetylome provides insights into changes in protein expression, cell signaling, cross-talk and epigenetic pathways involved in disease pathology and treatment. However, ubiquitylome and HLA peptidome data collection used to understand protein degradation and antigen presentation have not together been serialized, and instead require separate samples for parallel processing using distinct protocols. Here we present MONTE, a highly sensitive multi-omic native tissue enrichment workflow, that enables serial, deep-scale analysis of HLA-I and HLA-II immunopeptidome, ubiquitylome, proteome, phosphoproteome, and acetylome from the same tissue sample. We demonstrate that the depth of coverage and quantitative precision of each 'ome is not compromised by serialization, and the addition of HLA immunopeptidomics enables the identification of peptides derived from cancer/testis antigens and patient specific neoantigens. We evaluate the technical feasibility of the MONTE workflow using a small cohort of patient lung adenocarcinoma tumors.

The use of patient samples in biological research is critical for understanding the molecular pathways driving disease progression. These investigations routinely leverage deep-scale, multi-omic characterizations to broadly survey diverse biological pathways, such as cell signaling, protein degradation, and antigen presentation in hopes of discovering disease biomarkers or putative therapeutic targets. Because patient samples are generally available in limited amounts, decisions may have to be made as to which 'omic analyses are most desirable and feasible. Mass spectrometry–based proteomics is a proven technology for parallel analyses that involve the characterization of cell surface immunopeptidomes along with intracellular proteins and their post-translational modifications (PTMs)[1–10]. Technological advances to collect immunopeptidome, proteome, phosphoproteome, ubiquitylome, and acetylome datasets have often been made using parallel workflows. However, while parallel multi-omic analyses have yielded important biological insights, they are often

[1]Broad Institute of Massachusetts Institute of Technology and Harvard, Cambridge, MA 02142, USA. [2]Massachusetts General Hospital, Boston, MA 02114, USA. [3]These authors contributed equally: Jennifer G. Abelin, Erik J. Bergstrom. ✉e-mail: jabelin@broadinstitute.org; udeshi@broadinstitute.org; scarr@broad.mit.edu

reduced to the analysis of one or two post-translational modifications due to the requirement of larger amounts of sample (relative to proteome) necessary to achieve deep coverage. Therefore, workflows that facilitate a shift from parallel to serial multi-omic data collection are advantageous for large-scale discovery efforts in patient cohorts, as they result in comprehensive datasets that enable holistic insights into cellular pathways that may not have been possible with the sample input requirement of parallel multi-omic protocols[11,12].

One of the main ways to overcome the restricted availability of patient tissue has been to serialize sample processing such that the flow-through of one enrichment step is used as the input for the next enrichment step. Current serial enrichment workflows for measuring 'omes at high multiplex leverage isobaric reagents such as TMT[13–17] or iTRAQ[18,19], and have successfully serialized the collection of the proteome, phosphoproteome, and acetylome. However, immunopeptidome and ubiquitylome (i.e. anti-K-ε-GG antibody enrichment) methods have only been performed together in parallel to these other 'omes. This is because immunopeptidomics and ubiquitylomics protocols have specific requirements: immunopeptidome enrichment is done prior to sample digestion and enrichment of ubiquitylated peptides occurs prior to TMT labeling[3,10,20,21]. In settings where there are inadequate sample amounts for parallel processing protocols, the ability to concomitantly identify cancer driver signatures and detect changes in PTM-mediated signaling networks and HLA peptide processing and presentation is hindered.

There are additional complications that preclude adding immunopeptidome analysis into current serial processing strategies. For example, sample preparation for immunopeptidomics is distinct from conventional proteomics: immunopurification (IP) of HLA molecules requires the use of native lysis buffer containing mild detergent to maintain protein conformations and solubilize membrane-bound HLA proteins. In contrast, current serial proteome and PTM-ome enrichment protocols denature proteins using urea or SDS prior to tryptic digestion preventing upstream HLA peptide complex enrichment. Furthermore, more sample input has typically been used for immunopeptidomics than for proteomics and PTM-omics to enable detection of low-abundant, clinically relevant antigens such as neoantigens[2,6,22–24]. For immunopeptidomics workflows that attempt to directly identify neoantigens, a separate aliquot of tissue, usually 500–1000 mg of wet weight tissue or up to 1 billion cells[2,6,22,25,26], is needed compared to 25–50 mg for serial, multiplexed proteomics, phospho-, and acetyl-peptidomics[13–16,19]. Although multiple efforts are ongoing to decrease the input amounts for discovery immunopeptidome experiments with the goal of neoantigen identification[24,27–31] these reports do not leverage the HLA enriched flow-through for multiomic analyses, and if the proteome is performed, it is performed in parallel with a separate aliquot of cells[30,32]. Recently, we[33] and Nagler et al.[34] reported the first serial proteome from HLA-I enriched flowthrough, yet neither of these studies implemented downstream serial PTM-ome analyses.

To overcome the challenges of serializing deep-scale immunopeptidome and ubiquitylome workflows together with proteome, phosphoproteome, and acetylome profiling from a single tissue sample, we have developed an integrated proteomics workflow that we term MONTE (Multi-Omic Native Tissue Enrichment). MONTE extends recently published methods for isolation and analysis of immunoprecipitated HLA peptide complexes from clinical specimens[7,8,35,36] by use of recent improvements in MS instrumentation, off-line fractionation, and gas-phase separation using FAIMS[24] to increase immunopeptidome yield, enabling use of as little as 50 mg wet weight tissue. The flow-through of the HLA immunopeptidome purification contains the intact cellular proteome that is subjected to SDS-based lysis and tryptic digestion to make the post–HLA enrichment flow-through compatible with the current multiplexed, serialized multi-proteomics workflow[37]. The resulting protein digest is then processed and analyzed by

the UbiFast workflow for multiplexed ubiquitylation profiling using anti-K-ε-GG antibodies and on-antibody TMT labeling[3,38]. The peptide flow-throughs of the UbiFast enrichment step containing unlabeled, non-K-ε-GG peptides are further processed for deep-scale and highly multiplexed measurement of the proteome, phosphoproteome, and acetylome data collection.

Here, we systematically evaluate each step of the serial MONTE workflow and apply the optimized method in a proof-of-concept study of primary patient lung adenocarcinoma (LUAD) tumors. The results demonstrate that the depth of coverage and quantitative precision of each of the 'omes is not compromised by adding HLA peptidome and ubiquityl-peptide enrichments in serial with proteome, phosphoproteome, and acetylome analysis. HLA immunopeptidomics of these pilot samples identifies peptides derived from annotated cancer/testis antigens and patient specific neoantigens. We also provide a publicly available data viewer https://proteomics.broadapps.org/CPTAC-MONTE2022/ that enables researchers to visualize and explore this multi-omic dataset. Here we show that the MONTE workflow overcomes prior limitations of parallel processing workflows that have prevented concordant readout of the immunopeptidome, proteome, and PTM-omes from a single sample, thereby enabling new insights into cancer and other disease biology.

## Results
### Serialized immunopeptidome, proteome, and PTM-ome enrichment

To address the challenge of deeply characterizing clinically relevant samples with limited cellular input, we serialized HLA-I and HLA-II immunopeptidomics with ubiquitylome, proteome, phosphoproteome, and acetylome profiling workflows. The Multi-Omic Native Tissue Enrichment (MONTE) is represented in Fig. 1. Four changes were made to previously reported serial multi-omic enrichment protocols[3,4,15,16,18,19], each of which was evaluated to ensure that each proteomic data type was not significantly impacted. First, we incorporated UbiFast-based K-ε-GG peptide enrichment before serial, multiplexed proteome, phosphoproteome, and acetylome collection; previously, UbiFast had only been done in parallel with PTM-ome workflows[17]. Second, we optimized and added serial HLA-II and HLA-I immunopeptidome enrichment steps prior to the downstream multiomics analyses. Here we incorporated a broad set of protease inhibitors specific to each proteome and PTM-ome, used a pan anti–HLA-DR, -DP, and -DQ antibody mixture selected because it performed the best in a duplicate comparison study (Supplementary Data 1), and reversed the IP order relative to prior publications[39–41], opting to enrich HLA-II followed by HLA-I to prevent HLA-II peptide contamination in HLA-I data that is length filtered to canonical 8–11mers. Third, we replaced 8 M urea cell lysis with SDS denaturation and digestion on an S-Trap to facilitate removal of detergents present in the native lysis buffer used for HLA IP and confirmed the S-Trap method recovered the most unique proteins in a single shot proteome analysis (Supplementary Data 1). Fourth, to enable higher throughput and reproducibility, we incorporated an optimized version of a semi-automated, 96-well plate–based HLA immunopeptidomics workflow that enabled parallel desalting of serial HLA-II and HLA-I IP elutions[39]. We selected the semiautomated serial HLA enrichment[39] instead of previously reported fully automated enrichments[40,41] to enable the immunopurification to occur with end-over-end incubation at 4 °C because the stability of HLApeptide complexes is impacted by temperature[42]. We also implemented automated phosphopeptide enrichment[43] and UbiFast K-ε-GG peptide enrichment[38] workflows for downstream processing. Evaluation and optimization of each step of the MONTE workflow is detailed below.

### Ubiquitylomics in serial with multi-omic sample processing

We first sought to integrate the ubiquitylomics UbiFast workflow in serial with our well-established TMT-multiplexed proteome,

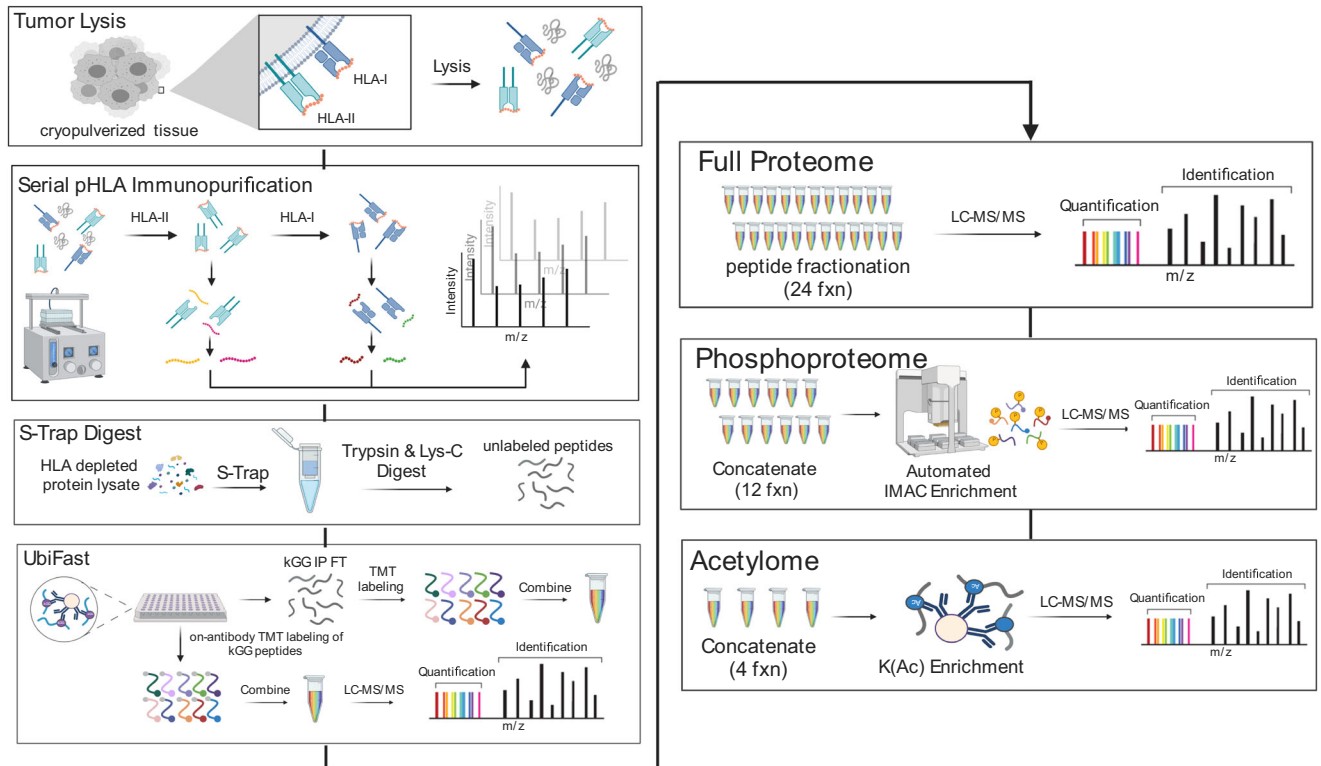

**Fig. 1 | Schematic Overview of the MONTE workflow.** The Multi-Omic Native Tissue Enrichments (MONTE) workflow for serial HLA immunopeptidome (label-free), ubiquitylome (label-free enrichment and on-antibody TMT labeling), phosphoproteome, and acetylome workflow, as UbiFast had only been carried out in parallel with these other 'omics workflows[4,18,19]. For this, proteome (TMT), phosphoproteome (TMT), and lysine acetylome (TMT). Black lines indicate the use of the flow-through from each subsequent step.

phosphoproteome, and acetylome workflow, as UbiFast had only been carried out in parallel with these other 'omics workflows[4,18,19]. For this, we created the workflow shown in Fig. 2A that starts with the UbiFast method for enrichment and on-antibody TMT labeling of K-ε-GG peptides[3,38]. After UbiFast processing, flow-throughs from the antibody enrichment step that contain unlabeled, non-K-ε-GG peptides are subsequently TMT labeled and used as input to generate proteome, phosphoproteome, and lysine acetylome datasets.

The addition of UbiFast was evaluated using tumors isolated from breast cancer patient–derived xenograft (PDX) models, representing Basal (WHIM2) and Luminal (WHIM16) subtypes[3,19,44]. WHIM2 and WHIM16 PDX models were selected because we previously showed that we obtain >14,000 distinct K-ε-GG-peptides from these samples starting with 0.5 mg peptide input per TMT channel[3,38]. Unlabeled peptide flow-throughs from K-ε-GG antibody captures corresponding to 0.25 mg input peptide per TMT channel were subsequently labeled with TMTPro and combined for serial proteome, phosphoproteome, and lysine acetylome analyses. LC-MS/MS showed expected coverage of the proteome, phosphoproteome, and acetylome with 9,402 human proteins, 28,523 human phosphorylation sites and 6,294 human lysine acetylation sites identified and quantified from the UbiFast flow-through samples (Fig. 2B, Supplementary Data 2). The overlap of proteins, phosphorylation, and acetylation sites between experiments with and without serial UbiFast processing was high (87.3% proteome, 63.0% phosphoproteome, 54.9% acetylome) (Fig. 2C). Pearson correlations of TMT ratios between intraplex replicates for UbiFast flow-throughs were high with median correlations of 0.96 for both Basal and Luminal subtypes in the proteome, 0.90 and 0.91 for Basal and Luminal subtypes in the phosphoproteome, and 0.84 and 0.83 for Basal and Luminal subtypes in the acetylome, indicating that UbiFast preprocessing does not negatively affect reproducibility in any of the 'omes (Fig. 2B, D). Basal vs. Luminal protein, phosphosite, and

acetylation site TMT ratios measured in UbiFast flow-through samples and non-UbiFast samples correlated well ($R^2 = 0.89$ proteome, $R^2 = 0.73$ phosphoproteome, $R^2 = 0.73$ acetylome). Median correlation values were similar for samples processed with and without UbiFast, however we note a minor increase in correlation spread for phosphoproteome data acquired from UbiFast flow-through samples (Fig. 2D). The number of regulated proteins, phosphorylation sites and acetylation sites in Basal vs. Luminal samples was very similar with and without UbiFast. In samples with no UbiFast, 8,462 (92%) of proteins, 24,373 (87%) of phosphorylation sites and 4867 (81%) of acetylation sites were significantly different in Basal vs Luminal samples using a moderated two-sample $t$-test (adj. $p$val = 0.05). For samples where UbiFast was implemented, 8,601 (91%) of proteins, 24,366 (85%) of phosphorylation sites, and 5012 (80%) of acetylation sites were significantly different in Basal vs Luminal samples (adj. $p$val = 0.05). We found that UbiFast does not affect the coverage of phosphoserine, phosphothreonine or phosphotyrosine peptides. For both +/- UbiFast samples, 87% of phosphopeptides harbored at least one phosphoserine, 16% harbored at least one phosphothreonine and 1% at least one phosphotyrosine. These results demonstrate that the incorporation of UbiFast does not affect the depth or distribution of phosphosites detected.

Unsupervised hierarchical clustering of proteome, phosphoproteome, and acetylome samples shows the expected separation of samples by breast cancer subtype with much smaller separation by experiment (Fig. 2E). In addition, Gene Set Enrichment Analysis (GSEA) of proteins, phosphorylation sites, and acetylation sites shows that gene sets associated with basal and luminal breast cancer subtypes are appropriately regulated in data acquired with and without initial enrichment of ubiquitylated peptides using UbiFast (Fig. 2F, Supplementary Data 2)[45]. Site-centric PTM Signature Enrichment Analysis (PTM-SEA)[46] was also performed on regulated phosphorylation sites

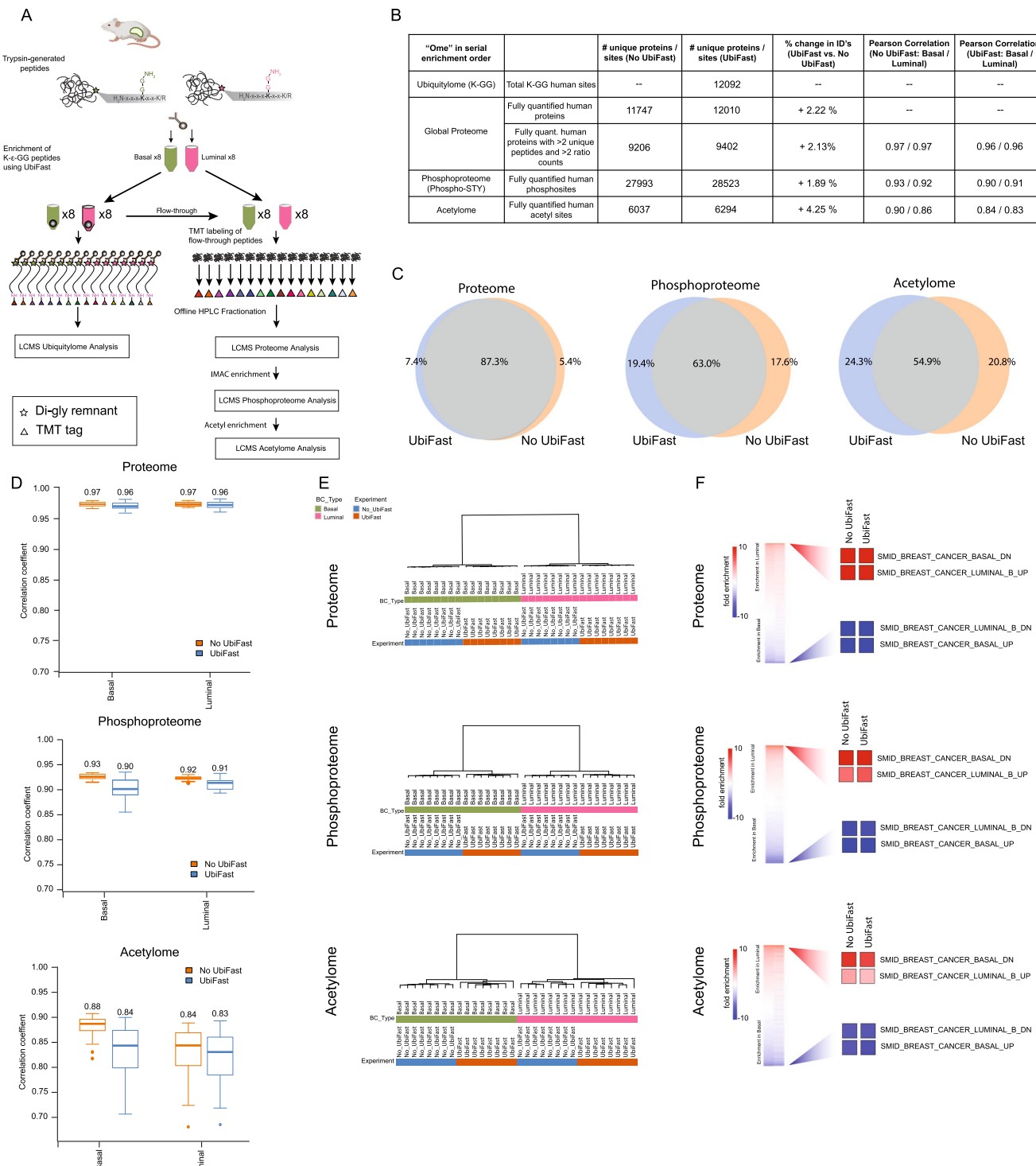

**Fig. 2 | Comparison of the proteome and phosphoproteome of PDX samples with and without UbiFast for pre-enrichment of ubiquitylated peptides.**
**A** Experimental design used to analyze K-ε-GG peptides, proteins, and phospho-peptides from the same luminal and basal PDX breast cancer samples. The results for PDX samples processed directly for proteome and phosphoproteome were compared with samples initially enriched for ubiquitylated peptides using UbiFast and the flow-through of the UbiFast enrichment then used for proteome and phosphoproteome; 250 μg peptide input was used per TMT state. **B** Summary table reporting the numbers of proteins and phosphorylation sites detected and regulated from non–UbiFast-enriched (No UbiFast-orange) and UbiFast-enriched (Ubi-Fast-blue) PDX samples. Regulated sites determined using a moderated two-sample *t*-test (adj. *p*val = 0.05). Pearson correlation values are also reported. **C** Venn

diagrams showing the overlap of proteins and phosphorylation sites for non–UbiFast-enriched and UbiFast-enriched samples. **D** Pearson correlation between Luminal and Basal PDX replicates (*n* = 8 luminal, *n* = 8 basal process replicates) for proteome, phosphoproteome and acetylome data. Boxplots depict upper and lower quartiles, with the medians shown as a solid line. Whiskers show 1.5 interquartile range. **E** Unsupervised hierarchical clustering of proteome, phos-phoproteome, and acetylome samples shows the expected separation of samples by breast cancer subtype. **F** Gene set enrichment of proteins, phosphorylation sites, and acetylation sites show that gene sets associated with basal and luminal breast cancer subtypes are appropriately regulated in data acquired with and without initial enrichment of ubiquitylated peptides using UbiFast (also see Supplementary Data 2). Source data for **B**, **D**, and **F** are provided as a Source Data file.

and the top gene sets show the same trends (Supplementary Data 2). Taken together, these results support the feasibility of incorporating multiplexed ubiquitylation profiling using UbiFast up front and serially with multiplexed proteome and PTM profiling workflows.

To further evaluate the effect of UbiFast on serially enriched phosphoproteomes, we used a single TMT experiment to compare phosphopeptide enrichment by IMAC with and without prior UbiFast processing. For this experiment, all phosphopeptide enrichment samples were derived from either Basal or Luminal PDX models and measured in the same TMTpro16 plex (Basal: $n = 4$ with UbiFast and $n = 4$ no UbiFast; Luminal: $n = 4$ with UbiFast and $n = 4$ without UbiFast) to quantify potential differences on exactly the same phosphopeptides between +/− UbiFast processed samples (Supplementary Fig. 1A). We identified 45,051 human phosphorylation sites of which only 3.2% showed significantly reduced intensity (>2-fold change and adj. $p$val < 0.05) in samples where UbiFast was incorporated in the workflow (Supplementary Fig. 1B, Supplementary Data 3). The majority of the depleted phosphopeptides are short (Supplementary Fig. 1C) and hydrophilic and are likely depleted by the extra desalting step following UbiFast processing and prior to TMT labeling and phosphopeptide enrichment. UbiFast does not significantly affect replicate correlation (Supplementary Fig. 1B) or unsupervised hierarchical clustering of phosphoproteome data by breast cancer subtype (Supplementary Fig. 1D).

### Evaluating HLA-II and HLA-I enrichment in serial workflow

We evaluated the impact of adding serial HLA-II and HLA-I enrichment prior to serial multi-omic enrichment workflow using ten cryopreserved primary LUAD tumors from the CPTAC cohort[14] (Supplementary Data 4). Human tumor samples are more relevant than tumor samples derived from immunocompromised mice, and in our initial testing, we found that the yield of HLA-I and HLA-II immunopeptidomes from the PDX breast cancer tumor models was too low to derive meaningful conclusions (Supplementary Data 5). LUAD tumors were selected because lung tissue is known to have HLA-I and HLA-II expression[47,48] and one LUAD primary tumor has been profiled successfully using serial HLA-II and HLA-I immunopeptidomics[36]. This set of LUAD samples was chosen to represent important biological differences of high relevance to lung adenocarcinoma, as five samples were driven by KRAS mutations and five by *EGFR* mutations. Each driver mutation subset included samples from both men and women and both Asian and Western/Caucasian ethnicity were represented; none of these previously characterized tumors were from the immune hot cluster[14]. The human LUAD tumors (50-86 mg cryopulverized tissue) were processed with and without initial serial HLA enrichment (Fig. 3A). In both cases, S-Trap–based protein digestion[37] was used instead of 8 M urea digestion following HLA enrichment because we have previously shown that serial HLA immunopeptidome and downstream whole-proteome analysis required the removal of detergents present in the native lysis buffer used for HLA enrichment[33].

Label-free, antibody-based, serial HLA enrichment identified a median of 11,387 HLA-I (8278–13,727) and 5,263 HLA-II (1123−9726) bound peptides from each of these ten LUAD tumors (Fig. 3B). Our depth of >10,000 HLA-I peptides from as little as 50 mg cryopulverized tumor corresponding to ~2 mg protein lysate was encouraging and clearly indicated that the method would likely be usable with even smaller amounts of input tumor material. We confirmed that the observed HLA-I and HLA-II peptides had the expected length distributions (Fig. 3C, D) and HLA-I binding characteristics (Fig. 3E, F) using a motif analysis and the HLA-I presentation predictor HLAthena[7,8]. Patient C3N-01416 had a larger representation of 8mers in the HLA-I immunopeptidome, which was expected because of the known preference for 8mers presented by HLA-B*18 alleles. We also confirmed that HLA-II immunopeptidomes contained motifs

consistent with patient HLA-II alleles called from RNA-Seq data by arcasHLA[49] (Supplementary Fig. 2).

The protein flow-throughs from HLA immunopeptidome enrichments were next digested with Lys-C and trypsin using S-Traps in parallel with half of each LUAD tumor that was not HLA enriched. A summary of the resulting depths of these head-to-head proteomes, ubiquitylomes, phosphoproteomes, and acetylomes is shown in Fig. 4A (Supplementary Data 7). The proteome and ubiquitylome results demonstrate that similar numbers of canonical human proteins (11,028 vs.10,729) and K-ε-GG peptides (9516 vs. 9419) were identified and fully quantified between the non−HLA-enriched ("No HLA") and HLA-enriched ("HLA FT") samples, respectively. A 16% decrease in the total number of phospho-sites (-8% phosphorylated proteins) was observed when using the HLA-enriched samples (No HLA: 26,627 phosphosites, 6745 phosphoproteins; HLA FT: 22,339 phosphosites, 6235 phosphoproteins), suggesting that the phosphatase inhibitors added to our lysis buffer may be losing their activity during the protein-level, HLA immunopeptidome enrichment. The number of lysine residues observed to be acetylated on internal lysine residues (i.e., not at the N- or C-terminus of the peptide) increased by 45% in the HLA-enriched samples (No HLA: 3702; HLA FT: 5380 internal K-acetylsites). The relative yield of acetylated peptides (i.e., the percentage of K-Ac peptides relative to the total peptides identified in the sample) in the HLA-processed samples was significantly higher (75% vs. 55%). Given that the protein lysates were incubated at 4 °C for 6 h during HLA enrichment, we sought to rule out possible non-enzymatic acetylation[50]. Acetylome analysis of A375 melanoma cells with and without the 6 h HLA IP incubation conditions yielded a similar number of acetylated peptides when compared to no HLA incubation conditions (Supplementary Data 6), suggesting the addition of the HLA IP did not cause non-enzymatic acetylation. We speculate that the increased yield of acetylation sites could be due to pre-clearing of non-specifically binding components in the complex tissue lysates by HLA- and K-ε-GG antibodies.

### The MONTE workflow recapitulates expected biological signals

To assess potential differences between HLA-enriched and non−HLA-enriched samples, we analyzed the ten LUAD tumor proteomes, ubiquitylomes, phosphoproteomes, and acetylomes using a principal component analysis (PCA) (Fig. 4B, Supplementary Data 7). PCA shows that samples cluster by LUAD tumor, not by the processing method used, demonstrating that biological differences among the samples are stronger than technical variation between these serial workflows. The acetylomes of HLA-enriched and non−HLA-enriched samples were somewhat less well correlated. The total number of proteins identified and quantified from HLA-enriched and non−HLA-enriched samples were shown to have a 93% overlap (Fig. 4C, D). Slightly fewer proteins (3%) were identified from the HLA enrichment flow-throughs. We looked into the proteome data for depletions in HLA-I and HLA-II chaperone proteins to confirm our serial HLA-II and HLA-I immuno-purification is not depleting known HLA protein binding partners. We did not observe proteome depletion of HLA-I chaperones CALR, CANX, or TAPBR or HLA-II chaperones HLA-DM and HLA-DO. The highly polymorphic nature of HLA molecules makes these proteins difficult to quantify by proteomics as digestion with trypsin does not always produce unique, LC-MS/MS detectable peptides suitable for differentiating one HLA allele from another in the sample. It is also plausible that HLA protein is present after the w6/32 enrichment, as this antibody is sensitive to the amino terminus of human beta2-microglobulin[51], and not all HLA proteins are in mature HLA-peptide complexes. Hence, HLA proteins were not used in this evaluation. Regardless, the observation that known HLA-I and HLA-II chaperones are not depleted suggests the addition of the serial HLA immunopurification does not have a negative impact on the downstream proteome analysis.

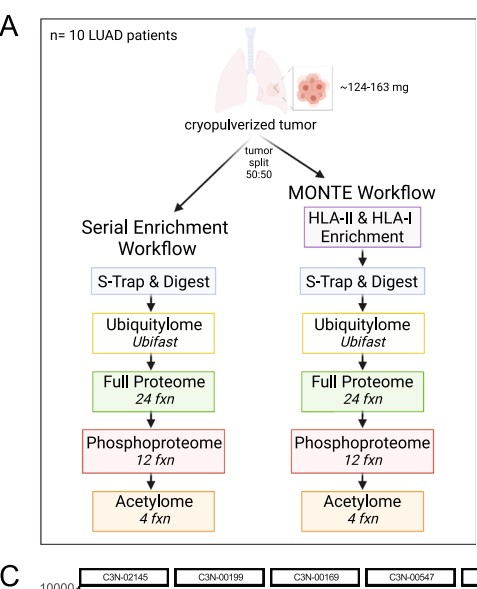

| Participant ID | Tumor Purity | Sample Input (mg) | HLA-I peptides | HLA-II peptides |
|---|---|---|---|---|
| C3N-02145 | 62% | 74.4 | 9397 | 2368 |
| C3N-00199 | 75% | 50 | 8808 | 5918 |
| C3N-00169 | 68% | 69.2 | 9248 | 7133 |
| C3N-00547 | 75% | 79.2 | 12954 | 2600 |
| C3N-00579 | 62% | 85.8 | 13727 | 9436 |
| C3N-01016 | 70% | 84.9 | 13609 | 8395 |
| C3L-01632 | 72% | 67.4 | 8278 | 4609 |
| C3N-01024 | 75% | 90.7 | 13064 | 1177 |
| C3N-01416 | 48% | 83.2 | 12183 | 9726 |
| C3L-02549 | 70% | 66.6 | 10590 | 1123 |

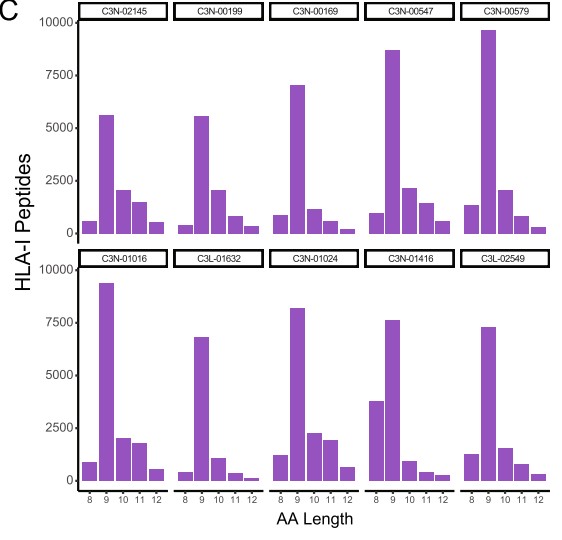

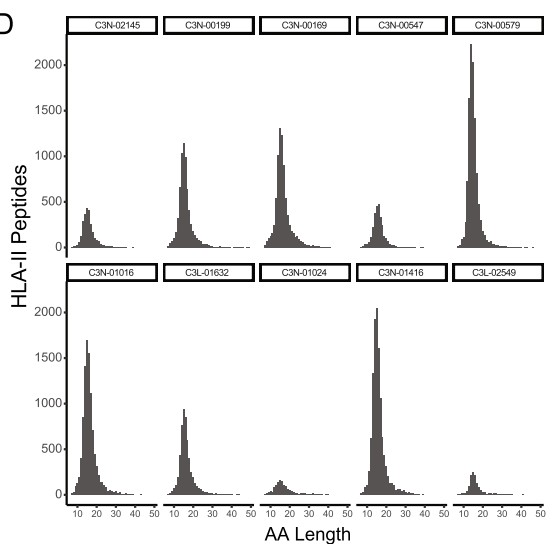

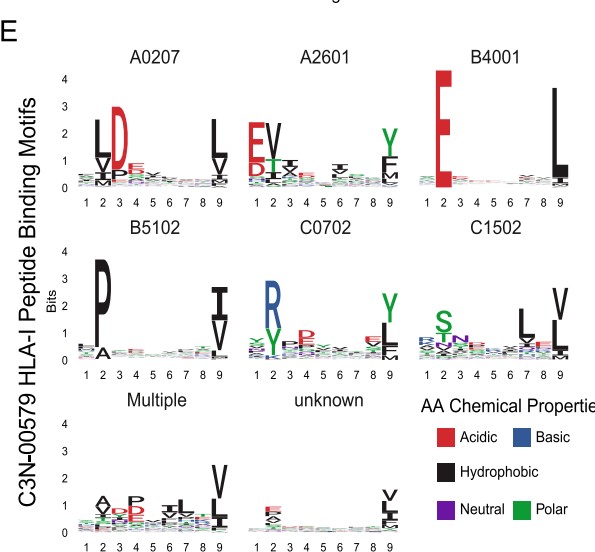

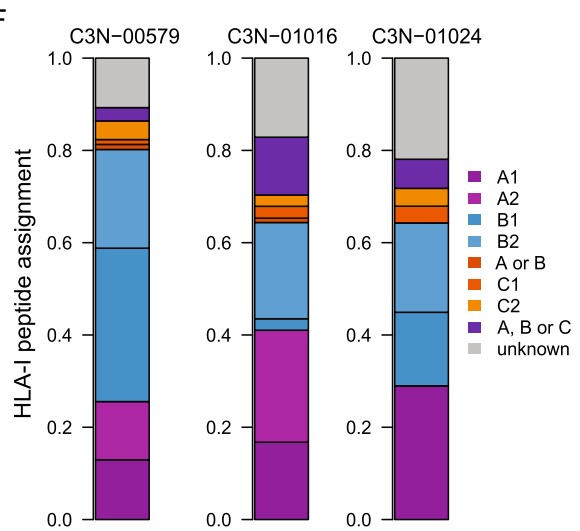

**Fig. 3 | LUAD tumor HLA-I and HLA-II immunopeptidome profiling using MONTE. A** Schematic overview of the head-to-head serial multi-omic enrichment vs. MONTE workflows used to characterize ten cryopulverized primary LUAD tumors. **B** Table summarizing the tumor input and resulting HLA-I and HLA-II peptides mapping to human source proteins detected from ten LUAD patients. **C** Length distributions of HLA-I peptides from the LUAD cohort. **D** Length distributions of HLA-II peptides from the LUAD cohort. **E** Example HLA-I peptide binding motifs and the alleles expressed by LUAD Patient C3N-00579. **F** Example HLA-I peptide-allele assignments to individual HLA-A, -B, -C alleles or combinations of these alleles obtained using the presentation predictions from HLAthena for three LUAD patients[7,8]. Source data for **C**, **D**, and **F** are provided as a Source Data file.

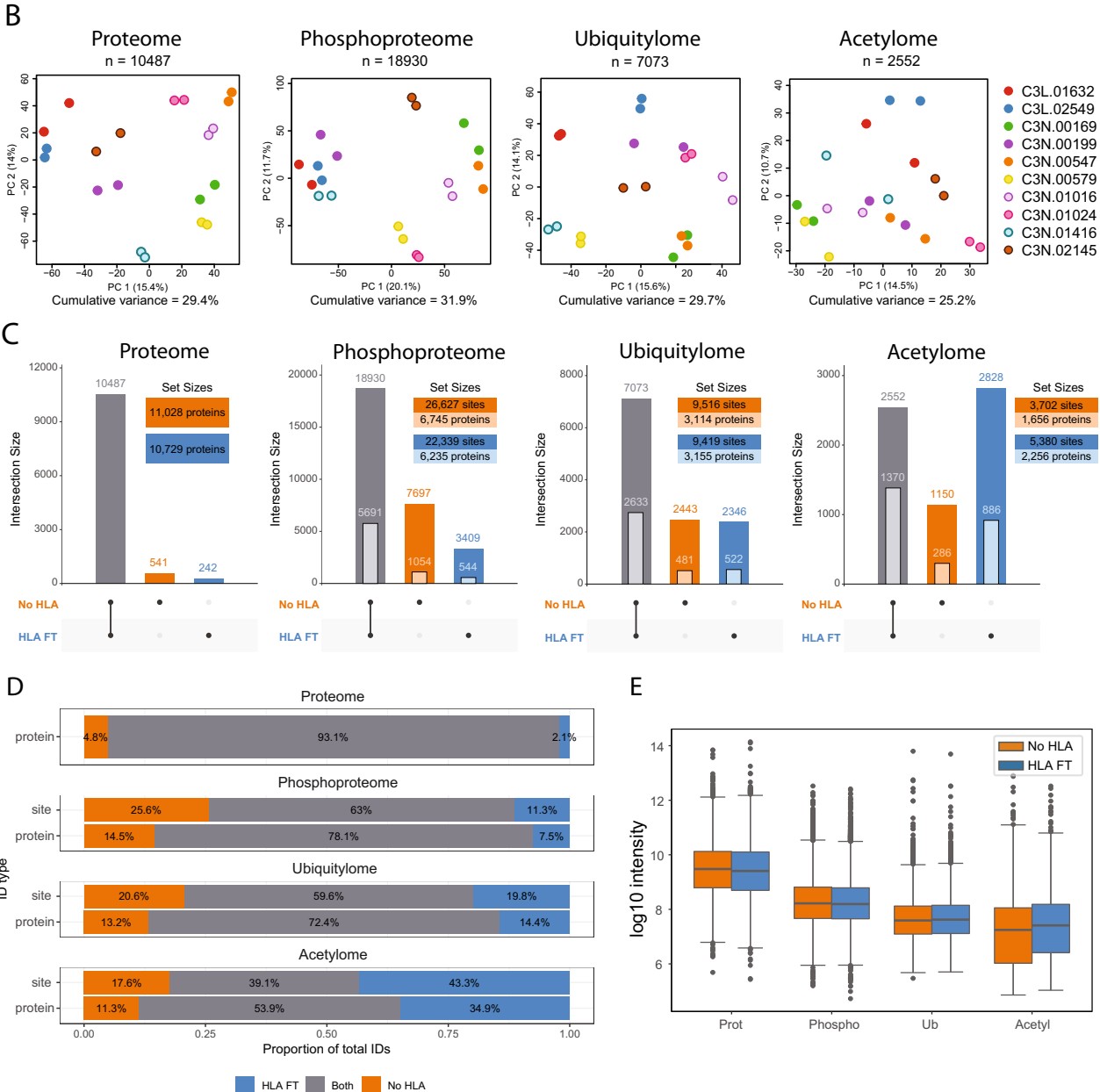

| "Ome" in serial enrichment order | | # unique proteins/sites, Serial Enrichment (no HLA IP) | # unique proteins/sites, MONTE (HLA IP FT) | % change in IDs (no IP vs HLA IP FT) |
|---|---|---|---|---|
| Ubiquitylome (K-GG) | Total human KGG sites | 11866 | | -- |
| | Fully quantified human KGG sites | 9516 | 9419 | -1.0% |
| | Fully quant. and localized KGG sites | 9485 | 9389 | -1.0% |
| Global proteome | Total human proteins | 11301 | | -- |
| | Fully quantified human proteins | 11028 | 10729 | -2.7% |
| | Fully quant. human proteins w/ > 1 unique peptide | 10056 | 9692 | -3.6% |
| Phosphoproteome (phospho-STY) | Total human phospho-sites | 30109 | | -- |
| | Fully quantified human phospho-sites | 26627 | 22339 | -16.1% |
| | Fully quant. and localized phospho-sites | 17793 | 14819 | -16.7% |
| Acetylproteome (acetyllysine, AcK) | Total human AcK sites | 6671 | | -- |
| | Fully quantified human AcK sites | 3702 | 5380 | +45.3% |
| | Fully quant. and localized AcK sites | 3624 | 5168 | +42.6% |

The overlap between HLA-enriched and non–HLA-enriched protein lysates was 60% for ubiquitylation sites, 72% for ubiquitylated proteins, 63% for phosphorylation sites, and 78% for phosphorylated proteins, which is an expected result using multiplexed, data-dependent LC-MS/MS methods for highly similar processing workflows[19] (Fig. 4C, D). A 16% loss of total phosphosites was observed in the HLA enriched lysates, which we attribute to the combination of the losses from the extra desalting step in UbiFast and the possible decrease of phosphatase inhibitor activity over the 6 h serial HLA enrichment. To improve this in future studies, we plan to implement a second addition of phosphatase inhibitors between the HLA-II and HLA-I enrichments. The lowest overlap across experiments was

**Fig. 4 | Evaluation of data depth between serial multi-omic enrichment with and without HLA enrichment. A** Summary table reporting the numbers of proteins and PTM-containing peptides detected from non–HLA-enriched (No HLA IP) and HLA-enriched (HLA IP FT) LUAD tumors; FT = flowthrough (*n* = 10 LUAD patient tumors). **B** Principal component analysis of LUAD tumors (*n* = 10) for all human proteins and PTM sites quantified in all TMT channels in both No HLA IP (orange) and HLA IP FT (blue) conditions. **C** The total number of quantified human proteins and PTM site identifications illustrated as 'UpSet' plots[93]. Vertical bars depict the number of uniquely or jointly detected features as indicated by the layout matrix

below. Darker colored bars represent unique proteins in the proteome and unique PTM sites in each PTM-ome. Lighter colored nested bars represent the number of unique modified proteins in each PTM-ome. **D** Stacked bar chart showing proportional overlap for No HLA IP and HLA IP FT conditions by unique proteins in the proteome and by PTM sites or modified proteins for each PTM-ome. **E** Log10 total intensity distributions of all human proteins and PTM sites from No HLA IP and HLA IP FT samples (*n* = 10 LUAD patient tumors). Boxplot depicts upper and lower quartiles with the median shown as a solid line. Whiskers show 1.5 × interquartile range. Source data for **C**, **D**, and **E** are provided as a Source Data file.

observed for acetylome data because 45% more acetylated peptides were observed in the HLA enriched samples. Overall, the HLA-enriched samples capture a similar depth of coverage observed in non–HLA-enriched samples, and adding HLA enrichment up front in a serial workflow does not introduce considerable bias in downstream proteome, ubiquitylome, phosphoproteome, and acetylome data collection.

We next investigated known oncogenic and tumor suppressor proteins. Oncogenes EGFR and KRAS and tumor suppressor genes RB1 and STK11 were detected across multiple 'omes, with similar patterns of protein and PTM site levels observed in both HLA-enriched and the non–HLA-enriched samples (Supplementary Figure 3; also available using the data viewer: https://proteomics.broadapps.org/CPTAC-MONTE2022/). For example, patient C3N-00199 showed the highest level of total EGFR ubiquitinylation across 10 of the 11 sites identified in both MONTE experiments, and patient C3N-00547 had the highest level of total RB1 phosphorylation. The high level of EGFR and RB1 protein expression is likely driving these high levels of total PTMs in these patients. The tumor suppressor protein TP53 had variable detection (7/10) in the "Discovery" dataset[14], which may have led to the lack of detection in both the HLA-enriched and non–HLA-enriched TMT plexes (Supplementary Fig. 4; also available using the data viewer: https://proteomics.broadapps.org/CPTAC-MONTE2022/). We also observed HLA-I peptides from wild-type EGFR (8/10), KRAS (3/10), RB1 (10/10), TP53 (9/10), and STK11 (1/10) across the LUAD patient cohort (Supplementary Data 8). Conversely, HLA-II peptides within expected nested sets (Supplementary Data 8) were only detected from EGFR (7/10), which is endocytosed upon activation allowing it entry into the HLA-II processing and presentation pathway. No clear trends between HLA-I and HLA-II peptide presentation and driver mutation status were detectable in this set of oncogenes and tumor suppressors. Nevertheless, the detection of these oncogenic and tumor suppressor proteins across multiple 'omes from samples that underwent HLA enrichment demonstrates that known biological signals can be recovered using the MONTE workflow.

**Mutated, noncanonical, and CT antigen-derived HLA peptides**

MONTE immunopeptidomes were analyzed using a personalized database containing canonical human proteins, noncanonical proteins from novel or unannotated open reading frames (nuORFs)[52], and patient-specific mutations (Fig. 5A, Supplementary Data 9). Initially, we looked in the LUAD immunopeptidomes for peptides derived from cancer/testis antigen (CTA) source proteins reported in the CTA database and observed peptides from 45 unique source proteins[53,54]. Across the set of LUAD tumors, peptides derived from seven CTA source proteins previously reported in lung cancer[54] were detected, including two from the MAGE family (Fig. 5B). Most peptides from CTA source proteins were presented by HLA-I except for TEXT101 and ACTL8, which were presented by HLA-II. Surprisingly, two unique HLA peptides derived from the bromodomain testis-specific protein (BRDT) were presented by 6/10 tumors in our LUAD set (Supplementary Data 9). To confirm BRDT protein expression, we leveraged our proteome data and the transcriptome data published by Gillette et al.[14]. BRDT was detected in the transcriptome and proteome data,

suggesting that this protein is expressed, making it a candidate for future immunogenicity investigations.

Next, we sought to detect peptides in our LUAD HLA immunopeptidomes derived from nuORFs whose translation has been supported by ribosome profiling using a recently published nuORF database[52] (Fig. 5C). High-confidence HLA-I and HLA-II peptide identifications derived from nuORFs were found across 9/10 patients. Because nuORFs represent rare observations within a large dataset, after false discovery rate (FDR) thresholding on the aggregate data set, we applied more stringent subset-specific FDR thresholding (see Methods). A majority of nuORF peptides also had predicted retention times that correlated well with their observed retention times, further increasing the confidence of detection (Supplementary Fig. 5). HLA-I immunopeptidomes contained far more unique nuORF source proteins than HLA-II, and the overall ranking of patients by number of unique nuORF source proteins did not correlate between HLA-I and HLA-II immunopeptidomes. The average representation of nuORF source protein categories per sample also differed between HLA-I and HLA-II, as a higher proportion of HLA-II nuORFs mapped to pseudogenes (19%) and few mapped to out-of-frame ORFs (5%), while the reverse was true for HLA-I, where the total percentage of pseudogenes and out-of-frame ORFs were 3% and 21%, respectively. These observations align with recent studies suggesting that the HLA-I pathway is more likely to sample less stable, shorter proteins, while the HLA-II pathway is more likely to sample stable source proteins[36,55]. The contrasting nuORF representations also highlight the differences in non-canonical source protein presentation between HLA-I and HLA-II pathways that are not yet fully understood but could be improved upon from data obtained on larger patient cohorts across diverse tissue types, as each tissue type may have unique nuORF expression characteristics.

We then assessed if the LUAD immunopeptidome depth enables the detection of HLA peptides containing patient-specific mutations (neoantigens). Historically, detection of neoantigens by LC-MS/MS has required enrichment from either billions of cells or gram levels of tissue, as neoantigens can represent only 0.01% of all unique peptide identifications in data dependent discovery experiments[2,22,56]. To find HLA-presented neoantigens, we analyzed the immunopeptidomes for peptides containing somatic mutations[57–59]. Two of the ten patients (20%) had at least one detected neoantigen in their HLA-I immunopeptidomes, of which four contained point mutations and one a frameshift deletion mutation (Fig. 5D, Supplementary Fig. 6). Neoantigen peptide identifications were supported using both retention time prediction and experimental comparisons of the mass spectra with synthetic peptides (Supplementary Figs. 5, 6). Most neoantigens were derived from mutations not shared across patient populations with the notable exception of the KRAS G12V neoantigen detected in patient C3N-00547. The KRAS G12V 10mer is a shared neoantigen that has been previously confirmed to be presented on HLA-A11[60,61]. We also detected two neoantigens bound to the less abundantly expressed HLA-C alleles, perhaps aided by the very similar binding specificity of the patient's two alleles C*08:01, C*12:03. In general, we observed that patients with high mutation burden and immunopeptidome depth (>10,000 peptides) were most

A

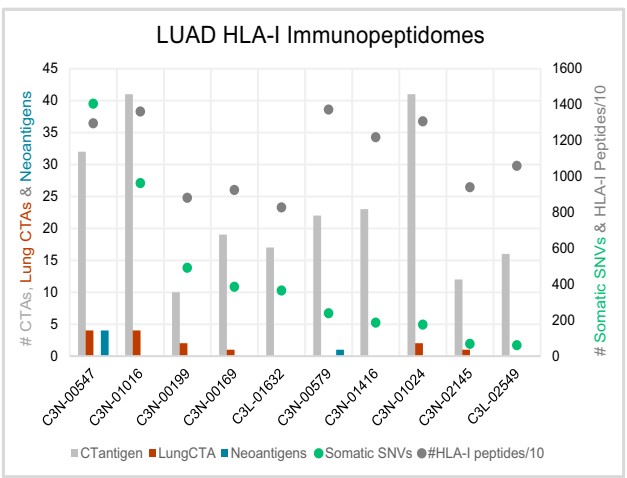

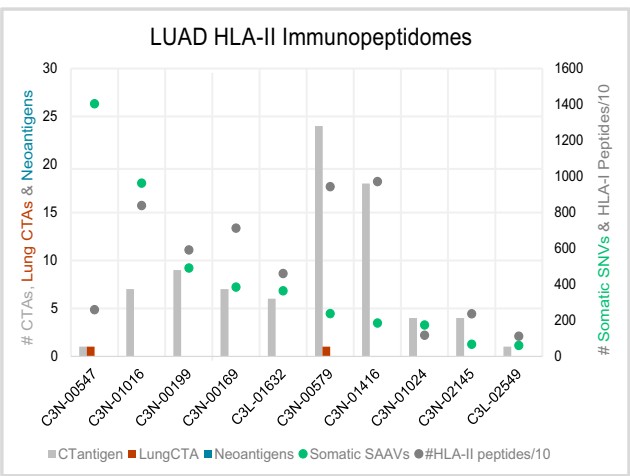

B

| CTAntigen Source Protein | # LUAD Patients >1 HLA peptide |
|---|---|
| BRDT | 6 |
| CABYR | 1 |
| TEX101 | 1 |
| ACTL8 | 1 |
| MAGEA3 | 1 |
| MAGEA6 | 1 |
| DDX53 | 1 |

C

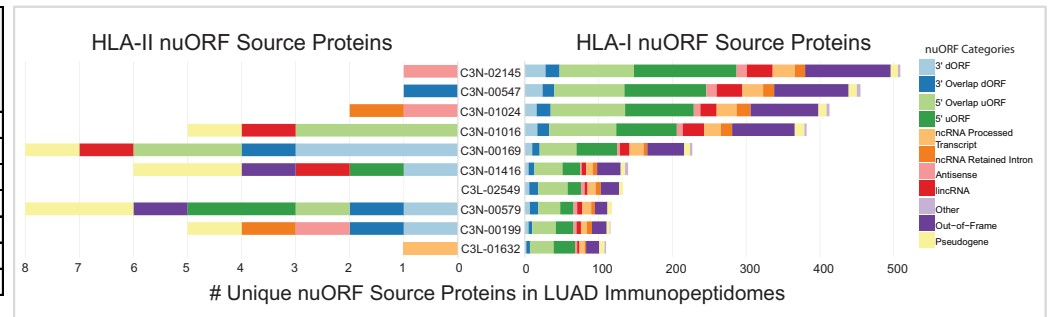

D

| LUAD Patient ID | Neoantigen Peptide | Source Protein Mutation | Predicted HLA-I Allele | HLAthena %Rank | Mutation detected in tryptic proteome |
|---|---|---|---|---|---|
| C3N-00547 | DQRLALVM | CIC R2377L | B*15:02 | 0.26 | not detected |
| C3N-00547 | ISNDLYLTL | DOCK3 R421S | C*08:01 | 0.07 | not detected |
| C3N-00547 | SAAADILLL | HUWE1 Q2591L | C*08:01 | 0.07 | not detected |
| C3N-00547 | VVVGAVGVGK | KRAS G12V | A*11:01 | 0.08 | not detected |
| C3N-00579 | SPHPGPATTI | RBM14 del frameshift | B*51:01 | 0.22 | not detected |

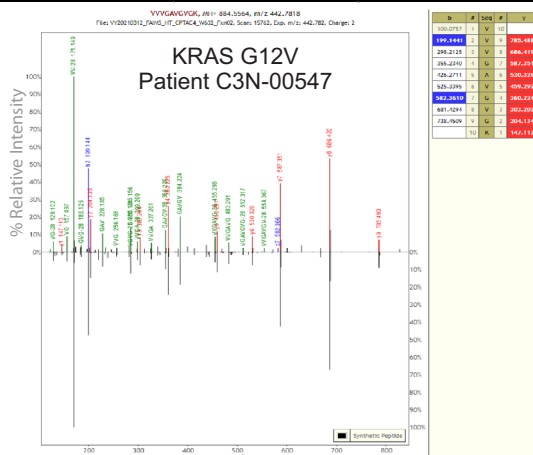

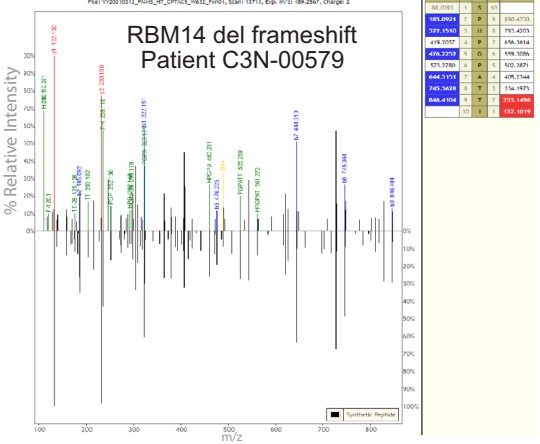

likely to have LC-MS/MS detectable neoantigens when using data dependent acquisition. These results suggest that detection of neoantigens by immunopeptidomics should, at present, be focused on tumor types with relatively high mutational burden, high HLA expression levels, and only the most highly optimized LC-MS/MS methods should be used.

**Evaluating HLA-peptide source protein presentation by MONTE**

An advantage of the MONTE workflow is that the resulting multi-omic data is derived from each single sample, enabling robust data integration. Thus, we evaluated whether integration of MONTE data would reveal insights into antigen processing and presentation. We first looked at how well HLA-I and HLA-II source proteins overlapped with

**Fig. 5 | Analysis of HLA peptides presented by LUAD tumors derived from CT antigen, nuORF, and mutation-containing source proteins. A** Summary of unique HLA-I and HLA-II peptides (dark gray), total somatic single nucleotide variants (SNVs; green), peptides mapping to source proteins from the CT database (light gray), peptides mapping to lung cancer–specific[54] CTA source proteins (red), and neoantigens containing patient-specific somatic mutations (blue) from ten LUAD tumors. The primary y-axis (left) shows counts of CTAs, lung CTAs, and neoantigens. The secondary y-axis (right) shows the total number of somatic SNVs and the immunopeptidome depth per LUAD patient. **B** Summary table reporting the number of LUAD patients presenting at least one HLA-I or HLA-II peptide from CTA source proteins reported by Djureinovid et al.[54]. **C** Mirrored stacked bar plots showing the number of unique nuORF source proteins that are presented by HLA-I (right) and HLA-II (left) colored by the nuORF category[52] across ten LUAD tumors. **D** Summary table of HLA-I neoantigens detected in primary LUAD patient tumors (top) accompanied by annotated MS/MS spectra for KRAS G12V (bottom, left) and the RBM14 frameshift (bottom, right) neoantigens. See Supplementary Fig. 6 for annotated MS/MS spectra of the others and synthetic peptide spectra. Leucine (L) and Isoleucine (I) cannot be distinguished by the MS instrumentation employed. Source data for **A** and **C** are provided as a Source Data file.

both the proteins detected in the proteome and ubiquitylome data (Fig. 6A). 78% of proteins identified in the proteome were also identified as HLA-I source proteins. In contrast, 30% of HLA-I source proteins were not observed in the proteome. For HLA-II, a 33% overlap between the proteins in the proteome and HLA-II source proteins was observed with 21% of HLA-II source proteins not detected in the proteome. The lower overlap of HLA-II source proteins and proteome is likely due to differences in biological sampling of source proteins by the HLA-I and HLA-II pathways. The HLA-II pathway primarily samples proteins that are degraded in the endosomal/lysosomal and autophagy pathways while the HLA-I pathway primarily samples proteins that are degraded by the proteasome. A higher proportion of ubiquitylated proteins, 89%, were detected as HLA-I source proteins, compared to 49% as HLA-II source proteins. This was expected because ubiquitylated proteins are a key source of proteasome-processed peptides that are HLA-I peptide precursors. Conversely, we noted only 26% of HLA-I source proteins were identified as ubiquitylated, suggesting that additional ubiquitylome datasets are required to capture all possible ubiquitylated proteins that enter the HLA-I processing and presentation pathway. Because HLA-I source protein expression levels and their ability to be processed by the proteasomal pathway are important factors for presentability[7], both proteome and ubiquitylome datasets are likely useful for incorporation into HLA-I prediction algorithms.

To better understand the variable levels of HLA-I and HLA-II peptides recovered in the LUAD immunopeptidomes, we looked at the trends of B2M and CD74 expression and PTMs across the 'omes with patients sorted from low to high ESTIMATE immune scores[62](Supplementary Fig. 7; also available using the data viewer: https://proteomics.broadapps.org/CPTAC-MONTE2022/). As expected, patients C3L-01632 and C3N-00199 with low mRNA and protein levels of B2M, a subunit of HLA-I complexes, had the lowest ESTIMATE immune scores and overall low HLA-I immunopeptidome depth. We also observed HLA-I peptides derived from B2M in 9/10 samples excluding patient C3L-01632. Next, we investigated CD74, a protein essential for HLA-II assembly and stabilization and the source of the CLIP peptides. We observed that the patients with the lowest HLA-II immunopeptidome depth, C3N-01024 and C3L-02549, did not have the lowest CD74 expression, and that the protein and RNA expression levels do not always correlate. Instead, these two patients had the most unique ubiquitination sites on CD74, suggesting that CD74 may be degraded at a higher rate in these patients. Understanding both the expression levels and PTM status of proteins involved in antigen presentation, such as B2M and CD74, may not directly correlate with HLA immunopeptidome depth, yet such analyzes do provide insights into the HLA presentation machinery in tumors.

We also investigated the representation of nuORFs in the MONTE proteomes and PTM-omes compared to those detected in the immunopeptidomes (Fig. 6B) and observed that the representation of nuORF categories[52] in HLA-I immunopeptidomes and phosphoproteomes were the most diverse compared to all other 'omes. In general, a higher proportion of pseudogenes were detected in the proteome and PTM-omes when compared to HLA-I and HLA-II immunopeptidomes. We also noted that the representation of nuORF categories varied across the different 'omes, with the acetylome and

ubiquitylome having the highest proportion of lincRNAs and non-canonical RNA processed transcripts, respectively, while the HLA-I immunopeptidome contained the most out-of-frame ORFs. The HLA-I immunopeptidome yielded detection of >5 times more nuORFs than any other 'ome. Hence, while many nuORFs are translated and may be capable of becoming antigens, some are post-translationally modified and therefore may be involved in regulating cellular pathways.

We next asked if mutations resulting in LC-MS/MS detectable neoantigens were present in MONTE proteomes. None of the mutations contained within detected HLA-I neoantigens (Fig. 5D) were detected within tryptic peptides from the proteome. Given that 5/10 of the LUAD patient samples analyzed carry a KRAS G12X mutation that is contained within the tryptic peptide LVVVGAXGVGK, we examined the overlap in KRAS mutation detection between the immunopeptidome and proteome (Fig. 6C). We noted that three patients with KRAS G12V/D/C mutations (C3N-00169, C3N-00547, and C3L-02549) expressed HLA-I alleles that have been validated to present KRAS neoantigens[60,61]. Only the KRAS G12V 10mer was detected in HLA-A11 homozygous patient C3N-00547, which is more likely than the G12C and G12D variant to be presented (HLAthena %rank 0.08 vs. 0.31 and 1.32, respectively). Although our MONTE immunopeptidomes were not able to capture all validated KRAS neoantigens found using targeted mass spectrometry of an overexpression system and a cell line endogenously expressing KRAS G12V[60], this lack of detectability will diminish as more sensitive MS instrumentation and data generation approaches are introduced. Surprisingly, tryptic peptides containing the KRAS G12V mutation were not detected by LC-MS/MS in the MONTE proteome data or in our earlier 111-patient LUAD study from which the ten patient samples analyzed here were obtained[14]. Overall, 32/111 patients in the study had KRAS G12X mutations, of which only G12C, G12D, and G12S were detected as tryptic peptides (SAAV/SNV: 4/16, 2/7, and 2/2 patients respectively), while G12A and G12V were not detected (SAAV/SNV: 0/2 and 0/5 patients respectively). The lack of KRAS G12V tryptic peptide detection in the proteome suggests that low source protein expression was likely overcome by strong HLA-I binding and stability resulting in neoantigen detection.

The immunopeptidomes were also searched for PTM-modified peptides. We observed HLA-I and HLA-II phosphopeptides (fully localized) made up 0.11% and 0.3%, respectively, and acetylpeptides made up 0.08% and 0.10%, respectively, of total unique peptides (Supplementary Data 9). Prior studies have shown that position four in HLA-I peptides is the residue most often phosphorylated[63–65]. Consistent with these studies, we find most HLA-I phosphorylation sites (54%) on the fourth amino acid. However, acetylation appeared only 10% in the fourth position and more often in the first position (31%). As expected, HLA-I phosphopeptide detection was more likely to occur in patients with HLA-I alleles that contain proline in their binding motifs that correspond to the kinase substrate motifs of MAPK and CDK[63,66]. We next evaluated if the abundance of phospho- or acetyl-sites (Fig. 6D) and corresponding phospho- or acetyl-proteins detected in the tryptic proteome (Fig. 6E) impacts HLA-I and HLA-II peptide presentation. While many of the PTM-containing source proteins were also observed in their corresponding PTM-ome (HLA-I: phospho 87%, acetyl 50%; HLA-II: phospho 42%, acetyl 43%), we observed that few of the specific

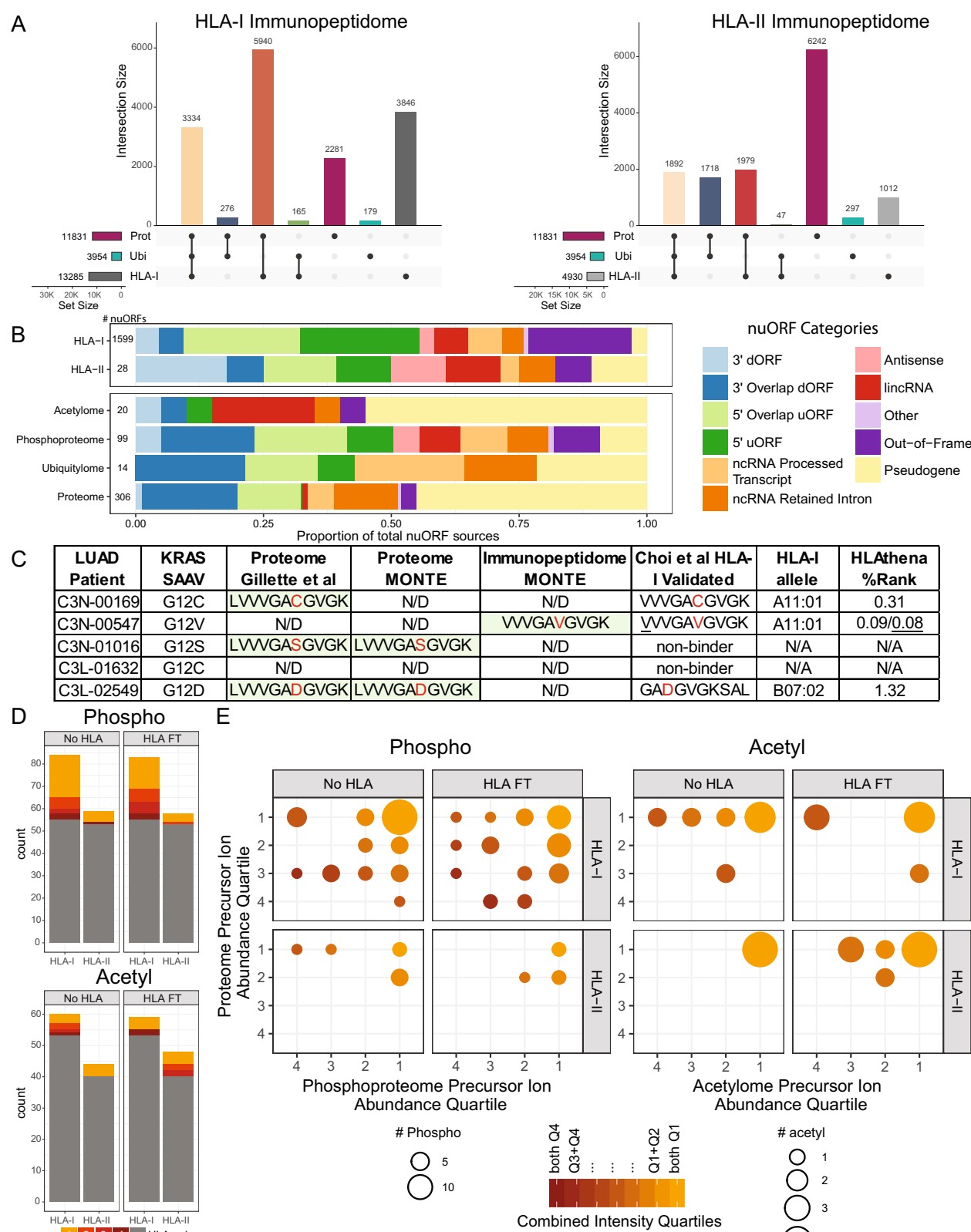

PTM sites presented by HLA were detectable in the tryptic phospho-proteomes and acetylomes (HLA-I: phospho 42%, acetyl 13%; HLA-II: phospho 10%, acetyl 17%). This may be due to the sequence context of HLA-presented PTM sites not being amenable to generating detectable tryptic peptides in the corresponding PTM-ome or the low abundance of these PTM sites that may be sampled by the HLA pathways. Of the

PTM sites that were observed in both the immunopeptidome and corresponding PTM-ome, we found that PTM-containing source pro-teins in the top abundance quartiles are most likely to result in HLA presentation. Thus, source proteins and PTM site abundance and the sequence context allowing for HLA binding are factors that should be evaluated in future PTM HLA peptide prediction efforts.

**Fig. 6 | Integration of LUAD MONTE immunopeptidome data with whole proteome and PTM-omes. A** The total number of proteins detected in proteome, ubiquitylome (ubiquityl-proteins), and immunopeptidome (source proteins) illustrated as 'UpSet' plots[93]. Vertical bars depict the number of uniquely or jointly detected features, as indicated by the layout matrix below. Color of vertical bars corresponds to each combination of sets. **B** Stacked bar chart showing proportion of detected nuORF source proteins by nuORF type[52] across the HLA-I and HLA-II immunopeptidomes (top) and tryptic proteomes, and PTM-omes (bottom). **C** Table of patient-specific KRAS mutations and their detection in different datasets, including: tryptic proteome of large cohort study of LUAD[14], MONTE tryptic proteome, and MONTE HLA-I immunopeptidome. KRAS neoantigens previously identified by LC-MS/MS[60] are shown as well as predictions for presentation for the best scoring HLA-I allele using HLAthena[8]. The wild-type KRAS G12 peptide was not

detected in the MONTE proteomes of this small subset of the LUAD patient samples with or without initial HLA IP, but it was detected in 23/25 LUAD TMT-10 plexes covering 102/111 patients in the original study[14]. **D** Stacked bar charts showing overlap of phosphorylated and acetylated HLA binding peptides with sites detected and unambiguously localized in the phospho- and acetyl-proteomes. Color indicates precursor ion intensity quartile of detected modification in the PTM-ome (1 = highest, 4 = lowest) with gray indicating that the site was uniquely detected in the immunopeptidome. **E** Marble plots show relative abundance of PTM sites identified in the immunopeptidome as detected in the phospho- or acetyl-proteome, as well as the abundance of corresponding phospho- or acetyl-proteins as detected in the proteome. Size of points encodes number of sites and color encodes combination of precursor intensity quartiles of PTM-ome and proteome. Source data for **A**, **B**, **D**, and **E** are provided as a Source Data file.

## Discussion

Discovery analyses that leverage patient tissue samples with limited input amounts face obstacles to deep and broad proteomic characterization. The MONTE workflow directly addresses this challenge by enabling serial HLA-II and HLA-I immunopeptidomics followed by ubiquitylome, proteome, phosphoproteome, and acetylome data collected from the same sample aliquot. After implementing HLA-II and HLA-I immunopeptidome and ubiquityl enrichment into an established serial proteome and PTM enrichment workflow, we observed high correlation between the proteomes, ubiquitylomes, phosphoproteomes, and acetylomes in both our breast cancer xenograft and LUAD datasets, showing that additional data layers can be acquired without prejudicing data quality and demonstrating the utility of the MONTE workflow. The order of 'omic analyses was determined by the biochemical requirements of each enrichment and previously established serial enrichment workflows. We anticipate that additional 'omes, such as phosphotyrosine peptide enrichment, could be incorporated into the MONTE workflow, and that these enhancements should be mindful of the compatibility of enrichment reagents and minimization of desalting steps to maximize peptide recovery. The current MONTE workflow can also be tailored to include or exclude enrichments based on the specific biological questions being addressed, demonstrating the flexibility of serial enrichment workflows. Overall, the MONTE workflow represents a path forward to deeply characterizing each single patient sample that was only possible previously with parallel processing of multiple tissue aliquots.

In proteogenomic studies of human tumor samples to date, ubiquitylomics has just begun to be used in parallel with the other 'omes[17] and HLA immunopeptidomics has not been routinely employed. Layering HLA-I and HLA-II immunopeptidomes on these other data types provides a window into the antigen landscape and improves our understanding of the rules that govern antigen processing and presentation. For example, patient C3N-00169 had a truncation mutation, E269*, in the proteasomal subunit PSMB7. We noted this patient expressed an HLA-A11 allele that has a lysine residue in the C-terminal anchor position. This observation could suggest that tryptic proteasomal subunits like PSMB7 may be under selection pressure in patients with HLA-I alleles that favor tryptic-like peptides. However, this hypothesis is based on just a single patient sample from the small cohort studied and will require additional studies in a larger sample set to validate. Furthermore, immunopeptidome and proteome datasets from the same sample could enable more accurate neoantigen and noncanonical HLA peptide prediction methods, as having both HLA presentation and protein expression data can be used to improve epitope prediction algorithms[7,8,67–70]. Although we demonstrate the usefulness of MONTE in a small LUAD cohort that expressed both HLA-I and HLA-II and where at least 50 mg of cryopulverized tissue was available, this workflow can be extended to other tumor types with less available tissue, low HLA expression, and unknown HLA-II expression. In these scenarios, performing the HLA serial enrichments will likely result in lower immunopeptidome depth, but will not prevent the

downstream multi-omic analyses, as these require less input material (25 mg wet weight or less) than the HLA enrichment. Even in cases where the HLA expression in tumors is low, useful information such as which HLA alleles are expressed and presenting peptides can be directly determined and leveraged to better understand changes in tumor HLA peptide presentation.

As noted with the KRAS G12V neoantigen and nuORF derived HLA peptides, epitope prediction based on tryptic proteome detection alone would likely under-represent the full neoantigen and noncanonical peptide repertoires. As such, MONTE immunopeptidome and proteome datasets from larger cohorts are required to fully understand how best to integrate tryptic proteome level mutation detection into epitope prediction workflows. Similarly, PTM-ome data combined with immunopeptidomics can uncover dependencies, such as PTM site abundance, that can be used to improve prediction of difficult to detect phosphorylated and acetylated HLA peptides. We remain intrigued by the observation that a majority of the PTM sites detected in unenriched immunopeptidome samples are not present in global phosphoproteome and acetylome data. It is possible that HLA-I and HLA-II immunopeptidomes may reveal undiscovered PTM sites because of their privileged access to rapidly degraded proteins and the autophagy and endosomal-lysosomal pathways, as well as access to regions of proteins not easily characterized using tryptic digestion. We envision that as the sensitivity of PTM enrichment improves, HLA immunopeptidomes can also be subjected to PTM enrichments in a serial fashion. Moreover, integrated MONTE datasets are likely to provide information regarding tumor immune cell infiltration status and dysregulation of signaling, degradation, and epigenetic pathways that can inform therapeutic intervention.

There are limitations to this study. Although it demonstrates the feasibility and utility of a workflow incorporating HLA-I and HLA-II immunopeptidomics and UbiFast ubiquitylomics into a serialized proteomic workflow using a clinically relevant sample set, its pilot-level scale precludes the statistically robust analyses, deep explorations of biology, or compelling assessments of the interplay between characterized 'omes that the approach is intended to facilitate. Rather than highlighting such underpowered and speculative results, we chose to focus on the added value and interpretable results provided by immunopeptidomic characterization of tumor samples. Recent large-scale cancer proteogenomics analyses have made a compelling case that the integration of proteomic, ubiquitylomic and especially phosphoproteomic data with genomic data helps to functionalize genomic aberrations, providing new perspectives on cancer biology and nominating potential therapeutic vulnerabilities[11]. Integration of diverse 'omics data types remains challenging, as each data type has distinct scaling, normalization, and transformation requirements to enable multi-omic interpretation. Missing values in each 'ome is also a limitation, as it may not be the case that genes of interest, their PTMs, or corresponding HLA-I and HLA-II peptides are observable due to stochastic sampling or for biological reasons. It also remains to be

shown that the integration of additional layers of data, such as the immunopeptidome, will continue to provide interpretable, and actionable insights. The MONTE workflow, when applied to samples from a suitably sized patient cohort, provides the means to test if the integration of the immunopeptidome, proteome, and PTM-omes will yield valuable biological insights.

High-throughput multi-omic data generation has proven to be a useful resource for understanding disease biology and identifying potential therapeutic targets[13,16,71–78]. By combining serial multi-ome enrichments with HLA-I and HLA-II immunopeptidomics into a single workflow, we have provided a method to understanding connections among antigen presentation, protein expression, signaling, protein degradation, and epigenetic regulation based on deep characterization of each single sample, which was only previously possible using parallel workflows that required multiple tissue samples. We also provide a publicly available multi-omic data viewer to enable researchers to explore these data and ask questions using a single or multiple gene names of interest. Further improvements to the MONTE workflow that address its current limitations will likely include decreasing the sample input further by incorporating low-input proteomic sample processing advances[79–82] and the incorporation of fully automated sample processing steps for all 'omes in the context of clinical trials. In addition to cancer, the MONTE workflow can be applied to the study of other disease states such as autoimmune and infectious diseases, and we anticipate that it will enable a comprehensive view of disease biology.

## Methods

### PDX and human tumor samples and cell lines

All experiments with live mice were performed according to institutional and national regulations and approved by the Institutional Animal Care and Use Committee at Washington University in St.Louis, MO. Patient-derived xenograft (PDX) tumors from established basal (WHIM6) and luminal (WHIM20) breast cancer subtypes were raised subcutaneously in 8-week-old NOD. Cg-Prkdcscid Il2rgtm1Wjl/SzJ mice (Jackson Labs, Bar Harbor, ME)[44,83]. Tumors from each animal were harvested by surgical excision at 1.5 cm³, below the maximum volume established by the animal committee.

LUAD samples were collected as part of the NIH/NCI CPTAC consortium (https://proteomics.cancer.gov/programs/cptac) with protocols mandated by the CPTAC program office. Data collection and analysis in this study was performed in accordance with the Declaration of Helsinki and Institutional review boards at tissue source sites reviewed protocols and consent documentation adhering to the CPTAC guidelines. Clinical data were obtained from tissue source sites and aggregated by an internal database called the CDR (Comprehensive Data Resource) that synchronizes with the CPTAC DCC (https://cptac-data-portal.georgetown.edu/). Clinical data can be accessed and downloaded from the DCC (Data Coordinating Center). Details about these samples have been published previously[14]. Information on participant compensation is not available to the investigators. This set of LUAD samples ($n = 10$) was chosen to represent important biological differences of high relevance to lung adenocarcinoma, as five samples were driven by KRAS mutations and five by EGFR mutations. Each driver mutation subset included samples from both men and women and both Asian and Western/Caucasian ethnicity were represented. Sex/gender was determined by self-reporting, and it was not considered for study design.

A375 cells were obtained from ATCC (ATCC ® CRL-1619). A375 cells were grown in ATCC-formulated Dulbecco's Modified Eagle's Medium (Catalog No. 30-2002) with fetal bovine serum to a final concentration of 10% using ATCC guidelines. A375 cells were harvested by trypsinization (Trypsin-EDTA 0.25%, Gibco™ 25200056), pelleted and rinsed in PBS twice. Pellets were snap frozen and stored at −80 °C.

### Processing of PDX tumor tissue (CompRef)

WHIM2 and WHIM16 patient-derived xenografts (PDX) underwent denaturing lysis in SDS to prepare for S-Trap digestion. Cryopulverized PDX samples were lysed in 500 μL SDS lysis buffer (5% SDS, 50 mM TEAB pH 8.5, 2 mM MgCl₂, 2 μg/ml Aprotinin, 10 μg/mL Leupeptin, 1 mM PSMF, 50 μM PR-619 (Lifesensors, SI9619: PR-619), 1 mM Chloroacetamide,10 mM NaF,1:100 dilution of Protease Inhibitor Cocktail 2 (Sigma-Aldrich, P5726), 1:100 dilution of Protease Inhibitor Cocktail 3 (Sigma-Aldrich, P0044), 10 mM Sodium Butyrate, 10 mM Nicotinamide). The samples were disrupted by gentle vortexing and incubated at room temperature for ~15 min. Samples were treated with 3 μL 250 units/μL Benzonase (Thomas Scientific, E1014-25KU) to shear DNA, mixed again, and incubated at room temperature for another ~15 min. The lysates were cleared by centrifugation for 15 min at 15,000 × g and the supernatant was prepared for S-Trap digestion. Protein concentration was estimated using a BCA protein assay. Disulfide bonds were reduced in 5 mM DTT for 30 min at 25 °C and 1000 rpm shaking, and cysteine residues alkylated in 10 mM IAA in the dark for 45 min at 25 °C and 1000 rpm shaking. Lysates were then transferred to a 15 mL conical tube to prepare for protein precipitation. 12% phosphoric acid was added at a 1:10 ratio of lysate volume to acidify, and proteins were precipitated with 6× sample volume of ice-cold S-Trap buffer (90% methanol, 100 mM TEAB). The precipitate was transferred in successive loads of 3 mL to a S-Trap Midi (Protifi) and loaded with 1 min centrifugation at 4000 × g, mixing the remaining precipitate thoroughly between transfers. The precipitated proteins were washed 4× with 3 mL S-Trap buffer at 4000 × g for 1 min. To digest the deposited protein material, 350 μL digestion buffer (50 mM TEAB) containing both trypsin and LysC, each at 1:50 enzyme:substrate, was passed through each S-Trap column with 1 min centrifugation at 4000 × g. The digestion buffer was then added back atop the S-Trap and the cartridges were left capped overnight at 25 °C.

Peptide digests were eluted from the S-Trap, first with 500 μL 50 mM TEAB and next with 500 μL 0.1% FA, each for 30 s at 1000 × g. The final elution of 500 μL 50% ACN/0.1% FA was centrifuged for 1 min at 4000 × g to clear the cartridge. Peptide concentration of the pooled elutions was estimated with a BCA assay. For the experiments shown in Fig. 2 and Supplementary Fig. 1, 0.5 mg aliquots of WHIM2 and WHIM16 peptides were created.

### Automated UbiFast K-ε-GG enrichment from CompRef tissue

Enrichment of K-ε-GG peptides using 0.5 mg of peptide per sample was performed using the automated UbiFast method[3,38] Briefly, peptide aliquots were reconstituted in 250 μL HS bind buffer (Cell Signaling Technology) w/ 0.01% CHAPS. All remaining steps for UbiFast enrichment excluding labeling and final bead collection contained 0.01% CHAPS. Reconstituted peptides were added to 5 μL PBS-washed HS anti-K-ε-GG antibody bead slurry with proprietary antibody amounts (Cell Signaling Technology, #59322) and incubated at 4 °C for 1 hour in a foil sealed KingFisher plate with end-over-end rotation. The plate containing peptides and anti-K-ε-GG antibody beads was then processed on the KingFisher. Briefly, bead-bound enriched peptides were washed with 50% ACN/50% HS wash buffer and washed again with PBS. K-ε-GG peptides were labeled on-bead with 400 μg TMTpro in 100 mM HEPES (prepared immediately before run) for 20 min and labeling was quenched with 2% hydroxylamine. Finally, the beads were washed with a HS wash buffer before being deposited into 100 μL PBS. All sixteen wells were combined, the supernatant was removed, and enriched peptides were eluted from the beads with 2 × 10 min 0.15% TFA. The eluate was desalted with a C18 stagetip, frozen, and dried in a vacuum centrifuge. For LC-MS/MS analysis, the K-ε-GG peptides were reconstituted in 9 μL 3% ACN/0.1% FA and 4 μL were injected twice onto an Orbitrap Exploris 480 mass spectrometer(Thermo Fisher Scientific) with FAIMS.

## UbiFast flow-through serial processing from CompRef tissue

For the proteome analysis shown in Fig. 2, peptides were acidified and desalted using a 50 mg tC18 SepPak cartridge. Eluates were frozen and a vacuum centrifuge was used to dry peptides. Peptides were reconstituted in 30% ACN, peptide concentration was determined using a BCA assay and peptides were dried again. Peptides corresponding to 0.25 mg from each sample (eight replicates of WHIM2 and eight replicates of WHIM16), were labeled with 0.5 mg TMTpro reagents in 20% ACN, 50 mM HEPES for 1 h. The TMT labeling reaction was quenched by adding 4 μL 5% hydroxylamine for 15 min at room temperature while shaking. Samples were combined into a 15 mL conical tube, frozen at −80 °C and dried in a vacuum centrifuge. The combined sample was desalted using a 200 mg tC18 SepPak cartridge and the eluate was snap frozen then dried in a vacuum centrifuge. Offline bRP fractionation was performed[19]. Briefly, peptides were separated over a 96 min gradient with a flow rate of 1 ml/min. The bRP solvent A was 5 mM ammonium formate, 2% ACN and solvent B was 5 mM ammonium formate, 90% ACN. 96 fractions were concatenated into 24 fractions for proteome analysis. For proteome analysis, 5% of each of the 24 fractions were transferred into HPLC vials, frozen and dried in a vacuum centrifuge.

For the experiment in Supplementary Fig. 1, UbiFast flowthrough peptides were acidified and desalted using a 50 mg tC18 SepPak cartridge. Eluates were frozen and a vacuum centrifuge was used to dry peptides. Peptides were reconstituted in 30% ACN, peptide concentration was determined using a BCA assay and peptides were dried again. UbiFast flowthrough peptides corresponding to 0.25 mg from each sample (four replicates of WHIM2 and four replicates of WHIM16) and non-UbiFast peptides (four replicates of WHIM2 and four replicates of WHIM16), were labeled with 0.5 mg TMTpro reagent in 20% ACN, 50 mM HEPES for 1 h. The TMT labeling reaction was quenched by adding 4 μL 5% hydroxylamine for 15 min at room temperature while shaking. Samples were combined into a 15 mL conical tube, frozen at −80 °C and dried in a vacuum centrifuge. The combined sample was desalted using a 200 mg tC18 SepPak cartridge and the eluate was snap frozen then dried in a vacuum centrifuge. Offline bRP fractionation was performed[19]. Briefly, peptides were separated over a 96 min gradient with a flow rate of 1 ml/min. The bRP solvent A was 5 mM ammonium formate, 2% ACN and solvent B was 5 mM ammonium formate, 90% ACN. 96 fractions were concatenated into 12 fractions and frozen before drying down in a vacuum centrifuge in preparation for phosphoproteome analysis.

## Automated IMAC phosphopeptide enrichment of CompRef tissue

For the experiments shown in Fig. 2, the remaining 95% of each bRP fraction was concatenated into 12 fractions and dried down before reconstituting to a final concentration of 80% ACN/0.1% TFA. For the experiment in Supplementary Fig. 1, the 12 concatenated fractions were reconstituted to a final concentration of 80% ACN/0.1% TFA. Phosphopeptides were enriched using the Agilent "AssayMAP Phosphopeptide Enrichment v2.1" protocol on an Agilent Bravo system. Briefly, 200 μL of sample was loaded onto AssayMap Fe(III)-NTA cartridges (Agilent, G5496-60085) at 5 μL/min. For the experiments shown in Fig. 2, the flow-through was collected and frozen for downstream acetyllysine enrichment. The cartridges were washed 3× with 80% ACN/0.1% TFA and phosphopeptides were eluted from the cartridges with 20 μL fresh 1% ammonium hydroxide into a plate containing 2.5 μL neat FA. Phosphopeptides were transferred to HPLC vials, frozen and dried in a vacuum centrifuge. For LC/MS-MS analysis, peptides were reconstituted in 9 μL 3% ACN/0.1% FA and 4 μL were injected from each of the 12 fractions.

## Acetyl-lysine immunoaffinity enrichment of CompRef tissue

Acetyl peptide enrichment was performed using the published protocol[14] with minor variations described below. Acetylated lysine peptides were enriched with 25 uL of PTMScan® Acetyl-Lysine Motif [Ac-K] immunoaffinity bead slurry with proprietary antibody amounts (PTMScan® Acetyl-Lysine Motif Kit #13416). Phosphopeptide-depleted IMAC flow-throughs were concatenated from 12 to 4 fractions (~750 μg peptides per fraction) and dried down using a SpeedVac apparatus. Prior to enrichment, antibody beads were washed 4x with IAP buffer (5 mM MOPS pH 7.2, 1 mM sodium phosphate [dibasic], 5 mM NaCl). Peptides were reconstituted with 1.4 mL of IAP buffer per fraction, added to washed beads, and incubated for 2 h at 4 °C. Bead-bound acetyl-enriched peptides were washed 4 times with ice-cold PBS followed by two elutions with 100 μLl of 0.15% TFA. Eluents were desalted using C18 stage tips, eluted with 50% ACN/0.1% FA, and dried down. Acetylpeptides were reconstituted in 7 μL of 3% ACN/0.1% FA and 4 μL were injected from each of the 4 fractions for LC-MS/MS analysis.

## LC-MS/MS analysis of CompRef tissue

All peptide samples were separated on an online nanoflow EASY-nLC 1200 UHPLC system (Thermo Fisher Scientific) and analyzed on an Orbitrap Exploris 480 mass spectrometer (Thermo Fisher Scientific) using Xcalibur 4.0. 1 μg of each proteome and fifty percent of each phosphopeptide, acetyl-lysine and K-ε-GG peptide sample was injected onto a capillary column (Picofrit with 10 μm tip opening/75 μm diameter, New Objective, PF360-75-10-N-5) packed in-house with 25 cm C18 silica material (1.9 μm ReproSil-Pur C18-AQ medium, Dr. Maisch GmbH, r119.aq). The UHPLC setup was connected with a custom-fit microadapting tee (360 μm, IDEX Health & Science, UH-753), and capillary columns were heated to 50 °C in column heater sleeves (PhoenixST) to reduce back pressure during UHPLC separation. For proteome and phosphoproteome samples, injected peptides were separated at a flow rate of 200 nL/min with a linear 85 min gradient from 100% solvent A (3% acetonitrile, 0.1% formic acid) to 30% solvent B (90% acetonitrile, 0.1% formic acid), followed by a linear 10 min gradient from 30% solvent B to 90% solvent B. For ubiquitin and acetyl-lysine samples, injected peptides were separated at a flow rate of 200 nL/min with a linear 120 min gradient from 100% solvent A (3% acetonitrile, 0.1% formic acid) to 35% solvent B (90% acetonitrile, 0.1% formic acid), followed by a linear 10 min gradient from 35% solvent B to 90% solvent B. Data-dependent acquisition was obtained using Xcalibur 4.4 software in positive ion mode at a spray voltage of 1.80 kV. MS1 Spectra were measured with a resolution of 60,000, an AGC target of 50% and a mass range from 300 to 1800 m/z. Up to 20 MS2 spectra per duty cycle were triggered at a resolution of 45,000, an AGC target of 300%, an isolation window of 0.7 m/z and a normalized collision energy of 34. Peptides that triggered MS2 scans were dynamically excluded from further MS2 scans for 20 s. For ubiquitin samples a FAIMS Pro Interface (Thermo Fisher Scientific) was in line with the mass spectrometer. The FAIMS device was operated in standard resolution mode at 100 °C, utilizing the compensation voltages (CVs) of −40, −60, and −80 for the first injection followed by a second injection with CVs of −40, −50, and −70.

## Data analysis of CompRef tissue

Mass spectrometry data was processed using Spectrum Mill v 7.08 (proteomics.broadinstitute.org). For all samples, extraction of raw files retained spectra within a precursor mass range of 800-6000 Da and a minimum MS1 signal-to-noise ratio of 25. MS1 spectra within a retention time range of +/−45 s, or within a precursor m/z tolerance of +/−1.4 m/z were merged. MS/MS searching of PDX samples was performed against a human and mouse RefSeq database with a release date of June 29, 2018 and containing 72,908 entries. Digestion parameters were set to "trypsin allow P" with an allowance of 4 missed cleavages. The MS/MS search included fixed modification of carbamidomethylation on cysteine. For TMT quantitation experiments TMTpro16 was searched using the full-mix function. Variable

modifications were acetylation of the protein N-terminus, oxidation of methionine, cyclization to pyroglutamic acid, deamidation, pyrocarbamidomethylation of cysteine and hydroxylation of proline. For PTM datasets, hydroxylation of proline was removed as a variable modification, and additional variable modifications were searched: phosphorylation of serine, threonine and tyrosine residues for IMAC enriched samples; diglycine modification of lysine residues for K(GG) enriched samples; lysine-acetylation for acetyl-lysine enriched samples. Restrictions for matching included a minimum matched peak intensity of 30% and a precursor and product mass tolerance of +/−20 ppm.

Peptide-spectrum matches were validated using a maximum false discovery rate (FDR) threshold of 0.8% for precursor charges 2 through 4 within each LC-MS/MS run, and 0.4% for precursor charges 5 and 6 within each directory of runs. TMTpro16 reporter ion intensities were corrected for isotopic impurities in the Spectrum Mill protein/peptide summary module using the afRICA correction method which implements determinant calculations according to Cramer's Rule. For proteome analysis, we required 2 or more fully quantified unique human peptides with a ratio count of 2 or more for protein identification and a ratio count of 2 or more for protein quantification. For PTM analysis, we filtered for fully quantified human proteins. To assign regulated proteins and PTM-sites we used the Proteomics Toolset for Integrative Data Analysis (Protigy, v0.9.1.3, Broad Institute, https://github.com/broadinstitute/protigy) to calculate moderated *t*-test *P* values. *P* values were adjusted for multiple hypothesis testing using the Benjamini−Hochberg method. Median/MAD normalization was performed on each TMT channel in each 'ome to center and scale the aggregate distribution of protein-level or PTM site−level log ratios around zero. Single sample Gene Set Enrichment Analysis (ssGSEA)[45] and site-centric PTM Signature Enrichment Analysis (PTM-SEA)[46] were performed as described in https://github.com/broadinstitute/ssGSEA2.0. Proteins, phosphorylation sites and acetylation sites were enriched using standard methods published by Subramanian et al.[45]. The C2: curated gene sets database from MSigDB[84] was used for enrichment.

## Serial Immunoprecipitation of HLA-I & HLA-II from LUAD tumor

Half of each of the ten cryopulverized LUAD patient tumors went through the HLA serial immunoprecipitation prior to multi-omic analysis. Each tumor was lysed with 4 °C lysis buffer (20 mM Tris pH 8.0, 100 mM NaCl, 6 mM MgCl2, 1 mM EDTA, 60 mM Octyl β-d-glucopyranoside, 0.2 mM Iodoacetamide, 1.5% Triton X-100, 1× Complete Protease Inhibitor Tablet-EDTA free, 1 mM PMSF, 10 mM NaF, 1:100 dilution of Protease Inhibitor Cocktail 2 (Sigma-Aldrich, P5726), 1:100 dilution of Protease Inhibitor Cocktail 3 (Sigma-Aldrich, P0044), 50 μM PR-619 (Lifesensors, SI9619: PR-619), 10 mM Sodium Butyrate (Sigma, B5887), 2 μM SAHA (Sigma,SML0061), 10 mM Nicotinamide (Sigma, N3376) obtaining a total of 1.2 ml lysate per tumor. Each lysate was moved into an Eppendorf tube, incubated on ice for 30 min with 2 μL of Benzonase (Thomas Scientific, E1014-25KU) to degrade nucleic acid and inverted after 15 min. The lysates were then centrifuged at 15,000 x g for 20 min at 4 °C and the supernatants were transferred to another set of Eppendorf tubes containing ~37.5 μL pre-washed Gammabind Plus Sepharose beads (Millipore Sigma, GE17-0886-01). The beads and lysate were rotated at 4 °C for one hour in order to preclear hydrophobic molecules and non-specifics that may interfere with the HLA IP.

The bead-lysate mixtures were centrifuged at 1500 × *g* for 1 min at 4 °C and each lysate was transferred to a tube containing ~37.5 μL prewashed beads and 15 μg of HLA-II antibody mix (9 μg TAL-1B5 (Abcam, ab20181), 3 μg EPR11226 (Abcam, ab157210), 3 μg B-K27 (Abcam, ab47342)). The HLA complexes were captured on the beads by incubating on a rotor at 4 °C for 3 h. Following the incubation all tubes were centrifuged at 1500 × *g* for 1 min at 4 °C and the lysates were

transferred from to new Eppendorf tubes containing ~37.5 μL prewashed beads and 15 μg of HLA-I antibody (W6/32) (Abcam, ab22432). The HLA-I antibody-bead-lysate mixture rotated for 3 h at 4 °C and was spun at 1500 × *g* for 1 min at 4 °C. The unbound lysates were transferred to new Eppendorf tubes and flash frozen with liquid nitrogen for multi-omic downstream analysis.

During HLA complex capture, a 10 μm PE fritted plate (Agilent, S7898A) was cut in half, placed on a Waters Positive Pressure Manifold, and washed using 1 mL acetonitrile and 3 × 1 mL room-temperature PBS. After each liquid addition, positive pressure of <5 psi was applied to the plate to achieve liquid movement. Immediately following each HLA capture, beads were resuspended in 1 mL cold PBS and transferred to one half of the pre-washed 10 μm PE fritted plate. Each tube was then rinsed with 500 μL cold PBS and remaining beads were transferred to the correct well. In total, four wash steps were performed to remove nonspecifically bound material: two washes with 2 mL of cold complete wash buffer (20 mM Tris pH 8.0, 100 mM NaCl, 1 mM EDTA, 6 mM Octyl β-d-glucopyranoside, 0.2 mM Iodoacetamide) and two washes with 2 mL of 10 mM Tris pH 8.0 buffer. The 10 μm PE fritted plate with dry HLA-II beads was wrapped with parafilm and stored at 4 °C until all HLA-I beads were washed on the other half of the plate and all samples were simultaneously prepared for mass spectrometry analysis via desalting.

## Desalt of HLA peptides using a positive pressure manifold

HLA peptides were eluted and desalted from beads as follows: 20 wells of the tC18 40 mg Sep-Pak desalting plate (Waters, Milford, MA) were activated with 2 × 1 mL of methanol (MeOH) and 500 μL of 99.9% acetonitrile (ACN)/0.1% formic acid (FA), then washed with 4 × 1 mL of 1% FA. The two halves of the 10μmPE fritted filter plate containing the beads were put together and placed on top of the Sep-Pak plate. To dissociate peptides from HLA molecules and facilitate peptides binding to the tC18 solid phase, 200 μL of 3% ACN/5% FA was added to the beads in the filter plate. 100 fmol internal retention time (iRT) standards (Biognosys SKU: Ki-3002-2) was spiked into each sample as a loading control and pushed through both the filter plate and 40 mg Sep-Pak plate. Following sample loading there was one wash with 400 μL of 1% FA. Beads were then incubated with 500 μL of 10% acetic acid (AcOH) three times for 5 min to further dissociate bound peptides from the HLA molecules. The beads were rinsed once with 1 mL 1% FA and the filter plate was removed. The Sep-Pak desalt plate was rinsed with 1 mL 1% FA an additional three times. The peptides were eluted from the Sep-Pak desalt plate using 250 μL of 15% ACN/1% FA and 2 × 250 μL of 50% ACN/1% FA. HLA peptides were eluted into 1.5 mL micro tubes (Sarstedt, Nümbrecht, Germany), frozen, and dried down via vacuum centrifugation. Dried peptides were stored at −80 °C until microscaled basic reverse phase separation.

Briefly, peptides were loaded on Stage-tips with 2 punches of SDB-XC material (Empore 3 M). HLA-I and HLA-II peptides were eluted in three fractions with increasing concentrations of ACN (HLA-I: 5%, 10%, and 30% in 0.1% NH4OH, pH 10; HLA-II: 5%, 15%, and 40% in 0.1% NH4OH, pH 10)[24]. Peptides were reconstituted in 3% ACN/5% FA prior to loading onto an analytical column (35 cm, 1.9 μm C18 (Dr. Maisch HPLC GmbH), packed in-house PicoFrit 75 μm inner diameter, 10 μm emitter (New Objective)). Peptides were eluted with a linear gradient (EasyNanoLC 1200, Thermo Fisher Scientific) ranging from 6–30% Solvent B (0.1% FA in 90% ACN) over 84 min, 30–90% B over 9 min and held at 90% B for 5 min at 200 nl/min. MS/MS data were acquired on a Orbitrap Exploris 480 mass spectrometer(Thermo Fisher Scientific) equipped with (HLA-I) and without (HLA-II) FAIMS (Thermo Fisher Scientific) in data-dependent acquisition. FAIMS compensation voltages (CVs) were set to −50 and −70 with a cycle time of 1.5 s per FAIMS experiment. MS2 fill time was set to 100 ms; collision energy was 30 CE for HLA-I and 34 CE for HLA-II.

## Serial ubiquitylome, proteome, phospho- and acetyl-ome of LUAD

Each set of 10 replicate tumors underwent denaturing lysis in SDS to prepare for S-Trap digestion. Flow-throughs of the HLA-I IP, at this point in native HLA lysis buffer and stored as flash-frozen unbound lysates, were briefly thawed on ice for ~15 min. Once thawed, 10% SDS was added for a final concentration of 2.5% SDS to denature the lysate, resulting in a final volume of ~1.5 mL lysate which was prepared for S-Trap digestion.

Replicates of the HLA-depleted samples were lysed from cryopulverized tissue in 1 mL 5% SDS buffer (5% SDS, 50 mM TEAB pH 8.5, 2 mM MgCl$_2$). The samples were disrupted by pipette mixing and gentle vortexing and incubated at room temperature for ~10 min. Samples were treated with 2 μL benzonase to shear DNA, mixed again, and incubated at room temperature for another ~20 min. Finally, non–HLA-depleted lysates were homogenized with a probe sonicator for 30 s and left to lyse again for ~10 min. The lysates were cleared by centrifugation for 15 min at 15,000 × g and the supernatant was prepared for S-Trap digestion.

In both sets of LUAD tumors, all further processing steps were executed identically. Protein concentration was estimated using a BCA assay for scaling of digestion enzymes. Disulfide bonds were reduced in 5 mM DTT for 30 min at 25 °C and 1000 rpm shaking and cysteine residues were alkylated in 10 mM IAA in the dark for 45 min at 25 °C and 1000 rpm shaking. Lysates were then transferred to a 15 mL conical tube to prepare for protein precipitation. 27% phosphoric acid was added at a 1:10 ratio of lysate volume to acidify and proteins were precipitated with 6× sample volume of ice cold S-Trap buffer (90% methanol, 100 mM TEAB). The precipitate was transferred in successive loads of 3 mL to a S-Trap Midi (Protifi) and loaded with 1 min centrifugation at 4000 × g, mixing the remaining precipitate thoroughly between transfers. The precipitated proteins were washed 4× with 3 mL S-Trap buffer at 4000 × g for 1 min. To digest the deposited protein material, 350 μL digestion buffer (50 mM TEAB) containing both trypsin and endopeptidase C (LysC), each at 1:50 enzyme:substrate, was passed through each S-Trap column with 1 min centrifugation at 4000 × g. The digestion buffer was then added back atop the S-Trap and the cartridges were left capped overnight at 25 °C.

Peptide digests were eluted from the S-Trap, first with 500 μL 50 mM TEAB and next with 500 μL 0.1% FA, each for 30 sec at 1000 × g. The final elution of 500 μL 50% ACN/0.1% FA was centrifuged for 1 min at 4000 × g to clear the cartridge. Peptide concentration of the pooled elutions was estimated with a BCA assay, divided into 750 μg peptide aliquots for K-ε-GG enrichment, snap frozen, and dried in a vacuum centrifuge.

### Automated UbiFast K-ε-GG enrichment of LUAD

Peptides containing the K-ε-GG tryptic remnant of ubiquitin/ubiquitin-like small protein modifications were enriched using an adaptation of the UbiFast protocol for the Thermo KingFisher automation platform[38]. Briefly, 750 μg peptide aliquots were reconstituted in 250 μL CST HS bind buffer with 0.01% CHAPS. All following steps for UbiFast enrichment excluding labeling and final bead collection contained 0.01% CHAPS. Reconstituted peptides were added to 5 μL PBS-washed HS anti-K-ε-GG antibody bead slurry with proprietary antibody amounts (Cell Signaling Technology, #59322) and incubated at 4 °C for 1 h in a foil sealed KingFisher plate with end-over-end rotation. Following removal of the beads from the incubation by the KingFisher robot, the incubation plate containing non-TMT labeled, K-ε-GG–depleted peptide flow-through was sealed and frozen for downstream proteome, phosphoproteome, and acetylproteome processing. Briefly, bead-bound enriched peptides were washed with 50% ACN/50% CST HS wash buffer and washed again with PBS. K-ε-GG peptides were labeled on bead with 400 μg TMT 10 reagent in 100 mM HEPES (prepared immediately before run) for 20 min and labeling was quenched with 2% hydroxylamine. Finally, the beads were washed with a CST HS wash buffer before being deposited into 100 μL PBS containing no CHAPS buffer. Each well containing each TMT channel was combined by 10-plex, the supernatant was removed, and enriched peptides were eluted from the beads with 2 × 10 min 0.15% TFA. The eluate was desalted with a C18 stagetip, frozen, and dried in a vacuum centrifuge. For LC-MS/MS analysis, the unfractionated K-ε-GG peptides were reconstituted in 9 μL 3% ACN/0.1% FA and 4 μL was injected twice back-to-back for each sample.

### TMT labeling of UbiFast flow-through for serial proteome

Non-TMT labeled, K-ε-GG-depleted peptide flow-throughs of the K-ε-GG IPs were acidified with neat formic acid to a final concentration of 1% FA and desalted with 100 mg tC18 SepPak cartridges. Eluates were frozen and dried in a vacuum centrifuge. Peptides were reconstituted in 30% ACN/0.1% FA, peptide concentration was estimated using a BCA assay, and peptides were aliquoted for downstream processing and dried again. 300 μg of each sample was reconstituted in 60 μL 50 mM HEPES and labeled with 300 μg TMT 10 reagent at a final concentration of 20% ACN for 1 h at 25 °C and 1000 rpm. Each tumor replicate was assigned the same TMT channel in its corresponding TMT 10-plex for an identical experimental design. Labeling reactions were diluted to 2.5 mg/mL with 50 mM HEPES. Complete labeling and balancing of input material were confirmed. TMT labeling was quenched with 3 μL 5% hydroxylamine for 15 min and each TMT 10-plex was combined, frozen, and dried. Dried, labeled, and combined peptides were reconstituted with 3 mL 1% FA and desalted with a 200 mg tC18 SepPak. The eluate was snap frozen and dried in a vacuum centrifuge.

Offline bRP fractionation was performed as described previously and above[19]. Briefly, peptides were separated over a 96-minute gradient with a flow rate of 1 ml/min. Solvent A was 5 mM ammonium formate/2% ACN and solvent B was 5 mM ammonium formate/90% ACN. 96 fractions were concatenated into 24 fractions for proteome analysis. 5% of each of the 24 fractions were transferred into HPLC vials, frozen, and dried in a vacuum centrifuge for analysis. The remaining 95% of each fraction was concatenated into 13 fractions for phosphopeptide enrichment. Proteome fractions were reconstituted in 3% ACN/0.1% FA and 500 ng at 0.25 μg/μL from each of the 24 fractions was injected for LC-MS/MS analysis.

### LUAD Automated IMAC phosphopeptide enrichment

IMAC enrichment of phosphopeptides was performed using AssayMap Fe(III)-NTA cartridges (Agilent, G5496-60085). Concatenated fractions were solubilized with 80 μL 50% ACN/0.1% TFA in a bath sonicator for 5 min followed by addition of 120 μL 100% ACN/0.1% TFA for a final concentration of 80% ACN/0.1% TFA. Peptide solution was clarified by centrifugation at 6000 × g for 5 min and 160 μL was transferred to a 96 well plate for enrichment. The remaining 40 μL was set aside for re-enrichment. The Agilent "AssayMAP Phosphopeptide Enrichment v2.1" protocol was used. Briefly, the syringes were rinsed with HPLC water and primed with 50% ACN/0.1% TFA. Cartridges were equilibrated with 80% ACN/0.1% TFA.160 μL of sample was loaded at 5 μL/min and the phosphopeptide-depleted flow-through was collected and frozen for downstream acetyl-lysine enrichment. The cartridges were washed 3× with 80% ACN/0.1% TFA to remove nonspecific peptides. Enriched phosphopeptides were eluted from the cartridges with 20 μL fresh 1% ammonium hydroxide at 5 uL/min into a plate containing 2.5 μL neat FA. Phosphopeptide-enriched eluates were transferred to HPLC vials, frozen, and dried in a vacuum centrifuge. For LC/MS-MS analysis, peptides were reconstituted in 9 μL 3% ACN/0.1% FA and 4 μL was injected from each of the 12 fractions.

### Acetyl-lysine immunoaffinity enrichment of LUAD and A375

Acetyl peptide enrichment was performed using the published protocol[14] with minor variations described below. Acetylated lysine

peptides were enriched with 25 uL of PTMScan® Acetyl-Lysine Motif [Ac-K] immunoaffinity bead slurry with proprietary antibody amounts (PTMScan® Acetyl-Lysine Motif Kit #13416). For the unfractionated A375 acetyl-lysine enrichments, 25 uL of beads was used per sample. For the LUAD samples, phosphopeptide-depleted IMAC flow-throughs were concatenated from 12 to 4 fractions (~750 µg peptide per fraction) and dried down using vacuum centrifugation. Prior to enrichment, antibody beads were washed 4x with IAP buffer (5 mM MOPS pH 7.2, 1 mM sodium phosphate [dibasic], 5 mM NaCl). Peptides were reconstituted with 1.4 mL IAP buffer per fraction, added to washed beads, and incubated for 2 h at 4 °C. Bead-bound acetyl-enriched peptides were washed 4× with ice-cold PBS followed by two elutions with 100 µL 0.15% TFA. Eluents were desalted using C18 stage tips, eluted with 50% ACN/0.1% FA, and dried down using vacuum centrifugation. Acetylpeptides were reconstituted in 7 µL of 3% ACN/0.1% FA and 4 µL was injected from each of the 4 fractions for LC-MS/MS analysis.

## LC-MS/MS data acquisition of LUAD samples processed by MONTE

Online separation was done with a nanoflow Proxeon EASY-nLC 1200 UHPLC system (Thermo Fisher Scientific). In this set up, the LC system, column, and platinum wire used to deliver electrospray source voltage were connected via a stainless steel cross (360 mm, IDEX Health & Science, UH-906x). The column was heated to 50 °C using a column heater sleeve (Phoenix-ST). Each sample was injected onto an in-house packed 27 cm× 75 µm internal diameter C18 silica picofrit capillary column (1.9 mm ReproSil-Pur C18-AQ beads, Dr. Maisch GmbH, r119.aq; Picofrit 10 µm tip opening, New Objective, PF360-75-10-N-5). Mobile phase flow rate was 200 nL/min, comprising 3% acetonitrile/0.1% formic acid (Solvent A) and 90% acetonitrile/0.1% formic acid (Solvent B). The same LC and column setup were used for ubiquitylome, proteome, phosphoproteome, and acetylproteome analyses. Each LC-MS/MS method consisted of a 10 min column-equilibration procedure, a 20 min sample-loading procedure, and the following gradient profiles (min:%B): ubiquitylome (154 min) = 0:2, 2:6, 122:35, 130:60, 133:90, 143:90, 144:50, 154:50; proteome/phosphoproteome (110 min) = 0:2, 1:6, 85:30, 94:60, 95:90, 100:90, 101:50, 110:50; acetylome (260 min) = 0:2, 1:6, 235:30, 244:60, 245:90, 250:90, 251:50, 260:50. The flow rate of the last two steps of each gradient was increased to 500 nL/min.

For ubiquitylome, proteome, phosphoproteome, and acetylproteome analysis, samples were analyzed with a Orbitrap Exploris 480 mass spectrometer(Thermo Fisher Scientific) with Xcalibur 4.0 equipped with a NanoSpray Flex NG ion source. Data-dependent acquisition was performed using Orbitrap Exploris 480 V2.0 software in positive ion mode at a spray voltage of 1.8 kV. MS1 spectra were measured with a resolution of 60,000, a normalized AGC target of 300% for proteome/phosphoproteome and 100% for ubiquitylome/acetylome, a maximum injection time of 10 ms, and a mass range from 350 to 1800 $m/z$. The data-dependent mode cycle was set to trigger MS/MS on up to the top 20 most abundant precursors per cycle at an MS2 resolution of 45,000, an AGC target of 30% for proteome/phosphoproteome and 50% for ubiquitylome/acetylome, an isolation window of 0.7 $m/z$, a maximum injection time of 105 ms for proteome/phosphoproteome and 120 ms for ubiquitylome/acetylome, and an HCD collision energy of 34%. Peptides that triggered MS/MS scans were dynamically excluded from further MS/MS scans for 20 s in proteome/phosphoproteome/ubiquitylome and for 30 s in acetylome, with a ±10 ppm mass tolerance. Theoretical precursor envelope fit filter was enabled with a fit threshold of 50% and window of 1.2 $m/z$. Monoisotopic peak determination was set to peptide and charge state screening was enabled to only include precursor charge states 2–6 with an intensity threshold of 5.0e3. Advanced peak determination (APD) was enabled. "Perform dependent scan on single charge state per precursor only" was disabled.

## LUAD MONTE LC-MS/MS data interpretation

MS/MS spectra from all 'omes were interpreted using Spectrum Mill (SM) v 7.08 (proteomics.broadinstitute.org) to provide identification and relative quantitation at the protein, peptide, and PTM-site (ubiquityl, phospho, and acetyl) site levels.

## Variant calls

Individual variant/indel.vcf files for each of the 10 LUAD patients in this study were extracted from the CPTAC Pancancer Harmonized Callset v1.1 which is the harmonized result of processing whole exome sequencing data from 10 CPTAC cancer cohorts independently through the variant calling pipelines of the Getz laboratory at the Broad Institute and the Ding laboratory at Washington University in St Louis. The Getz laboratory pipeline consists of GATK (v4.1.4.1) for DNA sequence data quality control and somatic copy number analysis, MuTect[57] Manta+Strelka v2[85,86] for discovery of somatic and germline SNVs and INDELs, DeTiN v1.8.9[87] and GATK4 Funcotator ver GATK 4.1.4.1 for post-discovery filtering followed by merging of adjacent somatic SNPs into DNPs, TNPs, and ONPs. The Ding laboratory employed the Somaticwrapper pipeline v1.6 (https://github.com/ding-lab/somaticwrapper), which includes four different callers: Strelka v.2[85,88], MUTECT v1.7[57], VarScan v.2.3.8[89], and Pindel v.0.2.5[90]. Rare mutations with VAF of [0.015, 0.05] in cancer driver genes were rescued based on the gene consensus list reported by Bailey et al.[91]. COCOON (https://github.com/ding-lab/COCOONS) was used to combine adjacent SNVs into DNPs.

## Personalized sequence database

For searching with LC-MS/MS datasets from all 'omes, we generated a personalized protein sequence database starting with a base human reference proteome to which we appended somatic and germline variants and indels for each of the 10 LUAD patients. The base proteome consisted of the human reference proteome Gencode 34 (ftp.ebi.ac.uk/pub/databases/gencode/Gencode_human/release_34/) with 47,429 non-redundant protein coding transcript biotypes mapped to the human reference genome GRCh38, 602 common laboratory contaminants, 2043 curated smORFs (lncRNA and uORFs), 237,427 novel unannotated ORFs (nuORFs) supported by ribosomal profiling nuORF DB v1.0[52], and 4,167 TCGA shared mutations from 26 tumor types (https://www.cancer.gov/tcga) for a total of 355,028 entries which yield 16,973,937 distinct 9-mers. The nuORFs alone yield 8,612,372 distinct 9-mers and thus increase the peptide search space by only a factor of ~2. The personalized protein sequence entries were prepared by processing each individual patient's somatic and germline variant calls from whole exome sequencing data, described above, using QUILTS v3[57–59] with no further variant quality filtering using a Ensembl v100 reference proteome and reference genome for sequence identifiers consistent with the variant calling. Gencode v34 is a contemporaneous subset of Ensembl v100 (March 2020). Using the SM Protein Database utilities, the base reference proteome and individual patient proteomes were combined and redundancy removed to produce a cohort-level protein sequence database and a variant summary table to enable subsequent mapping of sequence variants identified in TMT-multiplexed LC-MS/MS datasets back to individual patients.

## Spectrum quality filtering

Using the SM Data Extractor module for HLA-I and HLA-II immunopeptidomes, spectral merging was disabled, the precursor MH + inclusion range was 600–4000, and the spectral quality filter was a sequence tag length >1 (i.e., minimum of three peaks separated by the in-chain masses of two consecutive amino acids). For non-HLA 'omes,

similar MS/MS spectra with the same precursor $m/z$ acquired in the same chromatographic peak were merged, the precursor MH + inclusion range was 800–6000, and the spectral quality filter was a sequence tag length > 0.

## MS/MS search conditions

Parameters for the SM MS/MS search module for HLA-I and HLA-II immunopeptidomes included: no enzyme specificity; precursor and product mass tolerance of ±10 ppm; minimum matched peak intensity of 30%; ESI-QEXACTIVE-HCD-HLA-v3 scoring; fixed modification: carbamidomethylation of cysteine; variable modifications: cysteinylation of cysteine, oxidation of methionine, deamidation of asparagine, acetylation of protein N-termini, and pyroglutamic acid at peptide N-terminal glutamine; and precursor mass shift range of −18 to 81 Da. A second round search of remaining unassigned spectra was done with revised variable modifications to also allow for acetylation of lysine and phosphorylation of serine, threonine, and tyrosine with a precursor MH + shift range of −18 to 125 Da.

For non-HLA 'omes, parameters included: "trypsin allow P" enzyme specificity with up to 4 missed cleavages, precursor and product mass tolerance of ±20 ppm, and 30% minimum matched peak intensity (40% for acetylome). Scoring parameters were ESI-QEXACTIVE- HCD-v2 for whole proteome datasets and ESI-QEXACTIVE-HCD-v3 for phosphoproteome, acetylome, and ubiquitylome datasets. Allowed fixed modifications included carbamidomethylation of cysteine and selenocysteine. TMT labeling was required at lysine, but peptide N-termini were allowed to be either labeled or unlabeled. Allowed variable modifications for whole proteome datasets were acetylation of protein N-termini, oxidized methionine, deamidation of asparagine, hydroxylation of proline in PG motifs, pyro-glutamic acid at peptide N-terminal glutamine, and pyro-carbamidomethylation at peptide N-terminal cysteine with a precursor MH + shift range of −18 to 97 Da. For all PTM-omes, variable modifications were revised to omit hydroxylation of proline and allow deamidation only in NG motifs. The phosphoproteome was revised to allow phosphorylation of serine, threonine, and tyrosine with a precursor MH + shift range of −18 to 272 Da. The acetylome was revised to allow acetylation of lysine with a precursor MH + shift range of −400 to 70 Da. The ubiquitylome was revised to allow diglycine modification of lysine with a precursor MH + shift range of −375 to 70 Da.

## PTM site localization

Using the SM Autovalidation and Protein/Peptide Summary modules, the PTM-ome dataset results were filtered and reported at the ubiquityl, phospho, and acetyl site levels. When calculating scores at the variable modification (VM) site level and reporting the identified VM sites, redundancy was addressed in SM as follows: a VM site table was assembled with columns for individual TMT-plex experiments and rows for individual VM sites. PSMs were combined into a single row for all non-conflicting observations of a particular VM site (e.g., different missed cleavage forms, different precursor charges, confident and ambiguous localizations, and different sample-handling modifications). For related peptides, neither observations with a different number of VM sites nor different confident localizations were allowed to be combined. Selecting the representative peptide for a VM site from the combined observations was done such that once confident VM site localization was established, higher identification scores and longer peptide lengths were preferred. While an SM PSM identification score was based on the number of matching peaks, their ion type assignment, and the relative height of unmatched peaks, the VM site localization score was the difference in identification score between the top two localizations. The score threshold for confident localization, >1.1, corresponded to at least 1 b- or y-ion located between two candidate sites that has a peak height > 10% of the tallest fragment ion (neutral losses of phosphate from the precursor and related ions as well as immonium and TMT reporter ions were excluded from the relative height calculation). The ion type scores for b-H3PO4, y-H3PO4, b-H2O, and y-H2O ion types were all set to 0.5. This prevented inappropriate confident localization assignment when a spectrum lacked primary b- or y-ions between two possible sites but contained ions that could be assigned as either phosphate-loss ions for one localization or water-loss ions for another localization.

## Protein grouping of PSMs, peptides, and PTM sites

Using the SM Autovalidation and Protein/Peptide summary modules, results were filtered and reported at the protein level. Identified proteins were combined into the same protein group if they shared a peptide with sequence length >8. A protein group could be expanded into subgroups (isoforms or family members) when distinct peptides were present that uniquely represent a subset of the proteins in a group. For the proteome dataset, the protein grouping method "expand subgroups, top uses shared" (SGT) was employed, which allocates peptides shared by protein subgroups only to the highest scoring subgroup containing the peptide. For the PTM-ome datasets, the protein grouping method "unexpand subgroups" was employed, which reports a VM site only once per protein group allocated to the highest scoring subgroup containing the representative peptide. The SM protein score is the sum of the scores of distinct peptides. A distinct peptide is the single highest scoring instance of a peptide detected through an MS/MS spectrum. MS/MS spectra for a particular peptide may have been recorded multiple times (e.g., as different precursor charge states, in adjacent bRP fractions, modified by deamidation at Asn or oxidation of Met, or with different phosphosite localization), but are still counted as a single distinct peptide.

## Peptide-spectrum match filtering and false discovery rates

Using the SM Autovalidation module, peptide-spectrum matches (PSMs) for individual spectra were confidently assigned by applying target-decoy based FDR estimation to achieve <1.0% FDR at the PSM, peptide, VM site, and protein levels. For HLA-I and -II immunopeptidomes, PSM-level thresholding was done with a minimum peptide length of 7, minimum backbone cleavage score of 5, and <1.0% FDR across all three fractions. Allowed precursor charges were HLA-I: 1–4, HLA-II: 2–6. Immunopeptidomics data were further filtered to remove non-human contaminants, peptides that match peptides identified in blank bead negative control IPs[7,8], and tryptic contaminant peptides. Phospho and acetyl HLA peptides were quality filtered to include matches with scores >6 and scored peak intensity >60%; HLA-I data included only 8–11mers.

For the whole proteome dataset, thresholding was done in three steps: at the PSM level, at the protein level for each TMT-plex, and at the protein level for the cohort of two TMT-plexes obtained with and without initial HLA IP. For the PTM-omes (ubiquitylome, phosphoproteome, and acetylome), dataset thresholding was done in two steps: at the PSM level for each TMT-plex and at the VM site level for the cohort of two TMT-plexes. In step 1 for all datasets, PSM-level autovalidation was done first and separately for each TMT-plex experiment using an auto-thresholds strategy with a minimum sequence length of 7, automatic variable range precursor mass filtering, and with score and delta Rank1-Rank2 score thresholds optimized to yield a PSM-level FDR estimate for precursor charges 2–4 of <0.8% for each precursor charge state in each LC-MS/MS run. To achieve reasonable statistics for precursor charges 5–6, thresholds were optimized to yield a PSM-level FDR estimate of <0.4% across all runs per TMT-plex experiment (instead of per each run), since many fewer spectra are generated for the higher charge states.

In step 2 for the PTM-ome datasets, VM site polishing autovalidation was applied across both TMT-plexes to retain all VM site identifications with either a minimum ID score of 8.0 or observation in both TMT-plexes. The intention of the VM site polishing step is to

control FDR by eliminating unreliable VM site–level identifications, particularly low-scoring VM sites that are only detected as low-scoring peptides that are also infrequently detected across both TMT-plexes in the study. Using the SM Protein/Peptide Summary module to make VM site reports, the ubiquitylome and acetylome datasets were further filtered to remove peptides ending with the regular expression [^K][^K] k since trypsin and Lys-C cannot cleave at a ubiquitylated or acetylated lysine. The [^K] means retain if unmodified Lys present in one of the last two positions to allow for a missed cleavage with ambiguous PTM-site localization.

In step 2 for the whole proteome dataset, protein polishing autovalidation was applied separately to each TMT-plex experiment to further filter the PSMs using a target protein–level FDR threshold of zero. The primary goal of this step was to eliminate peptides identified with low-scoring PSMs that represent proteins identified by a single peptide, so-called "one-hit wonders." After assembling protein groups from the autovalidated PSMs, protein polishing determined the maximum protein level score of a protein group that consisted entirely of distinct peptides estimated to be false-positive identifications (PSMs with negative delta forward-reverse scores). PSMs were removed from the set obtained in the initial peptide level autovalidation step if they contributed to protein groups that had protein scores below the maximum false-positive protein score. Step 3 was then applied, consisting of protein polishing autovalidation across both TMT-plexes together using the protein grouping method "expand subgroups, top uses shared" to retain protein subgroups with either a minimum protein score of 25 or observation in both TMT-plexes. The primary goal of this step was to eliminate low-scoring proteins that were infrequently detected in the sample cohort. As a consequence of these two protein polishing steps, each identified protein reported in the study comprised multiple peptides, unless a single excellently scoring peptide was the sole match and that peptide was observed in both TMT-plexes.

### FDR filtering for neoantigens, nuORFs, and somatic variants
All MS/MS spectra of neoantigens were manually inspected and labeled spectra are provided in Fig. 5D and Supplementary Fig. 6. While the aggregate FDR for each dataset was set to <1%, as described above, FDR for certain subsets of rarely observed classes (<5% of total) of peptides, PTM sites, and proteins required more stringent score thresholding to reach a suitable subset-specific FDR < 1.0%. To this end, we devised and applied subset-specific filtering approaches.

Subsets of nuORF types were thresholded independently in the HLA and PTM-ome datasets using a two-step approach. First, PSM scoring metric thresholds were tightened in a fixed manner for all nuORF PSMs so that nuORF distributions for each metric improved to meet or exceed the aggregate distributions. For all 'omes, the fixed thresholds were: minimum score: 7, minimum percent scored peak intensity: 50%, precursor mass error: ± <5 ppm. For HLA 'omes, minimum backbone cleavage score (BCS): 5, sequence length: 8–12 (HLA-I), 9–50 (HLA-II). For PTM-omes, these fixed thresholds were: minimum score: 7, minimum backbone cleavage score (BCS): 4, sequence length: 7–50. Second, individual nuORF type subsets with FDR estimates remaining above 1% were further subject to a grid search to determine the lowest values of BCS (sequence coverage metric) and score (fragment ion assignment metric) that improved FDR to <1% for each ORF type in the dataset for each 'ome.

The subset of peptides containing single amino acid variants (SAAVs) and indels observed in the proteome was extracted after step 1 of PSM filtering described above using the SM Protein/Peptide Summary module to create a proteogenomics (PG) site report with quantitation normalized to nullify the effect of differential protein loading using the aggregate protein-level normalization factors from the fully filtered proteome dataset. The PG site report was manually filtered to

the final subset of somatic SAAVs and indels by retaining those in which the TMT ratios were extremely high only for the patients in which the corresponding SNV or indel was observed.

### Quantitation using TMT ratios
Using the SM Protein/Peptide Summary module, a protein comparison report was generated for the proteome dataset using the protein grouping method "expand subgroups, top uses shared" (SGT). For the PTM-ome datasets—ubiquitylome, phosphoproteome, and acetylome —VM site comparison reports limited to either ubiquityl, phospho, or acetyl sites, respectively, were generated using the protein grouping method "unexpand subgroups." Relative abundances of proteins and VM sites were determined in SM using TMT reporter ion log2 intensity ratios from each PSM. TMT reporter ion intensities were corrected for isotopic impurities in the SM Protein/Peptide Summary module using the afRICA correction method, which implements determinant calculations according to Cramer's Rule and correction factors obtained from the reagent manufacturer's certificate of analysis (https://www.thermofisher.com/order/catalog/product/90406) for TMT-10 lot number UA280170. Each protein-level or PTM site–level TMT ratio was calculated as the median of all PSM-level ratios contributing to a protein subgroup or PTM site. PSMs were excluded from the calculation if they lacked a TMT label, had a precursor ion purity <50% (MS/MS has significant precursor isolation contamination from co-eluting peptides), or had a negative delta forward-reverse identification score (half of all false-positive identifications). Using the SM Process Report module, non-quantifiable proteins and PTM sites (e.g., unlabeled peptides containing an acetylated protein N-terminus and ending in arginine rather than lysine) were removed and median/MAD normalization was performed on each TMT channel in each 'ome to center and scale the aggregate distribution of protein-level or PTM site–level log ratios around zero in order to nullify the effect of differential protein loading and/or systematic MS variation. Within subsets of an 'ome (e.g., nuORFs or SAAVs), the TMT ratios were normalized using the normalization factors for the aggregate distribution of the corresponding 'ome.

### HLA peptide prediction using HLAthena
HLA peptide prediction was performed using HLAthena[8]. Unless otherwise specified, peptides were assigned to an allele using a percentile rank cutoff ≤ 0.5.

### Synthetic peptide analysis of LC-MS/MS detected neoantigens
Synthetic peptides were purchased from Vivitide, LLC (Gardner, MA) for the MS/MS spectra comparisons shown in Supplementary Fig. 6. Synthetic peptides were analyzed at 10, 50, and 100 fmol/μL without background and at 5 fmol/μL spiked into an HLA-I immunopurification of 25 million A375 cells that was prepared and analyzed using the methods described above with the following deviations. The synthetic peptide data were collected on a Orbitrap Exploris 480 mass spectrometer(Thermo Fisher Scientific) equipped with a NanoSpray Flex NG ion source. All experimental and synthetic peptides had a similar abundance with the exception of ISNDLYLTL that was an order of magnitude lower in intensity when compared to the synthetic.

### Retention Time Prediction using DeepLC
The retention times of HLA-I peptides were predicted using DeepLC[92] and compared to the measured retention times in Supplementary Fig. 5.

### Principal component analysis (PCA) using ProTIGY
The PCA analysis shown in Fig. 4 were generated using ProTIGY, v0.9.1.3. (https://github.com/broadinstitute/protigy).

## Reporting summary

Further information on research design is available in the Nature Portfolio Reporting Summary linked to this article.

## Data availability

The original mass spectra and the protein sequence database used for searches have been deposited in the public proteomics repository MassIVE (http://massive.ucsd.edu) and are accessible under the accession code MSV000090437. The published LUAD discovery dataset[14] can be found on the CPTAC program website, which details program initiatives, investigators, and datasets at https://proteomics.cancer.gov/programs/cptac. Specifically, the proteomic data can be found in the public proteomics repository MassIVE (http://massive.ucsd.edu) and are accessible under the accession code MSV000086793. The genomic data can be found at the Genomic Data Commons (https://portal.gdc.cancer.gov/) via dbGaP Study Accession phs001287.v5.p4. The analyzed LUAD discovery sample annotations, processed and normalized data files are provided as Tables S1–S3 in ref. 14. Source data are provided with this paper.

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

## Acknowledgements

Biorender was used to make multiple figure panels in this manuscript. We thank Cadence Pearce for contributions related to preliminary studies not shown in this manuscript. We thank Yo Akiyama, Qing Zhang, Francois Aguet, Yifat Geffen, and Matthew Wyczalkowski for performing somatic and germline variant calling. This work was supported in part by grants P01CA206978 to SAC, and by the following grants from the National Cancer Institute (NCI) Clinical Proteomic Tumor Analysis Consortium (CPTAC) program: U24CA270823 to S.A.C. M.A.G. and S.S., U01CA271402 to S.A.C. and M.A.G. and U24-CA271075 to D.R.M., as well as a grant from the Swiss National Science Foundation (SNF) grant CRSII5_186405 to S.A.C., and from the Dr. Miriam and Sheldon G. Adelson Medical Research Foundation to N.D.U. and S.A.C. N.D.U. is also a recipient of a SPARC Award from the Broad Institute of MIT & Harvard (#800373) that partially supported this work.

## Author contributions

J.G.A., K.D.R., M.A.G., S.S., K.R.C., N.D.U., and S.A.C. conceptualized, designed, and supervised experiments. J.G.A., E.J.B., H.B.T., K.D.R., S.K., C.X., E.K.V., J.W., H.B.W., M.V., M.E.O., J.D.A., K.P., M.H.K., and S.R. performed experiments. J.G.A., E.J.B., H.B.T., K.D.R., S.K., C.X., E.K.V., H.B.W., C.J.W., K.R.C., and N.D.U. analyzed data. D.R.M. and K.R.C. developed computational tools, performed computational analyses, and supervised the development of the data viewer. M.M. enabled computation analyses and S.A.V. implemented the data viewer. J.G.A., E.J.B., H.B.T., K.D.R., S.K., E.K.V., S.S., M.A.G., K.R.C., N.D.U., and S.A.C. participated in manuscript writing.

## Competing interests

S.A.C. is a member of the scientific advisory boards of Kymera, PTM BioLabs, Seer and PrognomIQ. The remaining authors declare no competing interests.
