## [Peer Review File · Nature Communications]

REVIEWER COMMENTS

Reviewer #1 (Remarks to the Author):

Summary

Here, authors present the serial MONTE workflow for obtaining multi-omics datasets using a single tumor specimen. The workflow serializes the established workflows of HLA profiling and Ubifast to workflows for proteomics/phosphoproteomics/acetylomics previously implemented in other studies. Authors slightly alter existing protocols by denaturing samples following HLA enrichment in SDS and digest/cleanup samples using the S-trap instead of urea/C18 cleanup. Authors perform methods development experiments and apply MONTE to profile 10 LUAD tumors, previously profiled by the authors for proteome/phosphoproteome/acetylome datasets in a CPTAC study.

As it stands, I believe this manuscript is more appropriately suited for a methods-focused journal or a proteomics-focused journal, as the depth of novel methods development, biological interpretation of datasets, and follow up on findings to inform disease biology is significantly lacking. The method itself represents a concatenation of existing published workflows with some (but not entirely rigorous) evaluation of the consequences associated with the serial handling of the samples.

Notably, the acetylome profiling differences with and without HLA profiling require substantial follow-up to clarify the source of the variation, and the higher phosphoproteome CVs also require more interrogation. The Ubifast experiment in Figure 2 should be performed using the protocol described for LUAD analyses, and with biological replicates rather than just TMT-labeled technical replicates to better assess variability in identifications, quantitation, and biological interpretation of data (including adding on the acetylation analyses to this methods development experiment). Neoantigens reported need to be more rigorously validated with synthetic peptides. Some methods used to analyze data are not reported. While authors make a few efforts to integrate several multi-omics datasets together, authors could more deeply probe the available datasets to generate a more compelling narrative for why the MONTE workflow is useful for investigating disease biology.

Specific comments:

Introduction

Line 48-51: This claim is perhaps overstated—it might be more appropriate to note that the inability to perform all of these analysis types on a single sample impedes a holistic understanding of tumor biology, not that it completely impedes our ability to understand signaling, antigen presentation, etc. As the authors point out, these things can all be analyzed using separate samples, but tumor heterogeneity may be what results in a discordant readout. Clarification in the claim is required.

Line 54-59: Authors do not cite any examples referencing their claim of 500-1000 mg, but rather here cite 4 CPTAC papers and the group's method for multiplexed proteome/phosphoproteome profiling of tumor tissue. Furthermore, authors should acknowledge more recent advancements in the sensitivity of immunopeptidomics analysis in clinical samples, such as PMID 34497125 which demonstrated HLA identification in tumor punch biopsies of a similar size to the amounts used in this study. While the depth of immunopeptidomic profiling in this publication is strong, the starting material amount is not novel.

Line 66-79: This paragraph describes the UbiFast method, which has already been published in past years. As a result, the unique challenges associated with parallelizing the method should be highlighted rather than a description of the already-published method itself.

Line 79: Citation error.

UbiFast followed by serial multi-omic sample processing shows expected coverage

Figure 2 describes the results of processing WHIM2 and WHIM16 PDXs for +/- Ubifast to compare whether data quantity/accuracy changes by incorporating Ubifast prior to proteome/phosphoproteome. The methods section of this paper describes the samples were processed as described in the author's previous papers, which states they were lysed in Urea and further processed using Sep-Pak cartridges. As the authors point out, a key difference in their workflow is using SDS as a lysis buffer and S-trap cleanup, therefore this aspect of the methods validation should mirror as closely as possible the final protocol used in the tumor analysis, including using SDS as a lysis buffer and S-trap cleanup rather than how authors previously processed the same samples. As a result, this experiment should be repeated.

Figure 2A. Size of schematic is challenging to read—authors may consider simplifying visual (ex. show yellow tube x8 instead of 8 yellow tubes at each processing step, Labeling of protein with TMT “x16” instead of drawings of 16 labeled proteins).

Line 152/Figure 2C: Without having replicates of each condition of this experimental setup, it is challenging to interpret whether the expected overlap in proteins/phosphorylated proteins between +/- Ubifast is on-par for run-to-run variation, or if there is lower overlap when Ubifast is incorporated. The authors note on line 248 that these overlaps may be expected for highly similar processing workflows, but adding Ubifast before proteome/phosphoproteome analyses may not follow the previously reported results. To address this, I recommend the experiment should be performed in triplicate, particularly to assist in understanding quantitative variation in addition to peptide IDs. To ease sample burden, instead of n=8 replicates of each PDX type, n=3 could be used increasing the total number of samples per condition to n=9. This will also help better characterize correlation of quantitation (line 157).

Phosphoproteome correlation is lower than proteome, and it is challenging to interpret whether this is expected variability or whether this is due to the addition of Ubifast. N=3 would also allow for authors to apply statistics to this analysis. For example, how many phosphosites show a significant difference in phosphorylation between subtypes in the no-Ubifast versus Ubifast analyses? Does the fraction of tyrosine phosphorylated residues decrease when adding Ubifast? These additional

experiments will be important for robustly characterizing the impact of Ubifast on downstream analyses performed in serial.

Figure 2D. Noticeably, incorporation of Ubifast does appear to result in a wider range of correlation coefficients across both basal and luminal subtypes, particularly in the phosphoproteome analyses. This should be discussed.

Figure 2G. The methods do not describe how pathway enrichment is performed (what enrichment tool, what pathways are being tested, statistical cutoffs for significance, etc). Results/significance values from these analyses should be made available in the supplemental datasets.

For the phosphorylation datasets, authors should perform a phosphorylation-specific enrichment analysis such as PSEA to most accurately evaluate whether Ubifast impacts interpretation of phosphorylation data. This is particularly critical given the lower quantitative correlation in phosphorylation data between +/- Ubifast analyses.

This section of the publication should also incorporate acetylation data to fully characterize the potential impact of Ubifast on IDs/quantitation with statistical power, which isn't possible in the tumor analysis with n=1 of each sample.

Addition of serial HLA-II and HLA-I enrichment into the MONTE workflow achieves similar data depth to analyses done in parallel & Biological signals observed in a serial multi-ome enrichment workflow without HLA enrichment are recapitulated in the MONTE workflow

Line 175: Authors should include data from this analysis such as # of peptide IDs from these samples. Do authors think the HLA expression was just too low in these samples? Is it possible to perform a western blot for HLA-I and/or HLA-II comparing expression between the breast cancer PDX samples and several of the lung cancer tissues? Does this finding have implications for using MONTE for clinical samples that do not have high HLA expression?

Line 194: Caveat should be noted that reducing tumor input material may only be applicable for samples with high HLA expression.

Line 196. HLA enrichment protocols are well established in the field and the workflows here do not represent significant deviations from other analyses of patient material that would lead to a reader expecting the length distributions or binding characteristics would be abnormal, therefore the authors may consider moving the large figures of 3C-3F to the supplement or consider keeping just 1 representative image of each in the main text.

Line 214: Do the authors have a suggestion as to how to mitigate phosphorylation losses from the serial processing workflow? Authors mention at the end that perhaps tyrosine phosphorylation analyses could be incorporated in the future, but this finding in line 214 represents a critical concern to incorporating additional phosphorylation analyses, particularly tyrosine which has such low abundance.

Line 221: The increase in acetylated peptides following the serial enrichment is a peculiar finding. In the A375 experiment, did authors perform the entire workflow (Ubifast/phosphorylation/full

proteome) profiling after 6 hour HLA enrichment prior to the acetylome experiment? Did the authors before the Class II/Class I enrichments or just incubate the sample for 6 hours? The steps in between HLA immunopurification and the acetylome profile involve a lot of sample handling, so if the A375 experiment did not include these extra steps this experiment may be more accurate if included. If this result is a difference between cell lines and tissues, this finding could be better supported by incorporating acetylome profiling into the PDX samples used in Figure 2 as previously recommended. Is it also possible this was just a sample handling or analysis error? If so, having replicate analyses would be informative, and I understand this is not possible with the LUAD tissues but may be for PDX material. As it stands, this is a significant open question in the manuscript and the authors should take additional experimental steps to understand the source of variation and rule out as many sources as possible with rigorous methods and replicates.

Figure 4B: Authors suggest the lack of clustering in the acetylome PCA analysis is likely due to the increase in the number of peptides observed when using the HLA-enriched sample. PCA analyses require a matrix without missing values, therefore the figure correctly shows that N=2552 peptides that were overlapping between analyses were used for the PCA analysis. If that is the case, a difference in IDs cannot be the reason for the lack of clustering—rather it is the quantitative variation across the analyses that is yielding the results depicted in Figure 4B. This result and interpretation should be clarified. Methods section on PCA analysis (including software used) is also missing from the methods section. Some figure 4B colors are challenging to discern—greens and red/orange/pinks. Consider using different shapes/open & closed circles/ etc. to assist in reader interpretation and improve colorblind friendliness of figure.

Line 239: Understanding protein expression of HLA machinery and associated binding partners can be critical for understanding the immune environment of the tumor. The authors should note why depleting HLA and HLA-associated machinery may be unfortunate for a multi-omics analysis and provide any suggestions for solutions either in this section or in the discussion.

Line 252: Clarify – does not introduce bias in peptide identification (quantitation not assessed in this section yet). Authors should also clarify that bias is detected in the case of the proteome with HLA machinery and associated binding partners.

HLA peptides from mutated, noncanonical, and cancer-testis antigen source proteins are identified in MONTE immunopeptidomes

Line 264: “Specific to lung cancers” –MAGEA3/MAGEA6 are not specific to lung cancers and are found in a variety of other indications including melanoma, brain, lung, and ovarian cancers. Please modify claim for accuracy.

Line 268: It would be useful if authors could list the peptides highlighted in Figure 5B in a supplemental table. The BRDT finding is interesting and offers a biological hypothesis that would be compelling to explore within this paper. Some questions that come to mind:

What alleles is the peptide expected to bind to?

Since these samples were previously profiled as part of the Gillette et al. 2020 paper, does transcript abundance of BRDT between tumor and adjacent normal tissue look different for BRDT? What about the protein expression between Normal/Tumor? This could be interesting for other CTAs identified too. Alternatively, authors could look for protein targets with differential RNA/protein expression in these patient samples in the Gillette manuscript and look to see how many have a corresponding HLA peptide identified by the DDA analyses performed here.

Are there signaling pathways enriched in any of the tumors that highlight a mechanism of action or biomarker for the presentation of any tumor antigens highlighted? Integrating the datasets available will help strengthen the rationale for performing MONTE analyses.

Line 295: What are these epitopes and which alleles are the predicted to bind to?

Line 304: These data seem to suggest suggest that these may be incorrect identifications by the spectral search algorithm if they were not identified in the WES data. What are the 2nd highest scoring sequences for these identifications and what is the score differential? Is it really appropriate to search the immunopeptidome against shared TCGA mutations when WES data is available? To highlight these epitopes in the manuscript, annotated spectrum for these ATP6AP2, SEMA3A, and FCGBP antigens should be provided in the supplement, and synthetic peptide validation should be performed to report them as mutation-containing epitopes. Alternatively, they could be excluded altogether due to the lack of support from the WES data. In addition to these and the IDUA point mutation (germline), were any other identified not mentioned in the manuscript?

Line 309/Figure 5D/Figure S2. The spectral support for the five “identified” neoantigens is not compelling in all cases. First, authors should provide annotated spectrum that span the m/z range of the acquired scan. For sample, the SAAADILLL epitope only is annotated to ~550 m/z. Authors must also include in the methods whether all ions are being presented or whether they were filtered in some way (S/N threshold, top X ions plotted only, etc). SAAADILLL has very few unannotated ions, leading to this confusion.

While the two epitopes presented in the main figure have strong support, the 3 in the supplement each have concerns. Two contain leucines, which is of course isomeric to isoleucine. With the provided data, it is not possible to distinguish between the L or I in each sequence. This is significant in the SAAADILLL epitope where the presence of the L in 8 is solely based on the low mass y2 ion, and the abundance of these ions is very low (sub 3e3) with few additional ions supporting the sequence identification (unless there are additional ions I cannot see due to the way the spectrum is cropped). Does the SpectrumMill search algorithm assign Leucine over Isoleucine when there is a score tie? How can the authors be sure the assignments are correct?

As has become commonplace in recent years, reported identified neoantigens should be validated using synthetic peptides. I would recommend analyzing the synthetic against an HLA background and providing correlation/mirror plots to better support these identifications. As it stands, particularly two of the supplemental spectrum, lack confident spectral support.

Line 367: My suggestion would be this sentiment can be excluded as the citation referenced for (for which Abelin is an author) used parallel reaction monitoring to target KRAS epitopes for enhanced sensitivity over DDA analyses, therefore I would not anticipate this experimental format would

capture all KRAS antigens. Furthermore, the paper cited did use a cell line that endogenously presented mutant KRAS to identify mutant KRAS-containing epitopes (not artificial by my definition, though authors should clarify if by artificial they mean cell line).

Line 392: “Few of the specific PTM sites presented by HLA were detected in phosphoproteomes and acetylomes” –Is it possible this is because some of the phosphorylated HLA peptide identifications are incorrectly assigned? What are the second highest ranked sequences of the phosphorylated peptides and what is the score differential? Is it also possible that the proposed issue with protease inhibitors potentially losing their potency through processing may have biased phosphoproteome identifications? Please discuss.

Line 387: This is also another area for further exploration within this dataset, particularly because it is mentioned in the discussion these data can uncover dependencies that may improve prediction (Line 436). Are any of these phosphorylated HLA peptides that were also identified as phosphopeptides map to signaling pathways that are enriched in that particular tumor? Do you see increased phosphorylation in tumor versus normal of the samples previously profiled? Same question for acetylation. Do EGFR/KRAS mutant tumors show evidence of phosphorylated (or regular) HLA peptides that match disease signatures? These data again provide more support for using the MONTE workflow, and also may highlight how biomarkers can be used to predict immune presentation. This has the potential to provide a wealth of biological insight, however many of these questions aimed at integrating the datasets remain unexplored.

The ubiquitination data is not incorporated into any of the tumor analyses outside of reporting number of identified peptides. Is there a way to use this data to better understand the biology of the samples profiled? Can the authors provide a compelling example? Otherwise, there is not strong evidence for incorporating it into the workflow, particularly when it appears to increase CVs in phosphorylation data.

Discussion

Line 403: More accurate to write “Serial HLA-II and HLA-I” profiling.

Line 408: Unless I am misunderstanding, the technical hurdle of doing enrichment of ubiquitinated peptides before TMT-labeling has already been addressed in Udeshi et al., 2020, therefore it should not be highlighted a technical hurdle overcome by the MONTE workflow. Authors should rephrase to accurately report the innovations specific to this manuscript.

Methods

Please evaluate for completeness of methods used to generate and interpret the data within this study. Several examples of missing methods are pointed out above.

Reviewer #2 (Remarks to the Author):

In this work, Abelin et al. present a MONTE workflow describing the sequential isolation of HLA peptides, and peptides modified by ubiquitin, phosphorylation, and acetylation from the same piece of tissue. The authors demonstrate that the quality of each PTM dataset is not significantly hampered by the sequential use of the LUAD lysate, and that interpretation from each PTM dataset is informative in itself, though a synergistic interpretation from all datasets was not provided, to the disappointment of this reviewer. While the data boasts high quality and reproducibility, and the lab is well-known for high technical competency, the manuscript seems not packaged well for the Nat Comm audience, and reads like a “protocol” appended with small interesting observations, in each PTM or ligandome dataset, that do not tell one story. Concatenated PTM peptide isolation and analysis is itself not new, and this reviewer has reservations about the utility and cross-interpretation of PTM datasets. Specific concerns are listed below:

1. Introduction is too brief, and far from sufficient to appreciate why serialization is critical for better biological understanding in tissue specimen. Fundamentally, sequential purification of PTMs is not new, Olsen & Mann reviewed this in *Molecular and Cellular Proteomics* already in 2013 from the technical perspective, and Olsen has another one in *Cell Systems* in 2017 describing the shotgun generation of comprehensive proteomes. The latter even highlighted deep coverage of PTMs without specific enrichment (to arrive at 7k acetylations, 10k phosphosites). What is then the real gain after this huge amount of work with serial preparations? There are many more workflows that concatenate multiple PTM purifications. Without a compelling rationale to justify this basket of PTMs that the authors want to measure together, it is not convincing why MONTE will be frequently applied by investigators to understand the underlying biology. But rather, this comes across as a mere demonstration of yes you can automate, and piggy-backed to the new UbiFast procedure the group has established and published elsewhere.

2. A large assortment of PTM purification can in theory be serialized from the same tissue. Why this particular assortment? And why in this particular order? It would be more satisfying to explain how these PTMs are connected in the biology, and why collectively studying these modifications is important to sketch the mechanism. Can you also connect PTMs on the same protein? For instance, p53 is known to be phosphorylated, ubiquitinated and acetylated, but also degraded and possibly presented as antigen peptides. Would more instances of such examples strengthen the real need for MONTE?

3. The authors listed 3 major changes in the serial preparation steps: (I) UbiFast first before other purifications, (II) add HLA-I/II purification before multi-omics, (III) SDS with S-trap instead of Urea. It is very not intuitive to list these in this order, because these do not follow the sequence of retrieval

in the presented workflow. Sodium deoxycholate might even be a better detergent to use than SDS for native pulldowns. It's easily precipitated on acid elution of HLA peptides, and almost as potent as 8M Urea lysis for proteomics. UbiFast is a good technique for the community, but that has been accepted as a different paper. Minus that, this current set of "improvement", is quite minor.

4. PTM data needs to be carefully interpreted. For instance, when comparing peptide intensities obtained from different PTM preps, it might even be invalid to conclude on occupancy of PTMs, relative proportion of modified peptides versus unmodified. And more importantly, HLA peptides are not tryptic, so these will not align with multi-omic peptides. Experimental digestion with trypsin/lysC can split the precursor of one HLA peptide into multiple tryptic peptides that ionize differently etc. Ubiquitinated peptides, though eventually will become source peptides for antigen presentation, will also be trimmed by other proteases in the loading path. Hence, again, it will be likely hard to find back the same precursor peptides to compare between PTM datasets. The cross-interpretation between different PTM datasets seems very convoluted, difficult, and even impractical to use. I envision this is going to be difficult even for highly skilled proteomics data analysts.

5. Obviously the order of retrieval would make a big difference for peptides harboring more than one PTM. This might partially explain the big variability observed in Figure 2C-D. UbiFast before phospho-enrichment obviously changes the coverage, and introduces more variation. Could phospho-degrons influence the non-overlap in Figure 2C? If the authors choose to feature this basket of PTMs, it is almost impossible not to think of phospho-degrons.

6. Then the authors feature bits of small interesting observations all jam-packed at the end of the manuscript. For instance, neoantigens, nuORFs, whether WES captures mutations, possibly using PTM abundance to improve detection of PTM HLA peptides, whether KRAS epitopes are conserved in presentation... Some of these are interesting ideas to pursue, but the presented data is not enough to make a strong point in any of these, that is a pity.

7. Overall it reads as if the authors are undecided, as to whether this manuscript should be a protocol for a multi-omics audience, or an antigen discovery audience, or potential broad biological users that read Nat Comm. In the current form, the manuscript seems most suited to go to a proteomics journal like MCP. The technical aspects of this work are excellent, but biological interpretation leaves a lot more to be desired. There is so much data in here that is undigested and not interpreted.

Reviewer #3 (Remarks to the Author):

In this manuscript, the Carr lab reports on an extension of their methods aiming at characterizing proteomes at several levels, notably PTMs and immunopeptidomes. The authors call their approach MONTE that effectively enables the serial serial enrichments/processing of HLA, ubi, ac, phospho and full proteomes from the same sample (cell lines or cancer tissue). The previously published serial methods have enjoyed success and have been taken up by many labs. The extension portrait here, promises to attract further followers. Therefore, this reviewer supports publication of the work provided that the two following issues are adequately addressed:

line 85: here and in general, please avoid giving readers the impression that MONTE does not require a lot of starting material. It may be less than what other approaches require, but 50 million cells or 200 mg of wet tissue is still a lot of material by any standards and may not be available in many clinical scenarios. Please make sure readers are aware of the limitations and provide some guidance where MONTE is likely to be successful and where not.

validation of expression of nuORFs, lincRNAs, mutated KRAS etc: the authors should comment on their confidence in the identification of such non-canonical gene products. This has been much debated in the field and adding an HLA or PTM component to this repertoire may arouse further criticism. It may be outside the scope of the current work, but it would have been desirable to see some validation that these peptides are true identifications. Labeled spectra are one way of doing this but given the availability of several spectrum prediction tools, a more systematic analysis may be warranted (at least for peptides that do not carry PTMs). Please provide some guidance to readers here so that scientists less aware of these issues are not carried away in excitement that may not always be well founded.

Workflow enabling deepscale immunopeptidome, proteome, ubiquitylome, phosphoproteome and acetylome analyses of sample-limited tissues

We thank the Editor and the Reviewers for their helpful comments and providing us with the additional time needed to perform extensive new experiments and analyses to address their concerns related to our manuscript titled “***Workflow enabling deepscale immunopeptidome, proteome, ubiquitylome, phosphoproteome and acetylome analyses of sample-limited tissues***”. We believe that we now provide a much-improved manuscript that has undergone major revisions with greatly strengthened results and conclusions. We hope that the detailed responses to Reviewers and the outline of the major revisions below address the concerns and result in a manuscript that is a good fit for publication in *Nature Communications*.

- Performed acetylome analysis of serially enriched CompRef xenograft tissue to rule out that UbiFast is biasing results obtained in downstream analyses (**Figure 2, Table S2**).
- Performed a new 16-plex TMT phosphoproteomics experiment on mouse xenograft CompRef with and without UbiFast implemented prior to phosphopeptide enrichment to define effects on the downstream phosphoproteome (**Figure S1, Table S3**).
- Purchased and analyzed synthetic peptides to confirm reported neoantigen identifications (**Figure S7**). We also provide a retention time prediction analysis of both neoantigen and nuORF HLA peptides to provide additional evidence that we believe these identifications are real (**Figure S5**).
- Provided HLA-I and HLA-II immunopeptidome results for mouse xenograft CompRef in the form of a new table to demonstrate why we further validated the MONTE workflow using LUAD tumor tissues (**Table S5**).
- Provided MONTE optimization data related to the HLA-IP and digestion that describes the extensive testing we performed that included swapping the HLA-I and HLA-II order, testing different protease inhibitors, and comparing Urea vs. S-trap vs. crash for protein digestion efficiency in new **Table S1**. The associated .raw data is also included in the MassIVE upload.
- Created and provided a publicly available multi-omic data viewer to enable readers to ask gene centric questions of the LUAD MONTE dataset:
<https://proteomics.broadapps.org/CPTAC-MONTE2022/>
- Performed a new analysis of the BRDT source protein in the proteome and transcriptome data from the Gillette *et al.*¹ where we leveraged both the proteome and transcriptome data and observed BRDT detection only in the transcriptome, suggesting that this protein is rapidly turned over prior to HLA-I presentation, making it a candidate for future immunogenicity investigations.
- Provided a new multi-omic observation of known HLA-I and HLA-II chaperones demonstrating they are not depleted after the HLA enrichment (see pg 13 below).
- Provided a new multi-omic analysis of known oncogenic and tumor suppressor proteins (**Figure S3, S4**).

- Provided a new multi-omic analysis of B2M and CD74 to better understand the HLA-I and HLA-II pathway genes in the LUAD tumor samples (**Figure S8**).

Reviewer 1:

Summary

Here, authors present the serial MONTE workflow for obtaining multi-omics datasets using a single tumor specimen. The workflow serializes the established workflows of HLA profiling and Ubifast to workflows for proteomics/phosphoproteomics/acetylomics previously implemented in other studies. Authors slightly alter existing protocols by denaturing samples following HLA enrichment in SDS and digest/cleanup samples using the S-trap instead of urea/C18 cleanup. Authors perform methods development experiments and apply MONTE to profile 10 LUAD tumors, previously profiled by the authors for proteome/phosphoproteome/acetylome datasets in a CPTAC study.

As it stands, I believe this manuscript is more appropriately suited for a methods-focused journal or a proteomics-focused journal, as the depth of novel methods development, biological interpretation of datasets, and follow up on findings to inform disease biology is significantly lacking.

- We have developed and optimized the first method that enables deepscale HLA-I and HLA-II immunopeptidomics, ubiquitylome, proteome, phosphoproteome, and acetylome data to be obtained on the exact same sample without significant compromise in the depth or reproducibility of the data for any individual ome. We have demonstrated the method using patient derived xenograft tumor tissue and patient-derived lung cancer tumor samples. In order to make serial enrichment that includes HLA immunopeptidomics a reality, multiple changes needed to be tested and optimized to establish the serial HLA immunopeptidome protocol we describe including 1) testing different protease inhibitors specific to each proteome and PTM ome to ensure no negative impact on the HLA-I and HLA-II IP 2) reversing the order of the HLA immunopeptidomes to perform the HLA-II IP first to ensure no HLA-I contamination in the HLA-II immunopeptidome data 3) implementing a mixture of HLA-DR, -DP, -DQ antibodies at a 3:1:1 mixture to mimic the levels of these different HLA-II alleles to best capture a pan HLA-II immunopeptidome 4) scaling the immunopeptidome protocol to be performed in a 96 well plate 5) reducing the HLA-I and HLA-II immunopeptidome processing time and variation in the desalting step by implementing washing samples in ½ plates and rejoining to desalt together in one desalting plate when sample sets are <40. Additionally, we tested multiple ways of digesting the proteins that remained post HLA enrichment (8M urea, chloroform/methanol precipitation, S-Trap), to determine that the S-Trap resulted in the best tryptic peptide coverage of the proteome and was the most scalable of the approaches as it can be performed in a 96 well plate format. These details are now included in the manuscript (**Table S1**), and the associated raw data has been added to the MassIVE repository.
- Papers in top tier journals describing the biological and clinical significance of HLA immunopeptidomics data alone appear regularly²⁻⁵, but these studies are accomplished using hundreds of mgs of wet weight tissue dedicated to this analysis, whereas our serialized approach that generates data for multiple omes uses just 50 mg wet weight

tissue total for input for HLA-I, HLA-II and all downstream omics. The value of obtaining HLA immunopeptidomics data in serial on the same samples used for multiomics studies is clear, and this is the first report describing in detail how to accomplish it. We have also further strengthened the discussion of the essentiality of obtaining as much high quality, deep “omics” information as is possible from each single precious human tissue sample.

- We agree that biological interpretation of the data was limited, a direct result of the small LUAD sample set that was available to us. We now extend these biological insights with specific analyses outlined in the responses below to address the specific concerns of the Reviewers. For example, we have provided several new analyses that demonstrate having immunopeptidome, proteome, and PTMome datasets can help to answer specific biological questions, such as the expression of BRDT that we believe may lead to tumor associated antigens for future follow-up, and multi-omic analyses in the new Supplemental **Figures S3, S4 and S8** that look at oncogenic and tumor suppressor proteins as well as HLA-I and HLA-II pathway proteins B2M and CD74. Importantly, we now also provide readers with a multi-omic data visualization tool to enable gene centric searches across all the -omes profiled with MONTE and use the tool to generate some of the results newly described in the paper: <https://proteomics.broadapps.org/CPTAC-MONTE2022/>
- We have added text to the Discussion (pg 13) describing the limitations of our study. Although we have demonstrated the feasibility and utility of a workflow incorporating HLA-I and -II immunopeptidomics and UbiFast ubiquitylproteomics into a serialized proteomic workflow using a clinically relevant sample set, its pilot-level scale precludes the statistically robust analyses, deep explorations of biology, or compelling assessments of the interplay between characterized -omes that the approach is intended to facilitate. Rather than highlighting such underpowered and speculative results, we therefore chose to focus on the added value and interpretable results provided by immunopeptidomic characterization of tumor samples. Recent large-scale cancer proteogenomics analyses have already made a compelling case that the integration of proteomic, ubiquitylproteomic and especially phosphoproteomic data with genomic data helps to functionalize genomic aberrations, providing new perspectives on cancer biology and nominating potential therapeutic vulnerabilities⁶. We agree with the reviewer that it remains to be shown that by application to a suitably sized patient cohort, that the integration of additional layers of data, such as the immunopeptidome, will continue to provide new, interpretable, and actionable insights. But these studies are beyond the scope of the present manuscript, which is a proof of principle of the method using relevant samples.

The method itself represents a concatenation of existing published workflows with some (but not entirely rigorous) evaluation of the consequences associated with the serial handling of the samples.

Notably, the acetylome profiling differences with and without HLA profiling require substantial follow-up to clarify the source of the variation, and the higher phosphoproteome CVs also require more interrogation.

- As addressed above, the MONTE method does not represent a simple concatenation of existing methods, and these method development data are now included in the revision (new **Table S1**).
- Related to the acetylome profiling differences- we provided supplemental data (**Table S6**) using A375 cells that demonstrates that adding the serial HLA-I and HLA-II immunoprecipitations prior to acetylome analysis does not alter the results obtained in acetylome profiling.
- We performed a new acetylome analysis of serially enriched CompRef xenograft tissue to rule out that UbiFast is biasing results obtained in downstream analyses (**Figure 2**, **Table S2**).
- It should also be noted that MONTE improved recovery of acetyllysine containing peptides and demonstrated that the increase is not due to non-enzymatic acetylation using new experimental data (pg 6). We speculate that the increased yield may be due to pre-clearing of non-specifically binding components in the complex tissue lysates by HLA- and K-ε-GG antibodies.

The Ubifast experiment in Figure 2 should be performed using the protocol described for LUAD analyses, and with biological replicates rather than just TMT-labeled technical replicates to better assess variability in identifications, quantitation, and biological interpretation of data (including adding on the acetylation analyses to this methods development experiment).

- We thank the Reviewer for pointing out that we made an error in citing previous work in the Methods. We indeed used the SDS based S-Trap protocol for the data shown in **Figure 2** and have updated the manuscript to reflect this.

Neoantigens reported need to be more rigorously validated with synthetic peptides. Some methods used to analyze data are not reported.

- Synthetic peptides corresponding to the five neoantigens identified in the LUAD dataset were purchased and used for validation. The reported neoantigens had high correlation with their synthetic peptide counterparts. Mirror plots have been added to **Figure S7** to demonstrate this new analysis. We also performed retention time prediction using DeepLC⁷ in new **Figure S5** for these neoantigens, which demonstrated their predicted retention times correlated well with the observed retention time.

While authors make a few efforts to integrate several multi-omics datasets together, authors could more deeply probe the available datasets to generate a more compelling narrative for why the MONTE workflow is useful for investigating disease biology.

- Given the limitation on the LUAD cohort size, we are not able to do a meaningful deep-scale multi-omic analysis. Therefore, we provided new analyses that demonstrate having immunopeptidome, proteome, and PTMome datasets can help to answer specific biological questions, such as the expression of BRDT that we believe may lead to tumor associated antigens for future follow-up, and multi-omic analyses in the new Supplemental **Figures S3, S4** and **S8** that look at oncogenic and tumor suppressor proteins as well as HLA-I and HLA-II pathway proteins B2M and CD74. However, even in the absence of more extensive demonstration of benefit using the small,

representative sample set analyzed, there is substantial literature on use of HLA peptidomics, proteomics, phosphopeptidomics, etc. alone or in some combination that makes it clear that being able to generate deepscale, high quality data on all of the omes has significant likelihood of being of value to provide new, previously unappreciated aspects of tumor biology. We show the added benefit in the context of HLA peptidomics for the 10 LUAD tumors.

Specific comments:

Introduction

Line 48-51: This claim is perhaps overstated—it might be more appropriate to note that the inability to perform all of these analysis types on a single sample impedes a holistic understanding of tumor biology, not that it completely impedes our ability to understand signaling, antigen presentation, etc. As the authors point out, these things can all be analyzed using separate samples, but tumor heterogeneity may be what results in a discordant readout. Clarification in the claim is required.

- We agree with the reviewer – the word holistic is key and have added it to the last sentence of the first paragraph of the introduction (pg 2). The introduction has been extensively rewritten to clarify that the MONTE protocol is best suited for samples where all analyses could not be done in parallel for samples with limited input amounts. The MONTE workflow enables serial HLA-I and HLA-II immunopeptide, proteome, and PTMome information out of a single sample, which has not been demonstrated previously.

Line 54-59: Authors do not cite any examples referencing their claim of 500-1000 mg, but rather here cite 4 CPTAC papers and the group's method for multiplexed proteome/phosphoproteome profiling of tumor tissue. Furthermore, authors should acknowledge more recent advancements in the sensitivity of immunopeptidomics analysis in clinical samples, such as PMID 34497125 which demonstrated HLA identification in tumor punch biopsies of a similar size to the amounts used in this study. While the depth of immunopeptidomic profiling in this publication is strong, the starting material amount is not novel.

- Additional immunopeptide input references have now been added (see below), and the immunopeptide input statement has been adjusted to reflect that it is specific to serial HLA-I and HLA-II immunopeptidomics.
Pg 2- *“For immunopeptidomics workflows that attempt to directly identify neoantigens, a separate aliquot of tissue, usually 500 to 1000 milligrams of wet weight tissue or up to 1 billion cells 2,6,24,26, is needed compared to 25-50 milligrams for serial, multiplexed proteomics, phospho-, and acetyl-peptidomics15–18,21. Moreover, sample preparation for immunopeptidomics is distinct from that used for conventional proteomics: immunopurification (IP) of HLA molecules requires the use of native lysis buffer containing mild detergent to maintain protein conformations and solubilize membrane-bound HLA proteins. In contrast, current serial proteome and PTM-ome enrichment*

protocols denature proteins using urea or SDS prior to tryptic digestion preventing upstream HLA peptide complex enrichment.”

- We believe that the immunopeptidome depth using 50mg tumor (~2 mg protein) is novel. As this Reviewer points out, PMID 34497125 used tumor punch biopsies with 1-5mg of total protein and reported in **Table S4** of this manuscript 254-5038 HLA-I peptides (median 1316) using the same highly sensitive mass spectrometer (Exploris480) as the MONTE data was collected. In comparison, we showed a median of 11,387 HLA-I (8,278-13,727) and 5,263 HLA-II (1,123-9,726) peptides from each of these ten LUAD tumors. Furthermore, the MONTE immunopeptidomes include both HLA-I and HLA-II from the exact same sample, which has not been routinely used for tumor/tissue input amounts of this size.

Line 66-79: This paragraph describes the UbiFast method, which has already been published in past years. As a result, the unique challenges associated with parallelizing the method should be highlighted rather than a description of the already-published method itself.

- We agree with the reviewer. Lines 66-79 have been removed to better focus the manuscript on the serialization of UbiFast with the proteome, phosphoproteome, and acetylome and the resulting impact on downstream data.

Line 79: Citation error.

- This citation has been removed during revision of the introduction.

UbiFast followed by serial multi-omic sample processing shows expected coverage Figure 2 describes the results of processing WHIM2 and WHIM16 PDXs for +/- Ubifast to compare whether data quantity/accuracy changes by incorporating Ubifast prior to proteome/phosphoproteome. The methods section of this paper describes the samples were processed as described in the author’s previous papers, which states they were lysed in Urea and further processed using Sep-Pak cartridges. As the authors point out, a key difference in their workflow is using SDS as a lysis buffer and S-trap cleanup, therefore this aspect of the methods validation should mirror as closely as possible the final protocol used in the tumor analysis, including using SDS as a lysis buffer and S-trap cleanup rather than how authors previously processed the same samples. As a result, this experiment should be repeated.

- We thank the reviewer for identifying an error in our Methods section. This PDX experiment was completed using the same SDS based lysis method as the LUAD samples, and the text in the Methods subsection on page 16 “*Serial Processing and Analysis of Proteome, and Phosphoproteome and Acetylome from UbiFast flow-through samples of Comparative Reference Tissue(CompRef)*” on pg 13 has been updated.

Figure 2A. Size of schematic is challenging to read—authors may consider simplifying visual (ex. show yellow tube x8 instead of 8 yellow tubes at each processing step, Labeling of protein with TMT “x16” instead of drawings of 16 labeled proteins.

- We thank the Reviewer for this suggestion and agree. **Figure 2A** has been updated to improve viewing and clarity.

Line 152/Figure 2C: Without having replicates of each condition of this experimental setup, it is challenging to interpret whether the expected overlap in proteins/phosphorylated proteins between +/- Ubifast is on-par for run-to-run variation, or if there is lower overlap when Ubifast is incorporated. The authors note on line 248 that these overlaps may be expected for highly similar processing workflows, but adding Ubifast before proteome/phosphoproteome analyses may not follow the previously reported results. To address this, I recommend the experiment should be performed in triplicate, particularly to assist in understanding quantitative variation in addition to peptide IDs. To ease sample burden, instead of n=8 replicates of each PDX type, n=3 could be used increasing the total number of samples per condition to n=9. This will also help better characterize correlation of quantitation (line 157).

- We thank the reviewer for this helpful suggestion. Using the same WHIM2 and WHIM16 PDX models, we completed an additional TMT phosphoproteomics experiment with and without UbiFast implemented prior to phosphopeptide enrichment. Specifically, all phosphopeptide enriched samples were measured in a single TMTpro16 plex (n=4 replicates per condition) to measure quantitative differences on exactly the same phosphopeptides between +/- UbiFast samples. A new figure (**Figure S1**) was added to the manuscript and summarizes this new experiment. We identified 38,193 human phosphorylation sites of which only 3.8% showed significantly reduced intensity (> 2-fold

and $p\text{val} < 0.05$) in samples where UbiFast was incorporated in the workflow. We have added a table summarizing this information in **Figure S1B**. Plots of peptide length and measured HPLC retention time for depleted phosphopeptides compared to all phosphopeptides have been added in **Figure S1C**. The majority of the depleted phosphopeptides are short and hydrophilic, and are likely depleted by the extra desalting step following UbiFast processing and prior to TMT labeling and phosphopeptide enrichment. This experiment shows that UbiFast does not significantly affect replicate correlation (**Figure S1B**) or unsupervised hierarchical clustering of phosphoproteome data by breast cancer subtype (**Figure S1D**). We have added an additional paragraph to the document summarizing the findings from the experiment shown in **Figure S1** on pg 5

Phosphoproteome correlation is lower than proteome, and it is challenging to interpret whether this is expected variability or whether this is due to the addition of UbiFast. $N=3$ would also allow for authors to apply statistics to this analysis. For example, how many phosphosites show a significant difference in phosphorylation between subtypes in the no-UbiFast versus UbiFast analyses? Does the fraction of tyrosine phosphorylated residues decrease when adding UbiFast? These additional experiments will be important for robustly characterizing the impact of UbiFast on downstream analyses performed in serial.

- As suggested by the reviewer, we carried out additional analysis of the data shown in **Figure 2** to address if UbiFast is specifically impacting pS, pT and pY detection in the downstream IMAC enrichment. We found no change in the fraction of pS, pT or pY in IMAC samples acquired with or without UbiFast in serial. Without UbiFast pS, pT and pY accounted for 83%, 16% and 1%, respectively. With UbiFast in serial, pS, pT and pY accounted for 83%, 16% and 1%, respectively.
- We have also carried out an entirely new experiment as shown in **Figure S1**. As described in the above response, the main difference observed in the phosphoproteome when UbiFast is incorporated in the workflow is a loss of a small percent (3.8%) of relatively short hydrophilic peptides. We have added additional text to the document summarizing the findings from the experiment shown in **Figure S1** on pg 5.

Figure 2D. Noticeably, incorporation of UbiFast does appear to result in a wider range of correlation coefficients across both basal and luminal subtypes, particularly in the phosphoproteome analyses. This should be discussed.

- The correlations are similar for samples where UbiFast has been incorporated serially. The correlation boxplots in **Figure 2D** were removed and the values were added to the table in **Figure 2B** to more clearly note the median the correlation coefficients across the basal and luminal subtypes in this analysis.
- As described above, we have also carried out an entirely new experiment as outlined in **Figure S1**. We have added Pearson correlation coefficients to **Figure S1D**. These results show no significant difference between the correlation of IMAC enriched phosphopeptides with and without UbiFast in the workflow.

Figure 2G. The methods do not describe how pathway enrichment is performed (what enrichment tool, what pathways are being tested, statistical cutoffs for significance, etc).

Results/significance values from these analyses should be made available in the supplemental datasets.

- The Methods have been expanded to describe how this pathway enrichment was performed. Text has been added to the manuscript on page 18 in the subsection CompRef PDX data analysis: “Single sample Gene Set Enrichment Analysis (ssGSEA)¹¹ and site-centric PTM Signature Enrichment Analysis (PTM-SEA) were performed as described in <https://github.com/broadinstitute/ssGSEA2.0>. Proteins, phosphorylation sites and acetylation sites were enriched using standard methods as previously described in Subramanian, et al. 2005. The C5 GO biological process gene set database from MSigDB was used for enrichment.” GSEA for proteome, phosphoproteome and acetylome as well as PTM-SEA results have been added to **Table S2**.

For the phosphorylation datasets, authors should perform a phosphorylation-specific enrichment analysis such as PSEA to most accurately evaluate whether UbiFast impacts interpretation of phosphorylation data. This is particularly critical given the lower quantitative correlation in phosphorylation data between +/- UbiFast analyses.

- As suggested by the reviewer, PTM-SEA¹² was performed to evaluate if UbiFast impacts the interpretation of phosphorylation data related to **Figure 2**. We found that the correlation was similar using either ssGSEA¹¹ or PTM-SEA. Text has been added to the manuscript on pg 5: “Site-centric PTM Signature Enrichment Analysis (PTM-SEA) was also performed on regulated phosphorylation sites and the top gene sets showed the same trends (**Table S2**).

This section of the publication should also incorporate acetylation data to fully characterize the potential impact of UbiFast on IDs/quantitation with statistical power, which isn't possible in the tumor analysis with n=1 of each sample.

- As suggested by the reviewer, we completed a new acetylome enrichment experiment for PDX samples shown in **Figure 2**. Antibody-based acetylation enrichment was completed using saved flow-through samples from the IMAC enrichment step and acetylome results have been incorporated into **Figure 2** and text has been added to the manuscript on pg 5. Similar to what we observed for the proteome and phosphoproteome data acquired from UbiFast flow-throughs, acetylome correlation values were high with median correlations of 0.84 and 0.83 for Basal and Luminal subtypes indicating that UbiFast pre-processing does not negatively affect reproducibility.

Addition of serial HLA-II and HLA-I enrichment into the MONTE workflow achieves similar data depth to analyses done in parallel & Biological signals observed in a serial multi-ome enrichment workflow without HLA enrichment are recapitulated in the MONTE workflow

- Although parallel immunopeptidome analyses are possible for some samples, not all primary tumor or patient tissue samples have enough material to do such an analysis- especially if multiomics analyses beyond immunopeptidomics is to be conducted. Therefore, we suggest MONTE as a path forward to collect multi-omic data that prevents

having to split the sample for different omic analyses, which would result in lower overall immunopeptidome depth. To address this concern, we have rewritten the introduction and discussion to better reflect this key advantage of the workflow.

Line 175: Authors should include data from this analysis such as # of peptide IDs from these samples. Do authors think the HLA expression was just too low in these samples? Is it possible to perform a western blot for HLA-I and/or HLA-II comparing expression between the breast cancer PDX samples and several of the lung cancer tissues? Does this finding have implications for using MONTE for clinical samples that do not have high HLA expression?

- HLA-I and HLA-II immunopeptidome data for the PDX xenograft CompRef samples has been added in **Table S5**. The depth of the HLA-I data (283-539 peptides) and HLA-II data (665-2651) were not deep enough to identify neoantigens.
- Low HLA expression would likely prevent neoantigen identification, therefore, we state in the manuscript lines 317-320 those current analyses focused on neoantigen ID should be focused on tumors with high HLA expression.
- It is not possible to perform a Western blot comparing HLA-I and HLA-II expression between the PDX and LUAD samples, as no sample remains. The newly presented immunopeptidome data can be used as a proxy for HLA expression showing low (PDX) vs. high (LUAD).
- If HLA expression is not known, performing the immunopeptidome analysis would not prevent the collection of downstream multi-omic data, therefore, including immunopeptidomics in the workflow may prove to be useful when combined with the other omics data. For example, if HLA expression is low, the resulting immunopeptidome data demonstrates which alleles are expressing peptides. Allele specific HLA expression data is useful biological information that is difficult to obtain from proteome and transcriptome data due to the highly polymorphic nature of HLA alleles. We have added these points to the discussion on pg 12.

“Although we demonstrate the usefulness of MONTE in a small LUAD cohort that expressed both HLA-I and HLA-II and where at least 50mg of cryopulverized tissue was available, this workflow can be extended to other tumor types with less available tissue, low HLA expression, and no HLA-II expression. In these scenarios, performing the HLA serial enrichments will likely result in lower immunopeptidome depth, but will not prevent the downstream multi-omic analyses, as these require less input material (25 mg wet weight or less) than the HLA enrichment. Even in cases where the HLA expression in tumors is low or knocked-down, useful information such as which HLA alleles are expressed and presenting peptides can be directly determined from these immunopeptidomes and leveraged to better understand changes in tumor HLA peptide presentation.”

Line 194: Caveat should be noted that reducing tumor input material may only be applicable for samples with high HLA expression.

- Text has been updated to clarify this claim is specific to tumors with similar HLA expression levels as follows:

pg 9- *“These results suggest that detection of neoantigens by immunopeptidomics should, at present, be focused on tumor types with relatively high mutational burden, high HLA expression levels, and only the most highly optimized LC-MS/MS methods should be used.”*

Line 196. HLA enrichment protocols are well established in the field and the workflows here do not represent significant deviations from other analyses of patient material that would lead to a reader expecting the length distributions or binding characteristics would be abnormal, therefore the authors may consider moving the large figures of 3C-3F to the supplement or consider keeping just 1 representative image of each in the main text.

- Although immunopeptidomics has been leveraged since the early 1990's, the protocols used are not standardized. We now describe and provide our optimization data to support the multiple changes to established HLA immunopeptidome protocols that we have made that include 1) testing different protease inhibitors specific to each proteome and PTM ome to ensure no strong impacts on the HLA-I and HLA-II IP 2) reversing the order of the HLA immunopeptidomes to perform the HLA-II IP first to ensure no HLA-I contamination in the data 3) scaling the immunopeptidome protocol to be performed in a 96 well plate 4) reducing the HLA-I and HLA-II immunopeptidome processing time by implementing washing samples in ½ plates and rejoining to desalt together in one desalting plate that is applicable for sample sets <40. Additionally, we tested multiple ways of digesting the proteins that remained post HLA enrichment (8M urea, chloroform/methanol precipitation, S-Trap), to determine that the S-Trap resulted in the best tryptic peptide coverage of the proteome and was the most scalable of the approaches as it can be performed in a 96 well plate format. A summary of these optimizations is now referenced in the manuscript (pg 4) and **Table S1**.

Line 214: Do the authors have a suggestion as to how to mitigate phosphorylation losses from the serial processing workflow? Authors mention at the end that perhaps tyrosine phosphorylation analyses could be incorporated in the future, but this finding in line 214 represents a critical concern to incorporating additional phosphorylation analyses, particularly tyrosine which has such low abundance.

- The text has been updated (see below) to suggest that spiking in more phosphatase inhibitors during the second HLA-I IP may help prevent the phosphorylation losses, as each HLA-I and HLA-II IP are each performed for 3hrs at 4°C for a total of 6hrs, likely reducing effectiveness of the phosphatase inhibitors.
- Although not shown because it is out of the scope of this current manuscript, we have performed a head-to-head experiment using a set of ovarian tumors where we performed the MONTE workflow with the additional spike of phosphatase inhibitors vs. the serial enrichment workflow without HLA enrichment and noted only an 3% decrease in the total number of recovered phosphosites (n=26,692) when compared to tissue that was not HLA depleted (n=26,335). This experiment demonstrated to us that we were able to resolve this issue with the additional spike of phosphatase inhibitors during the serial HLA-II and HLA-I enrichments. In addition, spiking in addition inhibitors did not

significantly impact the upstream ubiquityl-enrichment step (9,244 in no HLA IP vs. 8,900 in HLA IP FT).

- pg 6- *“The proteome and ubiquitylome results demonstrate that similar numbers of canonical human proteins (11,028 vs. 10,729) and K-ε-GG peptides (9,516 vs. 9,419) were identified and fully quantified between the non-HLA enriched (“No HLA”) and HLA enriched (“HLA FT”) samples, respectively. A 16% decrease in the total number of phospho-sites (-8% phosphorylated proteins) was observed when using the HLA enriched samples (No HLA: 26,627 phosphosites, 6,745 phosphoproteins; HLA FT: 22,339 phosphosites, 6,235 phosphoproteins), suggesting that the phosphatase inhibitors added to our lysis buffer may be losing their activity during the protein-level, HLA immunopeptidome enrichment. Therefore, we suggest spiking in another set of phosphatase inhibitors at the start of the HLA-I IP to ensure their activity remains during the 6hr serial IP process.”*

Line 221: The increase in acetylated peptides following the serial enrichment is a peculiar finding. In the A375 experiment, did authors perform the entire workflow (UbiFast/phosphorylation/full proteome) profiling after 6 hour HLA enrichment prior to the acetylome experiment? Did the authors before the Class II/Class I enrichments or just incubate the sample for 6 hours? The steps in between HLA immunopurification and the acetylome profile involve a lot of sample handling, so if the A375 experiment did not include these extra steps this experiment may be more accurate if included. If this result is a difference between cell lines and tissues, this finding could be better supported by incorporating acetylome profiling into the PDX samples used in Figure 2 as previously recommended. Is it also possible this was just a sample handling or analysis error? If so, having replicate analyses would be informative, and I understand this is not possible with the LUAD tissues but may be for PDX material. As it stands, this is a significant open question in the manuscript and the authors should take additional experimental steps to understand the source of variation and rule out as many sources as possible with rigorous methods and replicates.

- The A375 experiment reported in **Table S6** (samples 1 and 2) looked at only the impact of including the HLA-I and HLA-II purification prior to the acetylome experiment to ensure that step was not causing the increase.
- As suggested by the reviewer, we completed a new acetylome enrichment experiment for these samples as shown in **Figure 2, Table S2**. Antibody-based acetylation enrichment was completed using saved flow-through samples from the IMAC enrichment step and acetylome results have been incorporated into **Figure 2** and text has been added to the manuscript on pgs. 4-5. Similar to what we observed for the proteome and phosphoproteome data acquired from UbiFast flow-throughs, acetylome correlation values were high with median correlations of 0.84 and 0.83 for Basal and Luminal subtypes indicating that UbiFast pre-processing does not negatively affect reproducibility.

Figure 4B: Authors suggest the lack of clustering in the acetylome PCA analysis is likely due to the increase in the number of peptides observed when using the HLA-enriched sample. PCA analyses require a matrix without missing values, therefore the figure correctly shows that

N=2552 peptides that were overlapping between analyses were used for the PCA analysis. If that is the case, a difference in IDs cannot be the reason for the lack of clustering—rather it is the quantitative variation across the analyses that is yielding the results depicted in Figure 4B. This result and interpretation should be clarified.

- The text was adjusted to address this concern. We agree that the acetylome enrichment efficiency of the HLA enriched samples is better than the non HLA enriched sample. Therefore, the abundances of these sites may be different and could impact the clustering of overlapping sites. Text has been modified as follows:

pg 7- *“The acetylomes of HLA-enriched and non-HLA-enriched samples were somewhat less well correlated.”*

Methods section on PCA analysis (including software used) is also missing from the methods section. Some figure 4B colors are challenging to discern—greens and red/orange/pinks. Consider using different shapes/open & closed circles/ etc. to assist in reader interpretation and improve colorblind friendliness of figure.

- The Methods section was updated to include a citation to the analysis tool used for PCA analysis.

Pgs 28-29- *“The PCA analyses shown in Figure 4 were generated using ProTIGY, v0.9.1.3. (<https://github.com/broadinstitute/protigy>).”*

- The colors in **Figure 4B** were updated and half of the samples now include outline colors to help differentiate the samples in the PCA plot.

Line 239: Understanding protein expression of HLA machinery and associated binding partners can be critical for understanding the immune environment of the tumor. The authors should note why depleting HLA and HLA-associated machinery may be unfortunate for a multi-omics analysis and provide any suggestions for solutions either in this section or in the discussion.

Line 252: Clarify – does not introduce bias in peptide identification (quantitation not assessed in this section yet). Authors should also clarify that bias is detected in the case of the proteome with HLA machinery and associated binding partners.

- We performed a new analysis to look for depletions in known HLA-I and HLA-II chaperone proteins (generated using the provided data viewer: <https://proteomics.broadapps.org/CPTAC-MONTE2022/>). We did not observe global depletion in the proteome of HLA-I chaperones CALR, CANX, or TAPBR. We also did not observe global depletion in the proteome of HLA-II chaperones HLA-DM and HLA-DO. These observations are reported in the revised manuscript. We chose not to show the heatmap below because we observed a null result and provide readers with access to the data viewer that enables them to visualize these data as shown below.
- pg 7- *“We looked into the proteome data for depletions in known HLA-I and HLA-II chaperone proteins to confirm our serial HLA-II and HLA-I immunopurification is not depleting known HLA protein binding partners. We did not observe proteome depletion of HLA-I chaperones CALR, CANX, or TAPBR or HLA-II chaperones HLA-DM and HLA-*

DO. These observations suggest the addition of the serial HLA immunopurification does not have a significant negative impact on the downstream analysis.”

HLA peptides from mutated, noncanonical, and cancer-testis antigen source proteins are identified in MONTE immunopeptidomes Line 264: “Specific to lung cancers” – MAGEA3/MAGEA6 are not specific to lung cancers and are found in a variety of other indications including melanoma, brain, lung, and ovarian cancers. Please modify claim for accuracy.

- Text was updated (see below) to clarify that this citation refers to a previous lung cancer study, not that these TAAs are only found in lung cancer.
- pg 8- “Across the set of LUAD tumors, peptides derived from seven CTA source proteins previously reported in lung cancer¹³ were detected, including two from the MAGE family(Figure 5B).”

Line 268: It would be useful if authors could list the peptides highlighted in Figure 5B in a supplemental table. The BRDT finding is interesting and offers a biological hypothesis that would be compelling to explore within this paper. Some questions that come to mind: What alleles is the peptide expected to bind to?

- **Figure 5B** peptides are listed in **Table S9**, tab “HLAI,II_Lung_CTantigens”.
- HL Athena predictions have been added for all HLA-I peptides 8-11 AA long.

Since these samples were previously profiled as part of the Gillette et al. 2020 paper, does transcript abundance of BRDT between tumor and adjacent normal tissue look different for BRDT? What about the protein expression between Normal/Tumor? This could be interesting for other CTAs identified too. Alternatively, authors could look for protein targets with differential RNA/protein expression in these patient samples in the Gillette manuscript and look to see how many have a corresponding HLA peptide identified by the DDA analyses performed here.

- A new analysis of the BRDT source protein in the proteome and transcriptome data in the Gillette et al., Cell, 2020 paper was performed as suggested, and the following text added to the paper:
- pg 8- *“To confirm BRDT protein expression, we leveraged both the proteome and transcriptome data and observed BRDT detection only in the transcriptome, suggesting that this protein is rapidly turned over prior to HLA-I presentation, making it a candidate for future immunogenicity investigations.”*

Are there signaling pathways enriched in any of the tumors that highlight a mechanism of action or biomarker for the presentation of any tumor antigens highlighted? Integrating the datasets available will help strengthen the rationale for performing MONTE analyses.

- To address this question, we have carried out additional analyses as shown in **Figure S3, S4** to look at EGFR, KRAS, RB1, and STK11 across the different omes. We have also created a publicly accessible data viewer that facilitates exploration of the data by interested readers (<https://proteomics.broadapps.org/CPTAC-MONTE2022/>). We have added the following new paragraph to main text describing these results:
- pg 7- *“We next investigated known oncogenic and tumor suppressor proteins. Oncogenes EGFR and KRAS and tumor suppressor genes RB1 and STK11 were detected across multiple -omes, with similar patterns of protein and PTM site levels observed in both HLA enriched and the non enriched samples (**Figure S3**; also available using data viewer <https://proteomics.broadapps.org/CPTAC-MONTE2022/>). For example, patient C3N00199 showed the highest level of EGFR ubiquitinylation and patient C3N00547 had the highest level of RB1 phosphorylation in both datasets. The tumor suppressor protein TP53 had variable detection (7/10) in the discovery dataset (Gillette et al. 2020), which may have led to the lack of detection in both the HLA enriched and non-enriched TMTplexes (**Figure S4**; also available using data viewer). We also observed HLA-I peptides from wild-type EGFR (8/10), KRAS (3/10), RB1(10/10), TP53 (9/10) and STK11(1/10) across the LUAD patient cohort (Table S8). Conversely, HLA-II peptides within expected nested sets were only detected from EGFR (7/10), which is endocytosed upon activation allowing it entry to the HLA-II processing and presentation pathway. No clear trends between HLA-I and HLA-II peptide presentation and driver mutation status were detectable in this set of oncogenes and tumor suppressors. The detection of these oncogenic and tumor suppressor proteins*

across multiple -omes from samples that underwent HLA enrichment demonstrates that known biological signals can be recovered using the MONTE workflow.”

Line 295: What are these epitopes and which alleles are the predicted to bind to?

- HL Athena predictions were added for all HLA-I peptides 8-11 AA long in **Table S9**.

Line 304: These data seem to suggest that these may be incorrect identifications by the spectral search algorithm if they were not identified in the WES data. What are the 2nd highest scoring sequences for these identifications and what is the score differential? Is it really appropriate to search the immunopeptidome against shared TCGA mutations when WES data is available? In addition to these and the IDUA point mutation (germline), were any other identified not mentioned in the manuscript?

- A new analysis of the WES data was performed to determine the utility of including the TCGA shared mutations. Interestingly, the three TCGA shared mutations from MUC6, ADAMTSL4, and BEST3 were indels where a single nucleotide insertion produced a frameshift not called in the WES data of any patient and the HLA-I peptides we observed were from the downstream frameshifted region of the truncated protein. The shared TCGA mutations from ATP6AP2 and, SEMA3A, and FCGBP were not called in the WES data of any patient by the WES data and will require further validation before they can be confirmed as putative neoantigens. In all cases, the MS/MS spectra were of sufficiently high quality that mis-identification of the peptide is unlikely. Therefore, we believe that these indels and mutations may have been missed by the mutation calling pipeline. Regardless, we have decided to remove this section (see below) for simplicity, as it is difficult to justify including these peptides without the support of the mutation calls.
- The following text has been removed from the manuscript: *“We first mined our data for HLA peptides containing mutations shared across multiple tumor types reported in TCGA41. Six of the ten patients had at least one HLA-I peptide mapping to a shared TCGA mutation, while no HLA-II peptides containing TCGA mutations were identified. The most frequently presented TCGA mutation was a point mutation (H33Q) in IDUA, an enzyme found in lysosomes, that was presented by three of the ten patients on either HLA-B*55:02 or -B*37:01. We also observed HLA-I peptides containing TCGA mutations presented by at least two of the patients from SEMA3A (point mutation) and FCGBP (point mutation). By comparing published whole exome sequencing (WES) data from both blood (normal) and tumor from these LUAD patients, we determined that the IDUA mutation (10/10 patients) was germline and not a neoantigen. The peptide from FCGBP lacks a mutation and is instead derived from a region of FCGBP that is not part of the reference proteome isoforms of FCGBP, but is included in the TCGA isoform. Interestingly, the three several of the other TCGA shared mutations from MUC6, ADAMTSL4, and BEST3 were indels where a single nucleotide insertion produced a frameshift not called in the the WES data of any patient and the found in HLA-I peptides we observed were from the downstream frameshifted region of the truncated protein. The shared TCGA mutations from ATP6AP2 and, SEMA3A, and FCGBP were not called in the WES data of any patient by the WES data and will require further validation before they can be confirmed as putative neoantigens. In all cases, the MS/MS spectra are of*

sufficiently high quality that mis-identification of the peptide is unlikely (Supplemental Table 3)."

- The 2nd highest scoring sequences for HLA peptide identifications and score differential are reported in **Table S9**.

Line 309/Figure 5D/Figure S2. The spectral support for the five "identified" neoantigens is not compelling in all cases. First, authors should provide annotated spectrum that span the m/z range of the acquired scan. For sample, the SAAADILLL epitope only is annotated to ~550 m/z. Authors must also include in the methods whether all ions are being presented or whether they were filtered in some way (S/N threshold, top X ions plotted only, etc). SAAADILLL has very few unannotated ions, leading to this confusion.

While the two epitopes presented in the main figure have strong support, the 3 in the supplement each have concerns. Two contain leucines, which is of course isomeric to isoleucine. With the provided data, it is not possible to distinguish between the L or I in each sequence. This is significant in the SAAADILLL epitope where the presence of the L in 8 is solely based on the low mass y2 ion, and the abundance of these ions is very low (sub 3e3) with few additional ions supporting the sequence identification (unless there are additional ions I cannot see due to the way the spectrum is cropped). Does the SpectrumMill search algorithm assign Leucine over Isoleucine when there is a score tie? How can the authors be sure the assignments are correct? As has become commonplace in recent years, reported identified neoantigens should be validated using synthetic peptides. I would recommend analyzing the synthetic against an HLA background and providing correlation/mirror plots to better support these identifications. As it stands, particularly two of the supplemental spectrum, lack confident spectral support.

- Synthetic peptides were ordered, analyzed and now used to validate the neoantigens we claimed to have detected. Mirror plots for each are shown in new **Fig S7** for each. The legend for **Fig S7** has been updated to include the following:
 - The depicted spectra have no peak filtering applied.
 - The SAAADHILL spectrum cropping did not hide any ions. No higher mass ions were detected. The mass range shown facilitates visible labels on all peaks. In Spectrum Mill, Leu/Ile score equally and tie-scoring sequences if present, would be reported.
 - We also performed a retention time prediction analysis on the neoantigen peptides in Supplemental **Figure S5** to provide further evidence for their detection.

Line 367: My suggestion would be this sentiment can be excluded as the citation referenced for (for which Abelin is an author) used parallel reaction monitoring to target KRAS epitopes for enhanced sensitivity over DDA analyses, therefore I would not anticipate this experimental format would capture all KRAS antigens. Furthermore, the paper cited did use a cell line that endogenously presented mutant KRAS to identify mutant KRAS-containing epitopes (not artificial by my definition, though authors should clarify if by artificial they mean cell line).

- The text has been updated to address this concern as follows:
 - pg 7- "Although our MONTE immunopeptidomes were not able to capture all validated KRAS neoantigens found using targeted mass spectrometry and both an artificial

overexpression system and in one case a cell line endogenously expressing KRAS G12V¹⁴, this limitation on detectability will diminish as even more sensitive MS instrumentation and data generation approaches are introduced.“

Line 392: “Few of the specific PTM sites presented by HLA were detected in phosphoproteomes and acetylomes” –Is it possible this is because some of the phosphorylated HLA peptide identifications are incorrectly assigned? What are the second highest ranked sequences of the phosphorylated peptides and what is the score differential? Is it also possible that the proposed issue with protease inhibitors potentially losing their potency through processing may have biased phosphoproteome identifications? Please discuss.

The text was updated to acknowledge that all HLA detected phosphopeptides had fully localized sites. pg 11- *“The immunopeptidomes were also searched for PTM modified peptides. We observed HLA-I and HLA-II fully localized phosphopeptides at 0.11% and 0.3%, respectively, and acetylpeptides at 0.08% and 0.10%, respectively, of total unique peptides.”*

- Supplemental **Table S9** reported “numLocalizedVMSites_sty” and “numAmbiguousVMSites_sty” columns providing details about phosphosite localization.
- Supplemental **Table S9** reported the “rank2Score”, “deltaForwardReverseScore”, and “deltaRank1Rank2Score”, as described in the “Column Descriptions” Tab.
- The text has been updated to suggest adding a second round of phosphatase inhibitors to the serial HLA IPs on pgs 6-7.

“The proteome and ubiquitylome results demonstrate that similar numbers of canonical human proteins (11,028 vs.10,729) and K-ε-GG peptides (9,516 vs. 9,419) were identified and fully quantified between the non-HLA-enriched (“No HLA”) and HLA-enriched (“HLA FT”) samples, respectively. A 16% decrease in the total number of phospho-sites (~8% phosphorylated proteins) was observed when using the HLA-enriched samples (No HLA: 26,627 phosphosites, 6,745 phosphoproteins; HLA FT: 22,339 phosphosites, 6,235 phosphoproteins), suggesting that the phosphatase inhibitors added to our lysis buffer may be losing their activity during the protein-level, HLA immunopeptidome enrichment. The number of lysine residues observed to be acetylated on internal lysine residues (i.e, not at the N- or C-terminus of the peptide) increased by 45% in the HLA-enriched samples (No HLA: 3,702; HLA FT: 5,380 internal K-acetylsites). The relative yield of acetylated peptides (i.e., the percentage of K-Ac peptides relative to the total peptides identified in the sample) in the HLA processed samples was significantly higher (75% vs. 55%). Given that the protein lysates were incubated at 4°C for 6 h during HLA enrichment, we sought to rule out possible non-enzymatic acetylation⁴⁰. Acetylome analysis of A375 melanoma cells with and without the 6h HLA IP incubation conditions yielded a similar number of acetylated peptides when compared to no HLA incubation conditions (Table S6), suggesting the addition of the HLA IP did not cause non-enzymatic acetylation. We speculate that the increased yield of acetylation sites could be due to pre-clearing of non-specifically binding components in the complex tissue lysates by HLA- and K-ε-GG antibodies.

Biological signals observed in a serial multi-ome enrichment workflow without HLA enrichment are recapitulated in the MONTE workflow

To assess potential differences between HLA-enriched and non-HLA-enriched samples, we analyzed the ten LUAD tumor proteomes, ubiquitylomes, phosphoproteomes, and acetylomes using a principal component analysis (PCA) (Figure 4B, Table S7). PCA shows that samples cluster by LUAD tumor, not by the processing method used, demonstrating that biological differences among the samples are stronger than technical variation between these serial workflows. The acetylomes of HLA-enriched and non-HLA-enriched samples were somewhat less well correlated. The total number of proteins identified and quantified from HLA-enriched and non-HLA-enriched samples were shown to have a 93% overlap (Figure 4C, D). Slightly fewer proteins (3%) were identified from the HLA enrichment flow-throughs. We looked into the proteome data for depletions in known HLA-I and HLA-II chaperone proteins to confirm our serial HLA-II and HLA-I immunopurification is not depleting known HLA protein binding partners. We did not observe proteome depletion of HLA-I chaperones CALR, CANX, or TAPBR or HLA-II chaperones HLA-DM and HLA-DO. These observations suggest the addition of the serial HLA immunopurification does not have a significant negative impact on the downstream proteome analysis.

The overlap between HLA-enriched and non-HLA-enriched protein lysates was 60% for ubiquitylation sites, 72% for ubiquitylated proteins, 63% for phosphorylation sites, and 78% for phosphorylated proteins, which is an expected result using multiplexed, data-dependent LC-MS/MS methods for highly similar processing workflows²¹ (Figure 4C, D). **A 16% loss of total phosphosites was observed in the HLA enriched lysates, which we attribute to the combination of the losses from the extra desalting step in UbiFast and the possible decrease of phosphatase inhibitor activity over the 6 h HLA serial enrichment.** To improve this in future studies, we plan to implement a second addition of phosphatase inhibitors between the HLA-II and HLA-I enrichments. The lowest overlap across experiments was observed for acetylome data because 45% more acetylated peptides were observed in the HLA enriched samples. Overall, the HLA-enriched samples capture the same depth of coverage observed in non-HLA-enriched samples and adding this enrichment step up front in a serial workflow does not introduce significant bias in downstream proteome, ubiquitylome, phosphoproteome, and acetylome.”

Line 387: This is also another area for further exploration within this dataset, particularly because it is mentioned in the discussion these data can uncover dependencies that may improve prediction (Line 436). Are any of these phosphorylated HLA peptides that were also identified as phosphopeptides map to signaling pathways that are enriched in that particular tumor? Do you see increased phosphorylation in tumor versus normal of the samples previously profiled? Same question for acetylation. Do EGFR/KRAS mutant tumors show evidence of phosphorylated (or regular) HLA peptides that match disease signatures? These data again provide more support for using the MONTE workflow, and also may highlight how biomarkers can be used to predict immune presentation. This has the potential to provide a wealth of

biological insight, however many of these questions aimed at integrating the datasets remain unexplored.

- Due to the limited number of HLA phosphopeptides identified, and the small number of (10) of tumor samples available for these analyses, it was not possible to definitively answer these questions and are beyond scope of this method focused manuscript.

The ubiquitination data is not incorporated into any of the tumor analyses outside of reporting number of identified peptides. Is there a way to use this data to better understand the biology of the samples profiled? Can the authors provide a compelling example? Otherwise, there is not strong evidence for incorporating it into the workflow, particularly when it appears to increase CVs in phosphorylation data.

- Lines 334-342 (**Figure 6A**) discusses the overlap between the HLA immunopeptidome and ubiquitylome data, as it relates to improvement to HLA peptide prediction. Because the proteasome degradation pathway plays a major role in HLA-I presentation, we believe including both -omes will reveal biological insights in larger scale studies. We already discussed in the text that the presence of a ubiquitin modification on a protein may be a feature that could be used to further improve HLA-I peptide presentation predictions (see text copied, below).
- *pg 9-“A higher proportion of ubiquitylated proteins, 89% (3499/3954), were detected as HLA-I source proteins, compared to 49% (1939/3954) as HLA-II source proteins. This was expected because ubiquitylated proteins are a key source of proteasomal processed peptides that are HLA-I peptide precursors. Conversely, we noted only 26% (3499/13285) of HLA-I source proteins were identified as ubiquitylated, suggesting that deeper ubiquitylome datasets are required to fully overlap with HLA-I immunopeptidomes. Because HLA-I source protein expression levels and their ability to be processed by the proteasomal pathway are important factors for presentability, both proteome and ubiquitylome datasets are useful for incorporation into HLA-I prediction algorithms.”*
- As a result of the small number of tumor samples available for these analyses, it is not possible to provide a specific example of an important biological finding provided by ubiquitylomics. However, the lack of a specific example is not a reason to exclude ubiquitylome analysis from the workflow. The squamous cell lung cancer paper clearly shows the value of incorporating ubiquitylation into tumor analyses (Satpathy et al. *Cell* 2022) so having a workflow that incorporates it along with the other omes, and that is able to obtain that information for each sample without increasing the required amount of input sample should be clear.

Discussion

Line 403: More accurate to write “Serial HLA-II and HLA-I” profiling.

- Text has been updated to emphasize the order of the serial HLA-II and HLA-I workflow.

Line 408: Unless I am misunderstanding, the technical hurdle of doing enrichment of ubiquitinated peptides before TMT-labeling has already been addressed in Udeshi et al., 2020,

therefore it should not be highlighted a technical hurdle overcome by the MONTE workflow. Authors should rephrase to accurately report the innovations specific to this manuscript.

- Text related to the inclusion of UbiFast has been updated throughout to differentiate what is new in this manuscript vs. what has already been published.

Methods

Please evaluate for completeness of methods used to generate and interpret the data within this study. Several examples of missing methods are pointed out above.

- Methods have been updated to address concerns regarding the PCA analysis and that the SDS based S-trap was used for Figure 2 generation. Specifically, a new subsection named “**Principle Component Analysis (PCA) using ProTIGY**” on pg 29 has been added to the methods.

Reviewer #2 (Remarks to the Author):

In this work, Abelin et al. present a MONTE workflow describing the sequential isolation of HLA peptides, and peptides modified by ubiquitin, phosphorylation, and acetylation from the same piece of tissue. The authors demonstrate that the quality of each PTM dataset is not significantly hampered by the sequential use of the LUAD lysate, and that interpretation from each PTM dataset is informative in itself, though a synergistic interpretation from all datasets was not provided, to the disappointment of this reviewer. While the data boasts high quality and reproducibility, and the lab is well-known for high technical competency, the manuscript seems not packaged well for the Nat Comm audience, and reads like a “protocol” appended with small interesting observations, in each PTM or ligandome dataset, that do not tell one story. Concatenated PTM peptide isolation and analysis is itself not new, and this reviewer has reservations about the utility and cross-interpretation of PTM datasets. Specific concerns are listed below:

1. Introduction is too brief, and far from sufficient to appreciate why serialization is critical for better biological understanding in tissue specimen. Fundamentally, sequential purification of PTMs is not new, Olsen & Mann reviewed this in Molecular and Cellular Proteomics already in 2013 from the technical perspective, and Olsen has another one in Cell Systems in 2017 describing the shotgun generation of comprehensive proteomes. The latter even highlighted deep coverage of PTMs without specific enrichment (to arrive at 7k acetylations, 10k phosphosites). What is then the real gain after this huge amount of work with serial preparations? There are many more workflows that concatenate multiple PTM purifications. Without a compelling rationale to justify this basket of PTMs that the authors want to measure together, it is not convincing why MONTE will be frequently applied by investigators to understand the underlying biology. But rather, this comes across as a mere demonstration of yes you can automate, and piggy-backed to the new UbiFast procedure the group has established and published elsewhere.

- The introduction has been updated to focus on the essentiality of being able to obtain both HLA-I and HLA-II immunopeptidomes from exactly the same samples and using the same amounts of sample as have been used previously in well-established multiplexed multiomics studies such as those carried out by labs in the CPTAC and MoTrPAC consortia among many others. It is well recognized that having HLA peptidomics data on tumors can yield valuable biological information of potential clinical utility. We believe that adding immunopeptidomics to the already established serial processing that provides deep and quantitative analysis of the ubiquitylome, phosphoproteome and acetylome, and demonstrating that it can be done without requiring a separate large sample of precious human tumor tissue and without sacrificing depth in the HLA peptidome is highly relevant and beneficial.
- We have developed and optimized the first method (and presently the only method) that enables deepscale HLA-I and HLA-II immunopeptidomics, ubiquitylome, proteome, phosphoproteome, and acetylome data to be obtained on the exact same sample without significant compromise in the depth or reproducibility of the data for any individual ome. We have demonstrated the method using patient derived xenograft tumor tissue and patient-derived lung cancer tumor samples. Papers in top tier journals describing the biological and clinical significance of HLA immunopeptidomics data alone appear regularly^{4,5,8,9}, but these studies are accomplished using hundreds of mgs of wet weight tissue dedicated to this analysis, whereas our serialized approach that generates data for multiple omes uses just 50 mg wet weight tissue total for input for HLA-I, HLA-II and all downstream omics. The value of obtaining HLA immunopeptidomics data in serial on the same samples used for multiomics studies is clear, and this is the first report describing in detail how to accomplish it. We have also further strengthened the discussion of the essentiality of obtaining as much high quality, deep “omics” information as is possible from each single precious human tissue sample.

2. A large assortment of PTM purification can in theory be serialized from the same tissue. Why this particular assortment? And why in this particular order? It would be more satisfying to explain how these PTMs are connected in the biology, and why collectively studying these modifications is important to sketch the mechanism. Can you also connect PTMs on the same protein? For instance, p53 is known to be phosphorylated, ubiquitinated and acetylated, but also degraded and possibly presented as antigen peptides. Would more instances of such examples strengthen the real need for MONTE?

- The discussion was updated to address why this order of serial enrichments was selected. pgs 11-12- *“After implementing HLA-II and HLA-I immunopeptidomics and ubiquityl enrichment into a serial proteome and PTM enrichment workflow, we observed high correlation between the proteomes, ubiquitylproteomes, phosphoproteomes, and acetylproteomes in both our breast cancer xenograft and LUAD datasets, showing that additional data layers can be acquired without prejudicing data quality and demonstrating the utility of the MONTE workflow. The order of ‘omic analyses was determined by the biochemical requirements of each enrichment and the previously established serial enrichment workflows. We anticipate that additional ‘omes, such as*

phosphotyrosine peptide enrichment, could be incorporated into the MONTE workflow, with such enhancements mindful of the compatibility of enrichment reagents and minimization of desalting steps to maximize peptide recovery. The current MONTE workflow can also be tailored to include or exclude enrichments based on the specific biological questions being addressed, demonstrating the flexibility of serial enrichment workflows. Overall, the MONTE workflow represents a path forward to deeply characterizing each single patient sample that was only possible previously with parallel processing of multiple tissue aliquots.”

- An analysis of source proteins observed across all PTM-omes has been carried out, and we developed a publicly accessible data viewer that easily allows readers to interrogate if their protein of interest is post translationally modified and detected in HLA immunopeptidomes: <https://proteomics.broadapps.org/CPTAC-MONTE2022/>. We used this to look at known oncogenic and tumor suppressor proteins EGFR, KRAS, RB1 and STK11 in both the HLA enriched and non-enriched samples and confirmed that similar levels of protein and PTM sites were detected. We were also able to detect HLA-I peptides derived from EGFR, KRAS, RB1, and STK11 and HLA-II peptides from EGFR. These observations were added to the revised manuscript. We also looked at TP53, which is processed and presented on HLA-I and HLA-II, but not reproducibly detected in our proteome data. We confirmed that TP53 also had variable detection in the larger discovery dataset (Gillette et al, Cell, 2020) and shown below it is not detected in 3/10 LUAD tumors used for the MONTE cohort. This is an example of a protein that may be difficult to detect in the proteome because of its high turnover rate, and more easily detectable in the HLA-I immunopeptidome. These analyses are now shown in new **Figures S3, S4**.

3. The authors listed 3 major changes in the serial preparation steps: (I) UbiFast first before other purifications, (II) add HLA-I/II purification before multi-omics, (III) SDS with S-trap instead of Urea. It is very not intuitive to list these in this order, because these do not follow the sequence of retrieval in the presented workflow. Sodium deoxycholate might even be a better detergent to use than SDS for native pulldowns. It's easily precipitated on acid elution of HLA peptides, and almost as potent as 8M Urea lysis for proteomics. UbiFast is a good technique for the community, but that has been accepted as a different paper. Minus that, this current set of “improvement”, is quite minor.

- Multiple changes to established HLA immunopeptidome protocols had to be made and tested including 1) testing different protease inhibitors specific to each proteome and PTM ome to ensure no negative impact on the HLA-I and HLA-II IP 2) reversing the order of the HLA immunopeptidomes to perform the HLA-II IP first to ensure not HLA-I contamination in the data 3) scaling the immunopeptidome protocol to be performed in a 96 well plate 4) reducing the HLA-I and HLA-II immunopeptidome processing time by implementing washing samples in ½ plates and rejoining to desalt together in one desalting plate. Additionally, we tested multiple ways of digesting the proteins that remained post HLA enrichment (Urea, chloroform/methanol precipitation, S-Trap), to determine that the S-Trap resulted in the best tryptic peptide coverage of the proteome and was the most scalable of the

approaches as it can be performed in a 96 well plate format. Furthermore, while UbiFast has been published, this is the first time the method has been applied in a serial enrichment format to enable efficient use of limited biological and clinical samples. Previously, the UbiFast method has been used to measure separate sample of tumor (another 750 ug) and was not in serial with the proteome, phosphoproteome and acetylome as was done here. None of the above modifications and changes are minor tweaks to the method. We have now detailed these optimizations (**Table S1**) and added supporting data to the revised manuscript.

4. PTM data needs to be carefully interpreted. For instance, when comparing peptide intensities obtained from different PTM preps, it might even be invalid to conclude on occupancy of PTMs, relative proportion of modified peptides versus unmodified. And more importantly, HLA peptides are not tryptic, so these will not align with multi-omic peptides. Experimental digestion with trypsin/lysC can split the precursor of one HLA peptide into multiple tryptic peptides that ionize differently etc. Ubiquitinated peptides, though eventually will become source peptides for antigen presentation, will also be trimmed by other proteases in the loading path. Hence, again, it will be likely hard to find back the same precursor peptides to compare between PTM datasets. The cross-interpretation between different PTM datasets seems very convoluted, difficult, and even impractical to use. I envision this is going to be difficult even for highly skilled proteomics data analysts.

- We agree with the Reviewer that the PTM data needs careful interpretation, and multi-omic PTM and immunopeptidome data integration is not easily accomplished, but it can be done and has been done here. This comment seems more of an endorsement of our approach than a criticism. We use TMT labeling to achieve precise quantitation where all downstream sample handling effects occur equally to the samples being compared. We agree with the Reviewer that it is otherwise problematic to compare peptide intensities obtained from different PTM preps. We likewise do not seek to conclude on PTM occupancy vs unmodified. Furthermore, in multiomic comparisons we typically look at quantitative effects on the source proteins, relative to individual PTM sites and antigens. We typically do not seek to compare across 'omes at the peptide level. Consequently, we have made a few clarifications to the text:
 - For example, at the PTM-site level we attempted to address if highly abundant phospho and acetyl sites were more likely to be observed on HLA peptides (**Figure 6D-E**). We believe we were not clear that we were comparing phospho and acetyl sites in these analyses, which would account for multiple tryptic peptides containing the same site, so have updated the text and **Figure 6** to clarify our definition of PTM sites.
 - Related to the ubiquitylome and immunopeptidome data comparison in **Figure 6A**, we compared HLA immunopeptidome and ubiquitylome source proteins instead of peptides/sites to account for our inability to map specific sites, as the ubiquitin is removed during proteasome processing. We updated the text and **Figure 6A** to clarify our use of source protein to look at overlap.

5. Obviously the order of retrieval would make a big difference for peptides harboring more than one PTM. This might partially explain the big variability observed in Figure 2C-D. UbiFast before phospho-enrichment obviously changes the coverage, and introduces more variation. Could phospho-degrons influence the non-overlap in Figure 2C? If the authors choose to feature this basket of PTMs, it is almost impossible not to think of phospho-degrons.

- UbiFast is implemented at the ubiquitylated peptide level not the ubiquitylated protein level. Therefore, depletion of phosphorylated residues in the UbiFast step could occur for phosphorylation sites present on the same tryptic peptide harboring a K-GG site. To understand if a significant number of phosphorylation sites are enriched in the UbiFast step prior to IMAC enrichment and therefore depleted in the IMAC data, we searched our UbiFast data shown in **Figure 2** for peptides with co-occurring phosphorylation and K-GG sites. We identified 148 human peptides harboring both a K-GG and phosphorylation site. Given the number of co-occurring phospho and K-GG modified peptides is low, we determine that UbiFast does not significantly deplete phosphopeptides and affect correlation.

6. Then the authors feature bits of small interesting observations all jam-packed at the end of the manuscript. For instance, neoantigens, nuORFs, whether WES captures mutations, possibly using PTM abundance to improve detection of PTM HLA peptides, whether KRAS epitopes are conserved in presentation... Some of these are interesting ideas to pursue, but the presented data is not enough to make a strong point in any of these, that is a pity.

- We thank the Reviewer for acknowledging these interesting findings. We acknowledge that this manuscript is method focused, and that our small LUAD cohort of 10 patients is not enough to do a deep, large scale multi-omic analysis. In this paper describing a novel, new, information-rich method for tissue analysis we focus on the effects of serializing the omes on depth and quantitative reliability in PDX samples and use the LUAD samples to demonstrate what can be done with real clinical samples. We hope that the additional analyses and the availability of the publicly accessible data viewer <https://proteomics.broadapps.org/CPTAC-MONTE2022/> now included will provide greater evidence for the utility of our methods.

7. Overall it reads as if the authors are undecided, as to whether this manuscript should be a protocol for a multi-omics audience, or an antigen discovery audience, or potential broad biological users that read Nat Comm. In the current form, the manuscript seems most suited to go to a proteomics journal like MCP. The technical aspects of this work are excellent, but biological interpretation leaves a lot more to be desired. There is so much data in here that is undigested and not interpreted.

- We thank the Reviewer for acknowledging the quality of the technical aspects of our work. We hope that the additional analyses performed will provide further evidence that the MONTE workflow will enable further multi-omic analyses on samples that may have had to be split previously and limited the number of possible multi-omic analyses.

Reviewer #3 (Remarks to the Author):

In this manuscript, the Carr lab reports on an extension of their methods aiming at characterizing proteomes at several levels, notably PTMs and immunopeptidomes. The authors call their approach MONTE that effectively enables the serial serial enrichments/processing of HLA, ubi, ac, phospho and full proteomes from the same sample (cell lines or cancer tissue). The previously published serial methods have enjoyed success and have been taken up by many labs. The extension portrait here, promises to attract further followers. Therefore, this reviewer supports publication of the work provided that the two following issues are adequately addressed:

line 85: here and in general, please avoid giving readers the impression that MONTE does not require a lot of starting material. It may be less than what other approaches require, but 50 million cells or 200 mg of wet tissue is still a lot of material by any standards and may not be available in many clinical scenarios. Please make sure readers are aware of the limitations and provide some guidance where MONTE is likely to be successful and where not.

- The text has been updated to address this concern, and further guidance on MONTE limitations have been provided. Please see the examples of text additions, below.
- pg 12- *“Although we demonstrate the usefulness of MONTE in a small LUAD cohort that expressed both HLA-I and HLA-II where ~50mg or greater of cryopulverized tissue was available, this workflow can be extended to other tumor types with less available tissue, low HLA expression, and no HLA-II expression. In these scenarios, performing the HLA serial enrichments will likely result in lower immunopeptidome depth, but will not prevent the downstream multi-omic analyses, as these require less input than the HLA enrichment. Even in cases where the HLA expression in tumors is low or knocked-down, useful information like what HLA alleles are being expressed and presenting peptides can be directly determined from these immunopeptidomes and leveraged to better understand changes in tumor HLA peptide presentation.”*
- and in the Discussion, pg 13- *“There are limitations to this study. Although it demonstrates the feasibility and utility of a workflow incorporating HLA-I and -II immunopeptidomics and UbiFast ubiquitylproteomics into a serialized proteomic workflow using a clinically relevant sample set, its pilot-level scale precludes the statistically robust analyses, deep explorations of biology, or compelling assessments of the interplay between characterized ‘omes that the approach is intended to facilitate. Rather than highlighting such underpowered and speculative results, we therefore chose to focus on the added value and interpretable results provided by immunopeptidomic characterization of tumor samples. Recent large-scale cancer proteogenomics analyses have made a compelling case that the integration of proteomic, ubiquitylproteomic and especially phosphoproteomic data with genomic data helps to functionalize genomic aberrations, providing new perspectives on cancer biology and nominating potential therapeutic vulnerabilities¹⁴. Nevertheless, it remains to be shown that the integration of additional layers of data, such as the immunopeptidome, will continue to provide new,*

interpretable, and actionable insights. The MONTE pipeline, when applied to samples from a suitably sized patient cohort, provides the means to test that hypothesis.”

validation of expression of nuORFs, lincRNAs, mutated KRAS etc: the authors should comment on their confidence in the identification of such non-canonical gene products. This has been much debated in the field and adding an HLA or PTM component to this repertoire make arouse further criticism. It may be outside the scope of the current work, but it would have been desirable to see some validation that these peptides are true identifications. Labeled spectra are one way of doing this but given the availability of several spectrum prediction tools, a more systematic analysis may be warranted (at least for peptides that do not carry PTMs). Please provide some guidance to readers here so that scientists less aware of these issues are not carried away in excitement that may not always be well founded.

- To address the concern regarding the confidence of peptides derived from noncanonical peptides, we implemented a more stringent subset-specific FDR threshold for nuORF derived peptides. We also performed retention Time prediction analysis using DeepLC⁷ for nuORF and neoantigen HLA-I peptides in the new **Figure S5**. Please see updates to the text below:

*pg 8- “Next, we sought to detect peptides in our LUAD HLA immunopeptidomes derived from nuORFs whose translation has been supported by ribosome profiling using a recently published nuORF database¹⁵(**Figure 5C**). High-confidence HLA-I and HLA-II peptide identifications derived from nuORFs were found across 9/10 patients. Because nuORFs represent rare observations within a large dataset, after false discovery rate (FDR) thresholding on the aggregate data set, we applied more stringent subset-specific FDR thresholding to the nuORFs(see **Methods**). A majority of nuORF peptides also had predicted retention times that correlated well with their observed retention times, further increasing the confidence of detection (**Figure S5**).”*

References

1. Gillette, M. A. *et al.* Proteogenomic Characterization Reveals Therapeutic Vulnerabilities in Lung Adenocarcinoma. *Cell* **182**, 200–225.e35 (2020).
2. Chong, C. *et al.* Integrated proteogenomic deep sequencing and analytics accurately identify non-canonical peptides in tumor immunopeptidomes. *Nat. Commun.* **11**, 1293 (2020).
3. Ruiz Cuevas, M. V. *et al.* Most non-canonical proteins uniquely populate the proteome or immunopeptidome. *Cell Rep.* **34**, 108815 (2021).
4. Bassani-Sternberg, M. *et al.* Direct identification of clinically relevant neoepitopes presented

- on native human melanoma tissue by mass spectrometry. *Nat. Commun.* **7**, 13404–13404 (2016).
5. Marcu, A. *et al.* HLA Ligand Atlas: a benign reference of HLA-presented peptides to improve T-cell-based cancer immunotherapy. *J Immunother Cancer* **9**, (2021).
 6. Mani, D. R. *et al.* Cancer proteogenomics: current impact and future prospects. *Nat. Rev. Cancer* **22**, 298–313 (2022).
 7. Bouwmeester, R., Gabriels, R., Hulstaert, N., Martens, L. & Degroeve, S. DeepLC can predict retention times for peptides that carry as-yet unseen modifications. *Nat. Methods* **18**, 1363–1369 (2021).
 8. Bulik-Sullivan, B. *et al.* Deep learning using tumor HLA peptide mass spectrometry datasets improves neoantigen identification. *Nat. Biotechnol.* **37**, 55 (2018).
 9. Olsson, N. *et al.* An Integrated Genomic, Proteomic, and Immunopeptidomic Approach to Discover Treatment-Induced Neoantigens. *Front. Immunol.* **12**, 662443 (2021).
 10. Yadav, M. *et al.* Predicting immunogenic tumour mutations by combining mass spectrometry and exome sequencing. *Nature* **515**, 572–576 (2014).
 11. Subramanian, A. *et al.* Gene set enrichment analysis: a knowledge-based approach for interpreting genome-wide expression profiles. *Proc. Natl. Acad. Sci. U. S. A.* **102**, 15545–15550 (2005).
 12. Krug, K. *et al.* A Curated Resource for Phosphosite-specific Signature Analysis. *Mol. Cell. Proteomics* **18**, 576–593 (2019).
 13. Djureinovic, D. *et al.* Profiling cancer testis antigens in non-small-cell lung cancer. *JCI Insight* **1**, e86837 (2016).
 14. Choi, J. *et al.* Systematic discovery and validation of T cell targets directed against oncogenic KRAS mutations. *Cell Reports Methods* **1**, 100084 (2021).
 15. Ouspenskaia, T. *et al.* Unannotated proteins expand the MHC-I-restricted

immunopeptidome in cancer. *Nat. Biotechnol.* (2021) doi:10.1038/s41587-021-01021-3.

REVIEWER COMMENTS

Reviewer #1 (Remarks to the Author):

Summary

Here, the authors present a revised manuscript detailing a serial workflow, termed "MONTE," for obtaining multi-omics datasets from a single tumor sample of limited quantity. While authors have added several additional analyses and details to support the claims in this study, including synthetic peptide validation, integration with published datasets, and expansion of methodological optimization details, I remain convinced that this study is more appropriate for a methods-focused or proteomics-focused journal. The level of methodological novelty described within this study is limited to validating serialization does not (in most cases) negatively impact data quantity or quality. Still, there are examples, noted below, where claims made are not supported by the data, where variation resulting from serialization (ex. Impact on variation in phosphor-datasets with addition of UbiFast) is minimized or not addressed, and where statistics were not employed to rigorously test claims.

Furthermore, by the authors' own admission, this 10 tumor LUAD cohort is not large enough to demonstrate the utility of integrating these datasets with robust biological insight. When biological insight attempts are made, claims are often not supported by datasets upon closer inspection (examples below). In the response to reviewers, authors highlight the inclusion of Table S1, demonstrating the different methods development studies used to develop this protocol as an example of the methodological advancement. However, a close examination of this table highlights that many of these optimizations do not offer a substantial difference in data quantity. For example, the difference between an S-trap analysis and Urea tryptic proteome analysis resulted in a difference of <200 total proteins identified between 2 of the tested conditions. Yet, this is highlighted in the authors' rebuttal as a key point of novelty in their method. Furthermore, both methods are routinely used by others in the field, therefore selection of one over the other does not pose a significant improvement in methodology.

Other points of novelty highlighted by authors in the response to reviewers include testing different protease inhibitors (point #1) also do not show meaningful differences in IDs (described in detail below), and with n=2 replicates it is challenging to glean clear trends beyond run-to-run variation. Point #2 describes performing a class II IP first as a novelty, though studies exploring the impact of Class I>II and Class II>I have been previously published (described below). Point #3 focuses on a 3:1:1 antibody mixture of class II antibodies as a methodological advancement, however the average difference in the number of unique peptide IDs is again <200, with the 1:1:1 ratio giving the highest individual number of peptide identifications in a single analysis. Again, with analyses performed in duplicate, it is hard to untangle a meaningful difference in peptide IDs versus standard variation in MS analyses. Point #4 describes scaling to a 96-well format for HLA enrichment, though this has been described in various formats in papers from several years ago (PMID 29242379, PMID

34129938, PMID 33331781). Authors also focus on their ability to perform HLA profiling on samples of limited material but overstate the novelty of this element given other published works.

While authors took care to calibrate the various aspects of this serialization protocol, ensuring the data was not substantially compromised by the addition of HLA profiling and ubiquitination studies, in my view the MONTE method remains a concatenation, with adjustments, of previously published methods by this group, and a limited demonstration of the power of integrating these datasets. While I believe this work warrants publication, I do not believe Nature Communications is an appropriate venue based on methodological novelty and depth of biological insight provided.

Specific points:

Line 38-39: Presented as a universal fact that decisions must be made in limited quantity contexts. Consider re-wording from “Decisions have to be made” to “researchers may need to make decisions” or something similar.

Lines 71-75. I appreciate the author’s response to the initial concern raised about the lack of citations supporting the 1g/1B cell reference, however, still feel this section is overlooking other contributions/advancements made in this space in recent years and highlights the maximum number of cells or tissues used in select studies to embellish the advancement of using 50-91 mg of tissue for HLA analyses. For example, Chong et al. (Ref 28) utilized 200M patient derived cells. Bassani-Sternberg et al. (ref 2) used a maximum of 1g of tissue in a single analysis but a minimum of 100 mg, deeply analyzing the class I and class II immunopeptidomes serially by DDA (though class I followed by class II), identifying tumor antigens and neoantigens, as described here. Another study by the Carr group used 100mg of tissue (PMID: 34391888) that combined small tissue amounts with gas and liquid fractionation (very similar to method used in this paper) to extend immunopeptidome depth in class I analyses. It should not be overlooked that PMID 34497125 performed class I profiling in clinical biopsies of similar (or smaller) size, also identifying tumor antigens like shown in the analysis of Figure 4. While these studies did not integrate other ‘omics datasets in the MONTE way and the objective of the studies were different, studies like PMID 34497125 and PMID 34391888 should be cited as an example of the movement of the field towards using limited material for HLA studies, while acknowledging that additional methodological development is required to integrate those findings with other ‘omics level datasets—hence this publication.

Furthermore, the author’s rebuttal states, “We believe that the immunopeptidome depth using 50mg tumor (~2 mg protein) is novel,” further stating that the enhanced # of average peptide identifications in this study is evidence of methodological novelty in comparison to PMID 34497125. In this context, more peptide identifications does not translate directly to novelty. A variety of variables, including most importantly the natural HLA expression in the samples could contribute to the difference in average in identifications, even on the same instrument. Here, the authors perform

serial HLA-II/HLA-I enrichment on PDX samples and recover only <600 unique peptides from these limited sample amounts, underscoring the point that the sample itself can play a significant role in dictating the quantity of IDs obtained.

As acknowledged in my initial response, this study provides very impressive depth in immunopeptidome coverage from limited material in the LUAD tissues. However, I remain adamant authors not overstate the individual novelty of the tissue quantity component, or the ability to detect relevant antigens in small sample inputs due to other published works in this area. To fairly and appropriately acknowledge/reference other works which have made iterative progress prior to this work does not take away from the novelty that the MONTE workflow holistically provides. I would strongly recommend the authors revisit this section of the manuscript to most accurately highlight where their methodological advancement sits in the landscape of recent works attempting similar objectives (i.e. gleaning relevant HLA insight from limited tumor tissue samples) and give appropriate credit to other works where deserved.

Line 109: The inclusion of the data viewer is a nice addition. However, the viewer presents a heatmap that is a bit challenging to interpret at present- at least on my laptop screen looking only at EGFR (for example, below) the legends run into each other, and it is hard to tell which header goes with which color. Consider making legends on multiple lines to fit different computer sizes?

When you select to view “all” PTM sides, the “Relative abundan” legend overlays the site IDs (below)

Is there a way to hide either the HLAft or noIP samples? I think that could be beneficial for other use cases of this data, especially given the missing values between datasets.

For the HLA tables, it would be beneficial to have the typing data of each patient easily accessible in this tool. Integration with HLA-Athena, for example, to report which allele is the best scoring (most likely to present the peptide) per patient would be an additional piece of helpful information not necessarily required for publication, but just sharing some suggestions as I anticipate this could be used in future CPTAC papers.

Line 131: Note “Prior publications” is plural but only one publication is referenced.

Other studies have explored the impact of enrichment order of class I/II in serial (and compared to parallel), reporting class I contamination in the class II pulldown when class I is immunoprecipitated first (Zhang et al. PMID 33331781). This study (or another) should be referenced in line 132 to this point. Zhang et al. also reported class II contamination in the class I data when the class I IP was

performed first (Fig 3). Did the authors here perform a rigorous assessment of $II > I$ versus $I > II$? It is clear from other works such as Zhang et al. that the second IP will result in lower data quantity. Please discuss.

Line 134: Table S1, section “HLA-II antibody ratios” reports average IDs of $n=2$ samples using different antibody ratios. The difference in IDs between sample could easily be explained by run-to-run variation or sample handling and does not demonstrate an obvious innovation or clear case for selecting a 3:1:1 ratio. This variation is clear in other $n=2$ replicates performed in the “HLA_rawfile_mappings” tab. In fact, the highest number of IDs (narrowly) was the 1:1:1 condition, sample “01.” Without at least $n=3$ replicates, these data are challenging to interpret, and innovation in this area should not be overstated.

Table S1/Lines 133/“HLA-rawfile_mappings”. More information regarding the different conditions (ex. Concentration/catalogue # of PR-620/621/622) and interpretation of the results would be informative to understand the combination of conditions that were selected to optimize HLA IDs, and so the conditions are reproducible. This is probably most appropriate to add to the supplement.

Table S1 “Proteome method”. Authors should list the unique IDs in each replicate, not just combined data, to most transparently present results.

Line 166-169: Authors have removed the box and whisker plots and dot plots showing summary and individual correlation measurements and instead reported only median correlations in a table. The original figure 2D/2E (below) raised concerns with both myself and Reviewer #2, as it is clear that while the median correlation coefficients may be similar, there is much higher variation in the correlation coefficients in the +UbiFast method for the phosphoproteome data. The removal of these figures appears an attempt to detract attention from this result, which is not in line with transparent reporting of data. I repeat my initial sentiment, which is that the variation resulting in the addition of Ubitfast should be discussed and the figures illustrating this variation (with individual measurements or box plots) should remain in the manuscript. Again, without separate replicate plexes, it is challenging to interpret in the correlation plots whether the lower R^2 is a result of generally higher variation in phosphoproteome methods than proteome, or whether the noise has come from adding UbiFast. The new experiment did not address this concern, as it was all contained within a single 16-plex.

While I appreciate the author’s inclusion of the experiment depicted in Figure S1, unfortunately the experiment to address whether the # of pY residues decreases with UbiFast was setup within a single 16-plex (+/- UbiFast), greatly diminishing the possibility of detecting a meaningful difference

between the two experimental conditions. In the dataset depicted in Figure 2, can the authors report # of pS/pT/pY peptides to address the original question, given these analyses were run in separate plexes?

My other question initially posed in the response to reviewers asked whether there was a difference in the number of phosphorylation sites that are show significantly different ($p < 0.05$, maybe a fold-change cut-off) phosphorylation levels between basal and luminal subtypes between the +/- UbiFast datasets. I pose this question, as I was concerned the higher quantitative variation in the +UbiFast condition would result in some biological insight being masked. For example, perhaps 800 phosphosites show significant difference phosphorylation between the basal and luminal subtypes in the -UbiFast condition, but this drops to 600 in the +UbiFast dataset due to the higher CVs across measurements. If this is not the case (and higher variation is just a consequence of generating this additional valuable dataset but does not significantly impact biological insight), then that would be good to note.

Figure 2E. I recognize the enrichment analyses are a way to show that the biological insight gained out of the datasets +/- UbiFast are similar, however more information needs to be included in this figure. How were the 5 highlighted pathways selected? What are the associated p-values with each of these selections? The Table S2 tab is difficult to interpret—there are only 2 significance values reported, which appear to be the p-value calculated between the same subtype, +/- UbiFast. The headers are labeled “signed.Log.P.Value.UbiFT_Basal.over.No_Ubi_Basal” in column M of the GSEA_Phosphoproteome tab, for example. Below are 2 luminal/basal enrichment pathway p-values from Table S2 highlighted in the 2E phosphorylation analysis.

I also cannot find the “Fold enrichment” values (-10 to 10) shown in the legend of 2E, and the significance values do not make sense. Please revisit these enrichment analyses, adding in additional data or clarifying headers. It is challenging as a reviewer to rigorously review this analysis in its present form.

A last point on the enrichment analyses, given the importance of being able to glean biological insight from this collection of analyses it would be beneficial to include a brief discussion on whether the pathway results from GO term enrichment make sense, given the biology of the samples. There are many instances of RNA splicing pathways showing Basal enrichment. Does that make sense? What is “Animal organ morphogenesis” and why would we expect that to be one of the most differentially phosphorylated “pathways” in the basal condition compared to luminal? If these enrichments are not logical, a suggestion might be to utilize a smaller library of pathways to clarify the message.

Figure 2E: Formatting for acetylome figure is different than proteome/phosphoproteome.

Line 212-214: Reads like tumors were lysed in SDS prior to HLA enrichment.

“The human LUAD tumors (50-86 mg cryopulverized tissue) were lysed in SDS and processed with and without initial serial HLA enrichment (Figure 3A).”

The methods state otherwise and lysing in SDS would denature the pMHCs. Please re-word for clarity.

Line 240-242/280-281: Authors note in their rebuttal they have other datasets indicating continuing to add protease inhibitors helps mitigate the negative consequence of the serialization noted here. While they need not include these other datasets if they are destined for another paper, a cell-line example and description of the protocol changes validating the solution would be important to help readers who hope to use this platform get the most data as possible from their samples. Could go here, or in discussion.

Line 268-270: My only remaining concern here is that with the HLA enrichment, you are less likely to detect HLA-A/B/C molecules themselves, eliminating the possibility of judging relative HLA expression at a protein level? The Data viewer suggests this is not the case and that you do measure HLA-A/B/C expression (despite the pulldown depletion). Alternatively, does the presence of HLA-A/B/C in protein expression dataset imply an inefficient HLA pulldown? Please comment.

Line 284: This statement is not accurate, as the above text just described a 16% drop in phosphosite IDs with HLA enrichment. Please adjust claim for accuracy.

Line 285: Use of “significant” implies statistical testing in this context. Please clarify if statistics were used to inform this claim or adjust text.

Line 305: Suggest adding a “Still” or “Nevertheless” ahead of “the detection of these...”

Lines 344-347: Reword to clarify what could be improved upon with larger datasets.

Figure S5. Provide table of Neoantigen predicted versus actual retention times.

Line 352: “as low as 0.01%.” Describe what this is referring to- presumably rank ordered precursor ion abundances from a different study identifying neoantigens from DDA datasets. However, I

cannot find this statistic in the references cited (ref 2, 24, 45). Importantly, the Bassani-Sternberg paper cited (ref 2, only paper with human rank ordered abundances) used at most 1g of tissue (though most samples used less) and saw abundances predominantly in the second or third quartiles of rank ordered abundances, but none as low as 0.01% and actually highlighted that neoantigens were at a comparable level to other antigens, in contrast to the point being made here. Please revisit this claim, and ensure appropriate references are cited to support it.

Line 367: This is too strong of an interpretation given the 10 samples of a single tumor type used in this analysis, and without direct comparison to other LC-MS/MS methods. Please also clarify that this claim, once modified, pertains to discovery immunopeptidomics (not targeted, for example, which will likely have enhanced sensitivity).

Figure 5D/Figure S6/S7: It is great to see the mirror plot/synthetic analyses, and I think Figure 5 would benefit from the mirror plot figures over the figures currently depicted. Still, I have several specific concerns:

-How can authors be positive that neoantigens depicted as a Leucine are not an Isoleucine? Direct retention time comparisons were not (could not) be performed in these samples; therefore I am unsure how authors are certain the neoantigen sequences of SAAADILLL and DQRLALVM were decided. DeepLC cannot decipher this, as the predictions with a leucine and isoleucine will be too similar. Please explain.

-Authors in the rebuttal state that the SAAADILLL spectrum in Figure S6 does not have any ions beyond ~550 m/z. However, the mirror plot of the same peptide in Figure S7 does have ions beyond 550. Is the scan # different in the mirror plot? Can scan numbers be added for transparency? Please also report spectral correlation values described in caption.

-I maintain that the SAAADILLL and ISNDLYLTL epitopes have only moderate spectral support with few b/y ions and very low abundance (S6). I would be wary of stating these two as confident identifications on the same level as the other three.

Line 380: Suggest removing the math equations describing overlaps (ex. $3334+5940=9274$) as this can be easily gleaned from figure 6A.

Line 394: Is it reasonable to assume full overlap is even possible? Can ubiquitination comprehensively cover all the mechanisms by which proteins are degraded and loaded onto pMHCs for class I presentation?

Line 396: To date, have ubiquitination measurements been directly incorporated into prediction algorithms? If so, please cite. If not, perhaps it is a more appropriate claim to state that this data may be useful for prediction algorithms.

Figure S8/Line 402: Could authors extract the relevant values for the B2M plot and generate a bar plot of abundances for RNA expression and protein seen in the heatmap (perhaps next to a plot of HLA IDs) so this plot can be easier interpreted in the context of the claim in line 403? Are authors using the HLA or no IP data to support this data interpretation? Same comment for CD74 analysis.

Of concern, it appears that patient C3N01024 has lower B2M expression than C3L01632 by RNA-seq, yet this patient has one of the highest number of class I pMHCs identified in 3B. Patient C3L-01632 has by far the highest B2M expression by RNA-seq and protein expression yet has the lowest number of class I peptides identified. Patients C23N01045 and C3N00169 have the lowest protein expression (in HLA condition at least) but are not the two patients that authors call out in text. As a result, this analysis and interpretation of data seems flawed and should be addressed.

Line 409: Does having the greatest number of ubiquitination sites (not highest level of ubiquitination) give the most complete explanation of the data in S8 for the CD74 analysis? How do you interpret the wide quantitative variation in ubiquitination sites on CD74 within a single patient? Perhaps the takeaway from this analysis is that a single protein (B2M or CD74) cannot be used as a proxy into presentation levels.

Line 447: Please clarify this was overcome in 1 instance.

Line 498: This claim feels out of place in the discussion, and authors should note that patient C3N-00169 is not homozygous for A11 alleles as per Supp Table 4, therefore this claim needs to be closely inspected for accuracy. If this was a mistake, this type of analysis (looking for mutations in tryptic proteasomal subunit proteins) should be more universally applied to patients with alleles that favor “tryptic-like” peptides to look for general trends (more than 1 patient has an A11 allele in dataset, in fact 2 patients are homozygous for A11). If it is to be included, rather than extrapolating a single truncation mutation to a phenomenon, that while interesting, is lacking significant analytical support and may be best reserved for a study with a more sizable cohort. Alternatively, note this is a compelling hypothesis to test in a larger cohort.

Supp Table 7. Please revisit data format. What numbers are depicted in row 2? Why are columns and rows shown here included in every dataset if the value is always “na”? These tables are very challenging to interpret.

Reviewer #2 (Remarks to the Author):

The manuscript received a considerable number of comments and questions, some of which were of experimental nature, while some others involved detailed data re-analysis. In this respect, the authors have revised quite extensively, with the inclusion of new data, that either showed the initial reviewer comments were not a real concern, or improved on the sections that were previously inadequate. This is commendable, as the authors have clearly taken these input seriously, and in the process made the manuscript a better piece of work.

This reviewer is not completely, but reasonably convinced that this version should be accepted, provided:

1. Novelty of the serial enrichment strategy be moderately toned, because it remains that 96-well formats have been used previously, concatenated enrichment of different PTMs similarly attempted with plenty of success, the sequence of HLA-I/II pulldown has been optimised before (though authors seemed unaware)... etc. Otherwise any shuffling of these existing techniques can be another Nat Comms paper quite easily.
2. If the scope is technical, this should be accurately introduced somewhere in the abstract, to allow readers to decide upfront the level of biological insight to retrieve from the data.
3. A short ending paragraph highlighting the challenges and need for careful integrated analysis is appended.

Response to Reviewers:

Workflow enabling deepscale immunopeptidome, proteome, ubiquitylome, phosphoproteome, and acetylome analyses of sample-limited tissues

Comments from Editor:

REVIEWER COMMENTS

Reviewer #1 (Remarks to the Author, version with images in attached document):

Summary

Here, the authors present a revised manuscript detailing a serial workflow, termed “MONTE,” for obtaining multi-omics datasets from a single tumor sample of limited quantity. While authors have added several additional analyses and details to support the claims in this study, including synthetic peptide validation, integration with published datasets, and expansion of methodological optimization details, I remain convinced that this study is more appropriate for a methods-focused or proteomics-focused journal. The level of methodological novelty described within this study is limited to validating serialization does not (in most cases) negatively impact data quantity or quality. Still, there are examples, noted below, where claims made are not supported by the data, where variation resulting from serialization (ex. Impact on variation in phosphor-datasets with addition of UbiFast) is minimized or not addressed, and where statistics were not employed to rigorously test claims.

Furthermore, by the authors’ own admission, this 10 tumor LUAD cohort is not large enough to demonstrate the utility of integrating these datasets with robust biological insight. When biological insight attempts are made, claims are often not supported by datasets upon closer inspection (examples below). In the response to reviewers, authors highlight the inclusion of Table S1, demonstrating the different methods development studies used to develop this protocol as an example of the methodological advancement. However, a close examination of this table highlights that many of these optimizations do not offer a substantial difference in data quantity. For example, the difference between an S-trap analysis and Urea tryptic proteome analysis resulted in a difference of <200 total proteins identified between 2 of the tested conditions. Yet, this is highlighted in the authors’ rebuttal as a key point of novelty in their method. Furthermore, both methods are routinely used by others in the field, therefore selection of one over the other does not pose a significant improvement in methodology.

- Given the cost and bandwidth needed for method development, we choose to do these experiments in duplicate and provide the associated .raw data to enable readers the ability to make decisions on how best to digest their HLA enriched flow-throughs. Importantly, these experiments were performed without fractionation due to bandwidth and instrument limitations, and therefore likely underestimate the improvement that the S-Trap protocol has over Urea based lysis. Hence, a difference of ca. 200 proteins at the single shot, unfractionated level is significant in our view. We have added this detail to the revised manuscript. From these data, we selected the method that performed the

best in terms of total proteins identified, which was the S-Trap based digestion. These data more importantly demonstrate that the use of S-Trap for digestion after the HLA enrichment with a diverse set of inhibitors to prevent PTM losses does not negatively impact the proteome and downstream PTM-omes.

Pg 4- *“Third, we replaced 8M urea cell lysis with SDS denaturation and digestion on an S-Trap to facilitate removal of detergents present in the native lysis buffer used for HLA IP and confirmed the S-Trap method recovered the most unique proteins in a single shot proteome analysis (Table S1).”*

Other points of novelty highlighted by authors in the response to reviewers include testing different protease inhibitors (point #1) also do not show meaningful differences in IDs (described in detail below), and with n=2 replicates it is challenging to glean clear trends beyond run-to-run variation.

- PTM depletions in the downstream PTM-omes would be observed if inhibitors specific to phosphorylation[PMID: 23749302], ubiquitinylation[PMID: 22505724], and acetylation[PMID: 25953088] were not used. We demonstrate that including a unique, diverse set of protease inhibitors does not negatively impact the HLA immunopeptidome analysis. We are not aware of another manuscript that has reported an evaluation of a broad set of inhibitors that enables downstream ubiquitylome, phosphoproteome, and acetylome.

Point #2 describes performing a class II IP first as a novelty, though studies exploring the impact of Class I>II and Class II>I have been previously published (described below).

- The results reported by Zhang *et al.* [PMID 33331781] are not comparable to the positive pressure protocol adapted from [PMID: 29242379] reported here that leverages an end-over-end incubation at 4°C for HLA immunopurification. In contrast, Zhang *et al.* performed their high-throughput HLA immunopurification on an AssayMap Bravo at room temperature. As shown by Jappe *et al.*[PMID: 33298915], the stability of HLA-peptide complexes is impacted by temperature. Therefore, an HLA immunopurification performed at 4°C is more likely to recover peptides with a broad range of thermostabilities, while an HLA immunopurification performed at room temperature is more likely to not recover peptides with lower thermostability on HLA complexes.
- We cited both of these studies in the revised manuscript to address this concern.

Pg 3-4- *“Here we incorporated a broad set of protease inhibitors specific to each proteome and PTM-ome, used a pan anti-HLA-DR, -DP, and -DQ antibody mixture selected because it performed the best in a duplicate comparison study (Table S1), and reversed the IP order relative to prior publications(Chong *et al.* 2018; Pollock *et al.* 2021; Zhang *et al.* 2021), opting to enrich HLA-II followed by HLA-I to prevent HLA-II peptide contamination in HLA-I data that is length filtered to canonical 8–11mers.”*

Pg 4- *“We selected the semi-automated serial HLA enrichment(Chong *et al.* 2018) instead of previously reported fully automated enrichments(Pollock *et al.* 2021; Zhang *et al.* 2021) to enable the immunopurification to occur with end-over-end incubation at 4°C because the stability of HLA-peptide complexes is impacted by temperature(Jappe *et al.* 2020).”*

Point #3 focuses on a 3:1:1 antibody mixture of class II antibodies as a methodological advancement, however the average difference in the number of unique peptide IDs is again <200, with the 1:1:1 ratio giving the highest individual number of peptide identifications in a single analysis. Again, with analyses performed in duplicate, it is hard to untangle a meaningful difference in peptide IDs versus standard variation in MS analyses.

- Each analysis was performed in duplicate due to cost, sample, and instrument time limitations. The antibody mixture that resulted in the highest average number of peptides identified was used to move forward in the protocol. Most importantly, these analyses show that the use of the HLA-II antibody mix at a 3:1:1 ratio does not cause a negative impact on the immunopeptidome data. We have clarified this in the revised text as follows:
 - **Pg 3-4-** *Here we incorporated a broad set of protease inhibitors specific to each proteome and PTM-ome, used a pan anti-HLA-DR, -DP, and -DQ antibody mixture selected because it performed the best in a duplicate comparison study (Table S1), and reversed the IP order relative to prior publications (Chong et al. 2018; Pollock et al. 2021; Zhang et al. 2021), opting to enrich HLA-II followed by HLA-I to prevent HLA-II peptide contamination in HLA-I data that is length filtered to canonical 8–11mers (Table S1).*

Point #4 describes scaling to a 96-well format for HLA enrichment, though this has been described in various formats in papers from several years ago (PMID 29242379, PMID 34129938, PMID 33331781). Authors also focus on their ability to perform HLA profiling on samples of limited material but overstate the novelty of this element given other published works.

- The two key parameters we employ that distinguishes our method from the cited prior reports for high throughput HLA immunopurification are temperature (4°C) and end-over-end flipping. Unlike PMID 29242379, we perform the HLA immunopurification at 4°C with end-over-end flipping (not a push through stacked filter plates). We believe that this difference results in higher immunopeptidome yield. PMID 34129938 and PMID 33331781 perform their HLA immunopurifications using an AssayMap Bravo at room temperature. In addition, most immunopeptidome studies that use the amount of input we used for the LUAD immunopeptidomes do not perform serial HLA-I and HLA-II enrichments with a 96-well plate format.
- To address the reviewer's concerns, we added citations to the studies that leverage AssayMap Bravo for immunopeptidomics and explain in the revised text why we selected the positive pressure manifold and end-over-end incubation at 4°C:
 - **Pg 4-** *“Fourth, to enable higher throughput and reproducibility, we incorporated an optimized version of a semi-automated, 96-well plate-based HLA immunopeptidomics workflow that enabled parallel desalting of serial HLA-II and HLA-I IP elutions[32]. We selected the semi-automated serial HLA enrichment(Chong et al. 2018) instead of previously reported fully automated enrichments(Pollock et al. 2021; Zhang et al. 2021) to enable the immunopurification to occur with end-over-end incubation at 4°C because the*

stability of HLA-peptide complexes is impacted by temperature(Jappe et al. 2020)."

While authors took care to calibrate the various aspects of this serialization protocol, ensuring the data was not substantially compromised by the addition of HLA profiling and ubiquitination studies, in my view the MONTE method remains a concatenation, with adjustments, of previously published methods by this group, and a limited demonstration of the power of integrating these datasets. While I believe this work warrants publication, I do not believe Nature Communications is an appropriate venue based on methodological novelty and depth of biological insight provided.

Specific points:

Line 38-39: Presented as a universal fact that decisions must be made in limited quantity contexts. Consider re-wording from "Decisions have to be made" to "researchers may need to make decisions" or something similar.

- This line was edited in the revised manuscript.

Pg 1- *"Because patient samples are generally available in only limited amounts, decisions may have to be made as to which 'omic analyses are most desirable and feasible."*

Lines 71-75. I appreciate the author's response to the initial concern raised about the lack of citations supporting the 1g/1B cell reference, however, still feel this section is overlooking other contributions/advancements made in this space in recent years and highlights the maximum number of cells or tissues used in select studies to embellish the advancement of using 50-91 mg of tissue for HLA analyses. For example, Chong et al. (Ref 28) utilized 200M patient derived cells. Bassani-Sternberg et al. (ref 2) used a maximum of 1g of tissue in a single analysis but a minimum of 100 mg, deeply analyzing the class I and class II immunopeptidomes serially by DDA (though class I followed by class II), identifying tumor antigens and neoantigens, as described here. Another study by the Carr group used 100mg of tissue (PMID: 34391888) that combined small tissue amounts with gas and liquid fractionation (very similar to method used in this paper) to extend immunopeptidome depth in class I analyses. It should not be overlooked that PMID 34497125 performed class I profiling in clinical biopsies of similar (or smaller) size, also identifying tumor antigens like shown in the analysis of Figure 4. While these studies did not integrate other 'omics datasets in the MONTE way and the objective of the studies were different, studies like PMID 34497125 and PMID 34391888 should be cited as an example of the movement of the field towards using limited material for HLA studies, while acknowledging that additional methodological development is required to integrate those findings with other 'omics level datasets—hence this publication.

Furthermore, the author's rebuttal states, "We believe that the immunopeptidome depth using 50mg tumor (~2 mg protein) is novel," further stating that the enhanced # of average peptide identifications in this study is evidence of methodological novelty in comparison to PMID 34497125. In this context, more peptide identifications does not translate directly to novelty. A

variety of variables, including most importantly the natural HLA expression in the samples could contribute to the difference in average in identifications, even on the same instrument. Here, the authors perform serial HLA-II/HLA-I enrichment on PDX samples and recover only <600 unique peptides from these limited sample amounts, underscoring the point that the sample itself can play a significant role in dictating the quantity of IDs obtained.

As acknowledged in my initial response, this study provides very impressive depth in immunopeptidome coverage from limited material in the LUAD tissues. However, I remain adamant authors not overstate the individual novelty of the tissue quantity component, or the ability to detect relevant antigens in small sample inputs due to other published works in this area. To fairly and appropriately acknowledge/reference other works which have made iterative progress prior to this work does not take away from the novelty that the MONTE workflow holistically provides. I would strongly recommend the authors revisit this section of the manuscript to most accurately highlight where their methodological advancement sits in the landscape of recent works attempting similar objectives (i.e. gleaning relevant HLA insight from limited tumor tissue samples) and give appropriate credit to other works where deserved.

- In text we state *“There are two additional complications that preclude adding immunopeptidome analysis in a serial processing strategy. The first is that much more sample input has typically been used for immunopeptidomics than for proteomics and PTM-omics to enable detection of low-abundant, clinically relevant antigens such as neoantigens.”* We do not intend to diminish the contributions of others in the immunopeptidomics field working with low level clinical samples. Instead, our intent was to provide evidence that *“more sample input has typically been”* used by many published reports performing discovery immunopeptidome analysis on clinical samples that had the goal of identifying neoantigens. The citations selected here focused on discovery immunopeptidome studies that aim to identify neoantigens from patient tumor samples, similar to what we present for the LUAD tumors. We also included a citation to the study from Marcu *et al.* that generated a benign reference of HLA-presented peptides to improve T-cell-based cancer immunotherapy, as this is a large-scale discovery dataset across diverse clinical autopsy samples. Although PMID 34497125 is already cited in this manuscript (#29), we added this citation to lines 71-75 as it does look at neoantigen identification with the improvement of FAIMS and fractionation for immunopeptidomics. PMID 34497125 presents a method for absolute quantification of MHC peptides, which is not something we are claiming to be able to do. Therefore, we chose not to cite this specific study as we do not want readers to compare our data dependent discovery efforts with more quantitative methods like SureQuant-IsoMHC, as each of these methods have unique goals and may require different input amounts due to differences in their sensitivity.

Line 109: The inclusion of the data viewer is a nice addition. However, the viewer presents a heatmap that is a bit challenging to interpret at present- at least on my laptop screen looking only at EGFR (for example, below) the legends run into each other, and it is hard to tell which header goes with which color. Consider making legends on multiple lines to fit different computer sizes?

- The issue with the legends appears to be a bug with the R package used for rendering the legends on the browser. To address this concern, we used an image underneath the heat map that shows them more clearly and without being cut off in browsers with different window sizes. Of note, the “Download PDF” button produces the best formatting for the heat maps, since these aren’t technically rendered in the browser. The link below has the updated data viewer: <https://rstudio-connect.broadapps.org/CPTAC-MONTE2022/>

When you select to view “all” PTM sides, the “Relative abundan” legend overlays the site IDs (below)

- To address this concern, the “relative abundance” and “HLA” legends were moved underneath the heat map (instead of on the right side overlaying with the site IDs). The maximum width of the heat map was also increased so the site IDs won’t get cut off in the data viewer. The link below has the updated data viewer: <https://rstudio-connect.broadapps.org/CPTAC-MONTE2022/>

Is there a way to hide either the HLAft or noIP samples? I think that could be beneficial for other use cases of this data, especially given the missing values between datasets.

- To address this concern, we suggest the readers sort the heat map by “Experiment”. This will result in all of the samples appearing together for the HLAft or noIP groups. We have also set the browser to default sort by “Experiment” to make visualizing HLAft vs. noIP samples clearer.

For the HLA tables, it would be beneficial to have the typing data of each patient easily accessible in this tool. Integration with HLA-Athena, for example, to report which allele is the best scoring (most likely to present the peptide) per patient would be an additional piece of helpful information not necessarily required for publication, but just sharing some suggestions as I anticipate this could be used in future CPTAC papers.

- The HLA-I typing information has been added to the data viewer, as suggested by the Reviewer [<https://rstudio-connect.broadapps.org/CPTAC-MONTE2022/>]. We did not include the HLA-II typing, as there is some uncertainty of the HLA-II alleles based on arcashLA calling, as stated in the **Figure S2** legend. Specifically, HLA-DRB4 alleles were imputed from known genomic linkages because no determinations were made for these alleles by arcashLA. We confirmed the HLA-I typing reported by arcashLA from RNA-seq data matches the HLA typing reported by *Gillette et al.* [PMID: 36384096] for these 10 LUAD samples, which was called using Optitype (Szolek et al., 2014) from WES data.
- Regarding integration with HLAthena [PMID: 31844290], this is currently out of the scope of this manuscript. To address this concern, we included the best predicted allele from HLAthena into the downloadable table from the data viewer. This information can also be found in **Table S9**.

- HLA-I typing information can be viewed either as a table above the HLA peptide tables (and exported as a table) or viewed by hovering your mouse over the patient identifier in the HLA table.

HLA Table

Enter your gene of interest

EGFR

show HLA-I typing

Getting started

Enter your gene name of interest (official gene symbols, e.g. PRC2A) to view associated HLA-I and HLA-II sequences. Hover over participant names to view HLA-I types, or click the abbreviation "show HLA-I typing" to view as a table.

Download HLA Tables

Download HLA tables to see more information, including HLA typing and HLA allele predictions.

HLA-I Typing

Type	C3L01632	C3L02549	C3N00189	C3N00199	C3N00547	C3N00579	C3N01016	C3N01024	C3N01416	C3N02145
HLA.A1	A*02:05	A*01:01	A*04:03	A*01:01	A*11:02	A*02:07	A*02:07	A*33:03	A*04:03	A*11:01
HLA.A2	A*24:02	A*24:02	A*11:01	A*01:01	A*11:01	A*02:01	A*33:03	A*33:03	A*24:02	A*11:02
HLA.B1	B*44:02	B*07:02	B*15:02	B*44:03	B*05:02	B*01:02	B*50:01	B*50:01	B*40:01	B*54:01
HLA.B2	B*50:01	B*37:01	B*15:12	B*27:05	B*15:02	B*40:01	B*40:01	B*07:05	B*18:02	B*55:02
HLA.C1	C*05:01	C*07:02	C*03:03	C*02:02	C*12:03	C*07:02	C*03:02	C*03:02	C*01:02	C*01:02
HLA.C2	C*06:02	C*06:02	C*08:01	C*16:01	C*08:01	C*15:02	C*01:02	C*15:05	C*07:04	C*12:03

HLA-I Sequences

Sequence	C3L01632	C3L02549	C3N00189	C3N00199	C3N00547	C3N00579	C3N01016	C3N01024	C3N01416	C3N02145
AEGGKPPKIV				✓						
DEVLPQGGP									✓	
IFLENGR						✓				
ITTHGSDVIV		✓								
KPVDGPKS					✓					
MALESILNR							✓	✓		
MALESILNR							✓			
NEITYVRRID										✓
RILEDKRLNR							✓	✓		
STDSRIRIV							✓	✓		

1-10 of 12 rows

Line 131: Note “Prior publications” is plural but only one publication is referenced. Other studies have explored the impact of enrichment order of class I/II in serial (and compared to parallel), reporting class I contamination in the class II pulldown when class I is immunoprecipitated first (Zhang et al. PMID 33331781). This study (or another) should be referenced in line 132 to this point. Zhang et al. also reported class II contamination in the class I data when the class I IP was performed first (Fig 3). Did the authors here perform a rigorous assessment of $I > II$ versus $I > I$? It is clear from other works such as Zhang et al. that the second IP will result in lower data quantity. Please discuss.

- To address this concern, we added two additional citations to PMID 34129938 and PMID 33331781.
- The study PMID 33331781 by Zhang *et al.* is not directly comparable to our method, as it was done on a Bravo AssayMap at room temperature, while our enrichments were performed using end-over-end flipping at 4°C. As shown by Jappe *et al.* [PMID: 33298915], the stability of HLA-peptide complexes is impacted by temperature. It is plausible that the second IP performed at room temperature on a Bravo AssayMap robot would be lower quality due to the impact room temperature likely has on the stability of HLA-peptide complexes.
- Another plausible explanation for class II contamination in the class I data presented by Zhang *et al.* is due to incomplete lysis. This study reported using a less stringent lysis buffer (CHAPS based vs. the Triton and Octyl β-d-glucopyranoside based buffer used in MONTE), which would result in less complete lysis. Incomplete lysis could cause pieces of the membrane with adjacent class I and class II HLA to be isolated. Because we did not perform our study using this CHAPS-based lysis buffer or the Bravo AssayMap, we decided not to draw parallels to this study as these protocols are different from one another.

Line 134: Table S1, section “HLA-II antibody ratios” reports average IDs of n=2 samples using different antibody ratios. The difference in IDs between sample could easily be explained by run-to-run variation or sample handling and does not demonstrate an obvious innovation or

clear case for selecting a 3:1:1 ratio. This variation is clear in other n=2 replicates performed in the “HLA_rawfile_mappings” tab. In fact, the highest number of IDs (narrowly) was the 1:1:1 condition, sample “01.” Without at least n=3 replicates, these data are challenging to interpret, and innovation in this area should not be overstated.

- The text was revised to explicitly state that the condition selected was the best of an average of 2 replicates.

Pg 3---*“Here we incorporated a broad set of protease inhibitors specific to each proteome and PTM-ome, used a pan anti-HLA-DR, -DP, and -DQ antibody mixture selected because it performed the best in a duplicate comparison study (**Table S1**), and reversed the IP order relative to prior publications^(Chong et al. 2018; Pollock et al. 2021; Zhang et al. 2021), opting to enrich HLA-II followed by HLA-I to prevent HLA-II peptide contamination in HLA-I data that is length filtered to canonical 8–11mers.”*

Table S1/Lines 133/“HLA-rawfile_mappings”. More information regarding the different conditions (ex. Concentration/catalogue # of PR-620/621/622) and interpretation of the results would be informative to understand the combination of conditions that were selected to optimize HLA IDs, and so the conditions are reproducible. This is probably most appropriate to add to the supplement.

- To address this concern, the concentration of inhibitors was added to **Table S1**, and the catalogue numbers were added to the **Methods** section. We also corrected a fill down error in Table S1 that resulted in PR-619, which was used for all conditions (reported correctly in the Methods section) was incorrectly reported as PR-620/621/622.

Table S1 “Proteome method”. Authors should list the unique IDs in each replicate, not just combined data, to most transparently present results.

- All raw data were provided for these samples to enable transparency in the MassIVE upload. However, these file mappings were not included in **Table S1**. Therefore, we added a new tab named “ProteomeFileMappings” to better direct readers to this data and address this concern.

Line 166-169: Authors have removed the box and whisker blots and dot plots showing summary and individual correlation measurements and instead reported only median correlations in a table. The original figure 2D/2E (below) raised concerns with both myself and Reviewer #2, as it is clear that while the median correlation coefficients may be similar, there is much higher variation in the correlation coefficients in the +UbiFast method for the phosphoproteome data. The removal of these figures appears an attempt to detract attention from this result, which is not in line with transparent reporting of data. I repeat my initial sentiment, which is that the variation resulting in the addition of Ubitfast should be discussed and the figures illustrating this variation (with individual measurements or box plots) should remain in the manuscript. Again, without separate replicate plexes, it is challenging to interpret in the correlation plots whether the lower R2 is a result of generally higher variation in phosphoproteome methods than proteome, or whether the noise has come from adding UbiFast. The new experiment did not address this concern, as it was all contained within a single 16-plex.

- The box and whisker plots displaying correlation coefficients for +/- UbiFast samples were originally displayed in **Figure 2** with a limited scale y-axis making correlation differences appear significant. The box and whisker plots were removed in the first revision of the paper, instead reporting median correlation coefficient values in **Figure 2B**. The removal of the box and whisker plots was not to detract attention from the result, but to avoid the misinterpretation made by the reviewer that +UbiFast introduces “much higher variation in the correlation coefficients” for serially acquired proteome and phosphoproteome data.

As requested by the Reviewer, we have added correlation coefficient box and whisker plots back to the manuscript as **Figure 2D**. An expanded correlation coefficient range (0-1.0) is used for the y-axis to improve clarity and show that +UbiFast does not significantly affect variation. In addition, we have added a correlation coefficient plot for the acetylome data that was newly generated for the revision.

While I appreciate the author’s inclusion of the experiment depicted in Figure S1, unfortunately the experiment to address whether the # of pY residues decreases with UbiFast was setup within a single 16-plex (+/- UbiFast), greatly diminishing the possibility of detecting a meaningful difference between the two experimental conditions. In the dataset depicted in Figure 2, can the authors report # of pS/pT/pY peptides to address the original question, given these analyses were run in separate plexes?

My other question initially posed in the response to reviewers asked whether there was a difference in the number of phosphorylation sites that are show significantly different ($p < 0.05$, maybe a fold-change cut-off) phosphorylation levels between basal and luminal subtypes between the +/- UbiFast datasets. I pose this question, as I was concerned the higher quantitative variation in the +UbiFast condition would result in some biological insight being masked. For example, perhaps 800 phosphosites show significant difference phosphorylation between the basal and luminal subtypes in the -UbiFast condition, but this drops to 600 in the +UbiFast dataset due to the higher CVs across measurements. If this is not the case (and higher variation is just a consequence of generating this additional valuable dataset but does not significantly impact biological insight), then that would be good to note.

- The distribution of phosphosites for plexes with and without UbiFast has now been added to the main text:
 - **Pg 5-** *“Without UbiFast preprocessing, 24,632 (83 %) of all fully quantified phosphosites were phosphoserine, 4,601 (16 %) were phosphothreonine and 282 (1 %) were phosphotyrosine. When UbiFast was performed, 25,090 (83 %) phosphosites were phosphoserine, 4,694 (16 %) were phosphothreonine and 306 (1 %) were phosphotyrosine sites. These results demonstrate that the incorporation of UbiFast does not affect either the depth or distribution of phosphosites detected”*
- As discussed above, +UbiFast does not cause significantly higher variation in serially acquired -omes. To further support this, we analyzed % regulated phosphosites +/- UbiFast showing that UbiFast had 1.5 % less regulated phosphosites (18,498 vs. 18,776).

When separating out the regulated sites into up or down regulated the numbers were also very similar with and without UbiFast as seen in the table below.

	total sig (basal vs. luminal)	sig down	sig up
No UbiFast	18776	9388	9388
UbiFast	18498	9283	9215
overlap	15532	7785	7747
% overlap	82.70%	82.90%	82.50%

Figure 2E. I recognize the enrichment analyses are a way to show that the biological insight gained out of the datasets +/- UbiFast are similar, however more information needs to be included in this figure. How were the 5 highlighted pathways selected? What are the associated p-values with each of these selections? The Table S2 tab is difficult to interpret—there are only 2 significance values reported, which appear to be the p-value calculated between the same subtype, +/- UbiFast. The headers are labeled “signed.Log.P.Value.UbiFT_Basal.over.No_Ubi_Basal” in column M of the GSEA_Phosphoproteome tab, for example. Below are 2 luminal/basal enrichment pathway p-values from Table S2 highlighted in the 2E phosphorylation analysis. I also cannot find the “Fold enrichment” values (-10 to 10) shown in the legend of 2E, and the significance values do not make sense. Please revisit these enrichment analyses, adding in additional data or clarifying headers. It is challenging as a reviewer to rigorously review this analysis in its present form. A last point on the enrichment analyses, given the importance of being able to glean biological insight from this collection of analyses it would be beneficial to include a brief discussion on whether the pathway results from GO term enrichment make sense, given the biology of the samples. There are many instances of RNA splicing pathways showing Basal enrichment. Does that make sense? What is “Animal organ morphogenesis” and why would we expect that to be one of the most differentially phosphorylated “pathways” in the basal condition compared to luminal? If these enrichments are not logical, a suggestion might be to utilize a smaller library of pathways to clarify the message.

- In **Table S2** the appropriate fold enrichment columns have now been added for each comparison (basal vs. luminal) with and without UbiFast. Columns have been more clearly marked as “fold enrichment”, “p-value” and “fdr p-value”. Additionally, the datasets were analyzed against the C2 curated gene set database which includes gene sets containing markers of basal and luminal up and down regulation. These specific gene sets were highlighted to show that the biology present in the different subtypes is maintained with and without UbiFast.

Figure 2E: Formatting for acetylome figure is different than proteome/phosphoproteome.

- The formatting for the acetylome figure has been adjusted to match the proteome/phosphoproteome figure.

Line 212-214: Reads like tumors were lysed in SDS prior to HLA enrichment.

“The human LUAD tumors (50-86 mg cryopulverized tissue) were lysed in SDS and processed with and without initial serial HLA enrichment (Figure 3A).”

The methods state otherwise and lysing in SDS would denature the pMHCs. Please re-word for clarity.

- We thank the reviewer for catching this error. The text has been revised to indicate that the HLA enrichment was performed prior to lysis with SDS.

Pg 6- *"The human LUAD tumors (50-86 mg cryopulverized tissue) were processed with and without initial serial HLA enrichment (Figure 3A). In both cases, S-Trap–based protein digestion[30] was used instead of 8M urea digestion following HLA enrichment because we have previously shown that serial HLA immunopeptidome and downstream whole-proteome analysis required the removal of detergents present in the native lysis buffer used for HLA enrichment[38]."*

Line 240-242/280-281: Authors note in their rebuttal they have other datasets indicating continuing to add protease inhibitors helps mitigate the negative consequence of the serialization noted here. While they need not include these other datasets if they are destined for another paper, a cell-line example and description of the protocol changes validating the solution would be important to help readers who hope to use this platform get the most data as possible from their samples. Could go here, or in discussion.

- We believe that including these data is out of the scope of the current manuscript. To address this concern, we specifically state the steps that we believe should be taken to prevent loss of phosphopeptides in the downstream processing of MONTE. Examples of how we addressed this concern are below.

Pg 7- *"A 16% loss of total phosphosites was observed in the HLA enriched lysates, which we attribute to the combination of the losses from the extra desalting step in UbiFast and the possible decrease of phosphatase inhibitor activity over the 6 h HLA serial enrichment. To improve this in future studies, we plan to implement a second addition of phosphatase inhibitors between the HLA-II and HLA-I enrichments."*

Line 268-270: My only remaining concern here is that with the HLA enrichment, you are less likely to detect HLA-A/B/C molecules themselves, eliminating the possibility of judging relative HLA expression at a protein level? The Data viewer suggests this is not the case and that you do measure HLA-A/B/C expression (despite the pulldown depletion). Alternatively, does the presence of HLA-A/B/C in protein expression dataset imply an inefficient HLA pulldown? Please comment.

- HLA expression is most often quantified using RNA-seq data. The highly polymorphic nature of HLA molecules makes these proteins difficult to quantify by proteomics as digestion with trypsin does not always produce unique, LC-MS/MS detectable peptides suitable for differentiating one HLA allele from another in the sample. It is also plausible that there remains HLA protein after the w6/32 enrichment, as this antibody is conformationally sensitive, and should not capture HLA protein that is not in a mature HLA+B2M complex. Having RNA-Seq from the same sample would be a better method for quantifying the HLA-A, -B, and -C expression levels. To address this concern, we have discussed these technical limitations in the revised text.

Pg 7- *"The highly polymorphic nature of HLA molecules makes these proteins difficult to quantify by proteomics as digestion with trypsin does not always*

produce unique, LC-MS/MS detectable peptides suitable for differentiating one HLA allele from another in the sample. It is also plausible that HLA protein is present after the w6/32 enrichment, as this antibody is sensitive to the amino terminus of human beta2-microglobulin (Shields and Ribaldo 1998), and not all HLA proteins are in mature HLA-peptide complexes. Hence, these proteins were not used in this evaluation. Regardless, the observation that known HLA-I and HLA-II chaperones are not depleted suggests the addition of the serial HLA immunopurification does not have a negative impact on the downstream proteome analysis.”

Line 284: This statement is not accurate, as the above text just described a 16% drop in phosphosite IDs with HLA enrichment. Please adjust claim for accuracy.

Line 285: Use of “significant” implies statistical testing in this context. Please clarify if statistics were used to inform this claim or adjust text.

- The revised text was adjusted to address these concerns to replace the word “significant” with “considerable”.

Pg 7- *“Overall, the HLA-enriched samples capture the same depth of coverage observed in non-HLA-enriched samples and adding this enrichment step up front in a serial workflow does not introduce considerable bias in downstream proteome, ubiquitylome, phosphoproteome, and acetylome.”*

Line 305: Suggest adding a “Still” or “Nevertheless” ahead of “the detection of these...”

- The text was revised as suggested:

Pg 8- *“Nevertheless, the detection of these oncogenic and tumor suppressor proteins across multiple ‘omes from samples that underwent HLA enrichment demonstrates that known biological signals can be recovered using the MONTE workflow.”*

Lines 344-347: Reword to clarify what could be improved upon with larger datasets.

- The text was revised to clarify what could be learned from larger datasets.

Pg 9- *The contrasting nuORF representations also highlight the differences in noncanonical source protein presentation between HLA-I and HLA-II pathways that are not yet fully understood but could be improved upon from data obtained on larger patient cohorts across diverse tissue types, as each tissue type may have unique nuORF expression characteristics.*

Figure S5. Provide table of Neoantigen predicted versus actual retention times.

- These data are present in **Table S9**, tab “neoantigens”, columns “retentionTimeMin” and “tr_pred”. The column descriptions of “retentionTimeMin” and “tr_pred” are present in the “Column_Descriptions” tab.

Line 352: “as low as 0.01%.” Describe what this is referring to- presumably rank ordered precursor ion abundances from a different study identifying neoantigens from DDA datasets. However, I cannot find this statistic in the references cited (ref 2, 24, 45). Importantly, the

Bassani-Sternberg paper cited (ref 2, only paper with human rank ordered abundances) used at most 1g of tissue (though most samples used less) and saw abundances predominantly in the second or third quartiles of rank ordered abundances, but none as low as 0.01% and actually highlighted that neoantigens were at a comparable level to other antigens, in contrast to the point being made here. Please revisit this claim, and ensure appropriate references are cited to support it.

- This statement was intended to convey that neoantigens often represent only 0.01% of the total unique peptide identifications. Therefore, datasets that contain ~10,000 unique peptides are likely the appropriate depth needed to identify neoantigen peptides in a discovery experiment. We revised this text to better clarify this point so it would not be confused with a claim regarding abundance.

Pg 9- *“Historically, detection of neoantigens by LC-MS/MS has required enrichment from either billions of cells or gram levels of tissue, as neoantigens can represent only 0.01% of all unique peptide identifications in data dependent discovery experiments[2,24,45].”*

Line 367: This is too strong of an interpretation given the 10 samples of a single tumor type used in this analysis, and without direct comparison to other LC-MS/MS methods. Please also clarify that this claim, once modified, pertains to discovery immunopeptidomics (not targeted, for example, which will likely have enhanced sensitivity).

- This text was revised to clarify its reference to data dependent acquisition.

Pg 9- *“In general, we observed that patients with high mutation burden and immunopeptidome depth (>10,000 peptides) were most likely to have LC-MS/MS detectable neoantigens when using data dependent acquisition.”*

Figure 5D/Figure S6/S7: It is great to see the mirror plot/synthetic analyses, and I think Figure 5 would benefit from the mirror plot figures over the figures currently depicted. Still, I have several specific concerns:

-How can authors be positive that neoantigens depicted as a Leucine are not an Isoleucine?

- The synthetic peptide sequence was requested to be made to match the neoantigen identification as shown in **Figure 5**. To our knowledge, this is the gold standard for validation of peptide sequences, and the method that was suggested by the reviewers to address how well the experimental sequence matches that of the corresponding synthetic peptide.

Direct retention time comparisons were not (could not) be performed in these samples; therefore I am unsure how authors are certain the neoantigen sequences of SAAADILLL and DQRLALVM were decided. DeepLC cannot decipher this, as the predictions with a leucine and isoleucine will be too similar. Please explain.

- We provide multiple pieces of evidence to support the reported neoantigens. These include the detection of the mutation in the WES data unique to the tumor (not in a blood normal), synthetic peptide spectra supporting the identification, HLATHENA predictions that show each is predicted to bind strongly to at least one allele expressed in the sample, and predicted retention vs. measured retention time analysis. Taken together,

these pieces of evidence strongly support the neoantigen identifications. We have revised the text to replace the word “confirmed” with “supported” to address this concern.

Pg 9- “Neoantigen peptide identifications were supported using both retention time prediction and experimental comparisons of the mass spectra with synthetic peptides (**Figure S5, S7**).”

-Authors in the rebuttal state that the SAAADILLL spectrum in Figure S6 does not have any ions beyond ~550 m/z. However, the mirror plot of the same peptide in Figure S7 does have ions beyond 550. Is the scan # different in the mirror plot? Can scan numbers be added for transparency? Please also report spectral correlation values described in caption.

- We were mistaken with regard to peaks > 550 m/z. **Figure S6** was updated to show the higher mass peaks. Note that this spectrum is at the edge of sensitivity and the m/z digit after the decimal suggests those high m/z peaks are not from another peptide but instead are likely to be noise. In addition, a recent spectrum viewer update enabled us to add the annotations for internal ions to **Figure S7**, which were not shown previously, to further justify our confidence in these neoantigen identifications.
- There is no universal method for the calculation of spectral correlation or the cutoff that would distinguish a true from a false positive. Therefore, we have chosen to show the mirror plots for each neoantigen to enable readers to make their own conclusions as to how confident they believe these identifications are.
- Scan numbers continue to be present in **Figures S6 and S7** (see highlighted screenshots below that shows the location of the scan numbers in each figure). The upper spectra from **Figure S7** are identical scans to those shown in **Figure S6**. We show an example using the SAAADILLL peptide below.

-I maintain that the SAAADILLL and ISNDLYLTL epitopes have only moderate spectral support with few b/y ions and very low abundance (S6). I would be wary of stating these two as confident identifications on the same level as the other three.

- We provide multiple pieces of evidence to support the reported neoantigens. These include the detection of the mutation in the WES data unique to the tumor (not in a blood normal), synthetic peptide spectra supporting the identification, HLAtena predictions that show each is predicted to bind strongly to at least one allele expressed in the sample, and predicted retention vs. measured retention time analysis. Taken together, these pieces of evidence strongly support the neoantigen identifications. Nonetheless, we have softened the text by replacing the word “confirmed” with “supported” to address this concern.

Pg 9- *“Neoantigen peptide identifications were supported using both retention time prediction and experimental comparisons of the mass spectra with synthetic peptides (Figure S5, S7).”*

Line 380: Suggest removing the math equations describing overlaps (ex. $3334+5940=9274$) as this can be easily gleaned from figure 6A.

- The revised text was updated to remove these equations.

Line 394: Is it reasonable to assume full overlap is even possible? Can ubiquitination comprehensively cover all the mechanisms by which proteins are degraded and loaded onto pMHCs for class I presentation?

- A reasonable expectation is that a majority of proteins in the cell are degraded by the proteasome and enter the HLA-I processing pathway. Therefore, the overlap between these two ‘omes should be relatively high. We have revised the text to clarify this point.

Pg 10- *“Conversely, we noted only 26% of HLA-I source proteins were identified as ubiquitylated, suggesting that additional ubiquitylome datasets are required to capture all the degraded proteins that enter the HLA-I processing and presentation pathway.”*

Line 396: To date, have ubiquitination measurements been directly incorporated into prediction algorithms? If so, please cite. If not, perhaps it is a more appropriate claim to state that this data may be useful for prediction algorithms.

- We have shown in PMID: 28228285 that the count of ubiquitination sites (previously observed in KG-1, Jurkat, or MM1S cells (Krönke et al., 2015, Krönke et al., 2014, Udeshi et al., 2012, Udeshi et al., 2013), was positively associated with HLA-peptide presentation, consistent with the known role for ubiquitin in delivering proteins to the proteasome. We have revised the text to address this concern and added this citation.

Pg 10- *“Because HLA-I source protein expression levels and their ability to be processed by the proteasomal pathway are important factors for*

presentability(Abelin et al. 2017), both proteome and ubiquitylome datasets are likely useful for incorporation into HLA-I prediction algorithms.”

Figure S8/Line 402: Could authors extract the relevant values for the B2M plot and generate a bar plot of abundances for RNA expression and protein seen in the heatmap (perhaps next to a plot of HLA IDs) so this plot can be easier interpreted in the context of the claim in line 403? Are authors using the HLA or no IP data to support this data interpretation? Same comment for CD74 analysis.

Of concern, it appears that patient C3N01024 has lower B2M expression than C3L01632 by RNA-seq, yet this patient has one of the highest number of class I pMHCs identified in 3B. Patient C3L-01632 has by far the highest B2M expression by RNA-seq and protein expression yet has the lowest number of class I peptides identified. Patients C23N01045 and C3N00169 have the lowest protein expression (in HLA condition at least) but are not the two patients that authors call out in text. As a result, this analysis and interpretation of data seems flawed and should be addressed.

The claims presented in this section are not made across the patients. Instead, each claim is specific to the patients or patients stated, and each claim reports if it is using the RNA and or protein data. The comment made by that reviewer that “Patient C3L-01632 has by far the highest B2M expression by RNA-seq and protein expression yet has the lowest number of class I peptides identified” **is not what is shown in Figure S8** (pasted below) or what is stated in the text (see below). We assume that the reviewer mistook patient C3L-02549 with the adjacent patient C3L-01632 in **Figure S8** when reading the heatmap. To attempt to better convey the B2M expression differences across the patients, we remade **Figure S8** and by sorting the samples from low to high ESTIMATE Immune Score. Using this sorting, the two patients with the lowest ESTIMATE Immune Scores, C3L-01632 and C3N-00199, are also the two patients with low levels of B2M and the lowest HLA-I immunopeptidome recovery. We also toned down the claims in this section to convey that we are looking at trends in specific patients.

Pg 10- “To better understand the variable levels of HLA-I and HLA-II peptides recovered in the LUAD immunopeptidomes, we looked at the trends of B2M and CD74 expression and PTMs across the ‘omes with patients sorted from low to high ESTIMATE immune scores (Figure S8; also available using the data viewer: <https://proteomics.broadapps.org/CPTAC-MONTE2022/>). As expected, patients C3L-01632 and C3N-00199 with low mRNA and protein levels of B2M, a subunit of HLA-I complexes, had the lowest ESTIMATE immune scores and overall low HLA-I immunopeptidome depth.”

Previous Version Figure S8

Participant ID	Tumor Purity	Sample Input (mg)	HLA-I peptides	HLA-II peptides
C3N-02145	62%	74.4	9397	2368
C3N-00199	75%	50	8808	5918
C3N-00169	68%	69.2	9248	7133
C3N-00547	75%	79.2	12954	2600
C3N-00579	62%	85.8	13727	9436
C3N-01016	70%	84.9	13609	8395
C3L-01632	72%	67.4	8278	4609
C3N-01024	75%	90.7	13064	1177
C3N-01416	48%	83.2	12183	9726
C3L-02549	70%	66.6	10590	1123

Updated Figure S8 sorted by Estimate Immune Score

Line 409: Does having the greatest number of ubiquitination sites (not highest level of ubiquitination) give the most complete explanation of the data in S8 for the CD74 analysis? How do you interpret the wide quantitative variation in ubiquitination sites on CD74 within a single patient? Perhaps the takeaway from this analysis is that a single protein (B2M or CD74) cannot be used as a proxy into presentation levels.

- The goal of this analysis was to demonstrate what types of questions could be answered across the 'omes that MONTE provides data for. We believe the concern here is that the claims around this analysis were too strong. Therefore, we have revised the associated text as follows (see underlined portion):

Pg 10-- *To better understand the variable levels of HLA-I and HLA-II peptides recovered in the LUAD immunopeptidomes, we looked at trends of B2M and CD74 expression and PTMs across the 'omes (Figure S8; also available using the data viewer: <https://proteomics.broadapps.org/CPTAC-MONTE2022/>). As expected, B2M, a subunit of HLA-I complexes, had the lowest mRNA and protein expression in patients C3L-01632 and C3N-00199 that had the lowest HLA-I immunopeptidome depth. We also observed HLA-I peptides derived from B2M in 9/10 samples excluding patient C3L-01632. Next, we investigated CD74, a protein essential for HLA-II assembly and stabilization and the source of the CLIP peptides. We observed that the patients with the lowest HLA-II immunopeptidome depth, C3N-01024 and C3L-02549, did not have the lowest CD74 expression, and that the protein and RNA expression levels do not always correlate. Instead, these two patients had the most unique ubiquitination sites on CD74, suggesting that CD74 may be degraded at a higher rate in these patients. Understanding both the expression levels and PTM status of proteins involved in antigen presentation, such as B2M and CD74, may not directly correlate with*

HLA immunopeptidome depth, yet such analyzes do provide insights into the HLA presentation machinery in tumors.

Line 447: Please clarify this was overcome in 1 instance.

- Both HLA binding and stability data for this peptide was reported in *Choi et al.* Therefore, it is highly likely that the low expression of this KRAS mutation resulted in the presentation of corresponding neoantigen because this sequence has high affinity and high stability on the HLA molecule. The text was revised as follows:

Pg 11- *“The lack of KRAS G12V tryptic peptide detection in the proteome suggests that low source protein expression was likely overcome by strong HLA-I binding and stability resulting in neoantigen detection.”*

Line 498: This claim feels out of place in the discussion, and authors should note that patient C3N-00169 is not homozygous for A11 alleles as per Supp Table 4, therefore this claim needs to be closely inspected for accuracy. If this was a mistake, this type of analysis (looking for mutations in tryptic proteasomal subunit proteins) should be more universally applied to patients with alleles that favor “tryptic-like” peptides to look for general trends (more than 1 patient has an A11 allele in dataset, in fact 2 patients are homozygous for A11). If it is to be included, rather than extrapolating a single truncation mutation to a phenomenon, that while interesting, is lacking significant analytical support and may be best reserved for a study with a more sizable cohort. Alternatively, note this is a compelling hypothesis to test in a larger cohort.

- C3N-00169 does express HLA-A11 but is not homozygous as noted by the reviewer. Therefore, we revised the text to correct this error. We also toned down this claim in the revised text.

Pg 12- *“For example, patient C3N-00169 had a truncation mutation, E269*, in the proteasomal subunit PSMB7. We noted this patient expressed an HLA-A11 allele that has a lysine residue in the C-terminal anchor position. This observation could suggest that tryptic proteasomal subunits like PSMB7 may be under selection pressure in patients with HLA-I alleles that favor tryptic-like peptides.”*

Supp Table 7. Please revisit data format. What numbers are depicted in row 2? Why are columns and rows shown here included in every dataset if the value is always “na”? These tables are very challenging to interpret.

- Each tab in **Table S7** contains a table in Gene Cluster Text (GCT) v1.3 format. GCT is a tab-delimited text file format that is convenient for analysis of matrix-compatible datasets as it allows metadata about an experiment to be stored alongside the data from the experiment. It is noted at the top of the “Column Description” tab of **Table S7** that these datasets are in GCT format.

Reviewer #2 (Remarks to the Author):

The manuscript received a considerable number of comments and questions, some of which were of experimental nature, while some others involved detailed data re-analysis. In this respect, the authors have revised quite extensively, with the inclusion of new data, that either showed the initial reviewer comments were not a real concern, or improved on the sections that were previously inadequate. This is commendable, as the authors have clearly taken these input seriously, and in the process made the manuscript a better piece of work.

This reviewer is not completely, but reasonably convinced that this version should be accepted, provided:

1. Novelty of the serial enrichment strategy be moderately toned, because it remains that 96-well formats have been used previously, concatenated enrichment of different PTMs similarly attempted with plenty of success, the sequence of HLA-I/II pulldown has been optimised before (though authors seemed unaware)... etc. Otherwise any shuffling of these existing techniques can be another Nat Comms paper quite easily.

- To address this concern, we have revised the manuscript to ensure we are not overstating the novelty of the MONTE serial enrichment strategy and toned down claims throughout.

Examples:

Pg 3- *“The peptide flow-throughs of the UbiFast enrichment step containing unlabeled, non-K-ε-GG peptides are further processed for deep-scale and highly multiplexed measurement of the proteome, phosphoproteome, and acetylome using previously described methods.”*

Pg 3- *“Four ~~major~~ changes were made to previously reported serial multi-omic enrichment protocols^{3,4,17,18,20,21} each of which were evaluated to ensure that each proteomic data type was not significantly impacted.”*

Pg 7- *“Overall, the HLA-enriched samples capture the same depth of coverage observed in non-HLA-enriched samples and adding this enrichment step up front in a serial workflow does not introduce considerable ~~significant~~ bias in downstream proteome, ubiquitylome, phosphoproteome, and acetylome.*

Pg 10- *“To better understand the variable levels of HLA-I and HLA-II peptides recovered in the LUAD immunopeptidomes, we looked at the trends of B2M and CD74 expression and PTMs across the ‘omes (Figure S8; also available using the data viewer: <https://proteomics.broadapps.org/CPTAC-MONTE2022/>).*

Pg 10- *“Instead, these two patients had the most unique ubiquitination sites on CD74, suggesting that CD74 may ~~be~~ ~~likely~~ ~~being~~ degraded at a higher rate in these patients. Understanding both the expression levels and PTM status of proteins involved in antigen presentation, such as B2M and CD74, may not directly correlate with HLA immunopeptidome depth, yet such analyzes do provide insights into the HLA presentation machinery in tumors.”*

Pg 11- “The lack of KRAS G12V tryptic peptide detection in the proteome suggests demonstrates that low source protein expression was likely overcome by strong HLA-I binding and stability resulting in neoantigen detection.”

Pg 12- “This observation could suggest that tryptic proteasomal subunits like PSMB7 may be under selection pressure in patients with HLA-I alleles that favor tryptic-like peptides.

Pg 12- “After implementing HLA-II and HLA-I immunopeptidomics and ubiquityl enrichment into an established serial proteome and PTM enrichment workflow, we observed high correlation between the proteomes, ubiquitylproteomes, phosphoproteomes, and acetylproteomes in both our breast cancer xenograft and LUAD datasets, showing that additional data layers can be acquired without prejudicing data quality and demonstrating the utility of the MONTE workflow.”

Pg 14- “By combining serial multi-ome enrichments with HLA-I and HLA-II immunopeptidomics into a single workflow, we have provided a method to understanding connections between antigen presentation and protein expression, signaling, protein degradation, and epigenetic regulation based on deep characterization of each single sample, which was only previously possible using parallel workflows that required multiple tissue samples.”

2. If the scope is technical, this should be accurately introduced somewhere in the abstract, to allow readers to decide upfront the level of biological insight to retrieve from the data.

- To address this concern, we have revised the abstract (see underlined text, below) to more clearly communicate that this manuscript is a technical evaluation of the MONTE workflow using a small LUAD cohort.

Serial multiomic analysis of proteome, phosphoproteome, and acetylome provides insights into disease pathology while conserving precious human material. To date, ubiquitylome and HLA peptidome analyses have required separate samples for parallel processing each using distinct protocols. Here we present MONTE, a highly sensitive multi-omic native tissue enrichment workflow that enables serial, deep-scale analysis of HLA-I and HLA-II immunopeptidome, ubiquitylome, proteome, phosphoproteome, and acetylome from the same tissue samples. We demonstrate the technical feasibility of MONTE using in a small cohort of primary patient lung adenocarcinoma (LUAD) tumors. Depth of coverage and quantitative precision at each of the ‘omes is not compromised by serialization and the addition of HLA immunopeptidomics enables identification of putative immunotherapeutic targets such as cancer/testis antigens and neoantigens. MONTE will likely improve our understanding of changes in antigen

presentation, protein expression, protein degradation, cell signaling, cross-talk and epigenetic pathways involved in disease pathology and treatment.

3. A short ending paragraph highlighting the challenges and need for careful integrated analysis is appended.

To address this concern, we previously included the paragraph below in the Discussion that outlines the limitations of our study. We underlined the section that specifically discussed the integration of different data types that we added to the revised manuscript.

Pg 13- *“There are limitations to this study. Although it demonstrates the feasibility and utility of a workflow incorporating HLA-I and -II immunopeptidomics and UbiFast ubiquitylproteomics into a serialized proteomic workflow using a clinically relevant sample set, its pilot-level scale precludes the statistically robust analyses, deep explorations of biology, or compelling assessments of the interplay between characterized ‘omes that the approach is intended to facilitate. Rather than highlighting such underpowered and speculative results, we therefore chose to focus on the added value and interpretable results provided by immunopeptidomic characterization of tumor samples. Recent large-scale cancer proteogenomics analyses have made a compelling case that the integration of proteomic, ubiquitylproteomic and especially phosphoproteomic data with genomic data helps to functionalize genomic aberrations, providing new perspectives on cancer biology and nominating potential therapeutic vulnerabilities[14]. Integration of diverse ‘omics datatypes remains challenging, as each datatype has distinct scaling, normalization, and transformation requirements to enable multi-omic interpretation. Missing values in each -ome is also a limitation, as it may not be the case that genes of interest, their PTMs, or corresponding HLA-I and HLA-II peptides are observable due to stochastic sampling or for biological reasons. It also remains to be shown that the integration of additional layers of data, such as the immunopeptidome, will continue to provide new, interpretable, and actionable insights. The MONTE workflow, when applied to samples from a suitably sized patient cohort, provides the means to test if the integration of the immunopeptidome, proteome, and PTMomes will yield valuable biological insights.”*